# On Predictability of Reinforcement Learning Dynamics for Large Language Models

**Yuchen Cai**[1]*, **Ding Cao**[1]*, **Xin Xu**[3], **Zijun Yao**[2], **Yuqing Huang**[1],
**Zhenyu Tan**[1], **Benyi Zhang**[1], **Guiquan Liu**[1]†, **Junfeng Fang**[2]†
[1]USTC, [2]NUS, [3]HKUST
{caiyuchen,caoding}@mail.ustc.edu.cn

## Abstract

Recent advances in reasoning capabilities of large language models (LLMs) are largely driven by reinforcement learning (RL), yet the underlying parameter dynamics during RL training remain poorly understood. This work identifies two fundamental properties of RL-induced parameter updates in LLMs: (1) Rank-1 Dominance, where the top singular subspace of the parameter update matrix nearly fully determines reasoning improvements, recovering over 99% of performance gains; and (2) Rank-1 Linear Dynamics, where this dominant subspace evolves linearly throughout training, enabling accurate prediction from early checkpoints. Extensive experiments across 8 LLMs and 7 algorithms validate the generalizability of these properties. More importantly, based on these findings, we propose AlphaRL, a plug-in acceleration framework that extrapolates the final parameter update using a short early training window, achieving up to 2.5× speedup while retaining >96% of reasoning performance without extra modules or hyperparameter tuning. This positions our finding as a versatile and practical tool for large-scale RL, opening a path toward principled, interpretable, and efficient training paradigm for LLMs. Our model and code will be available at: https://github.com/caiyuchenustc/Alpha-RL.

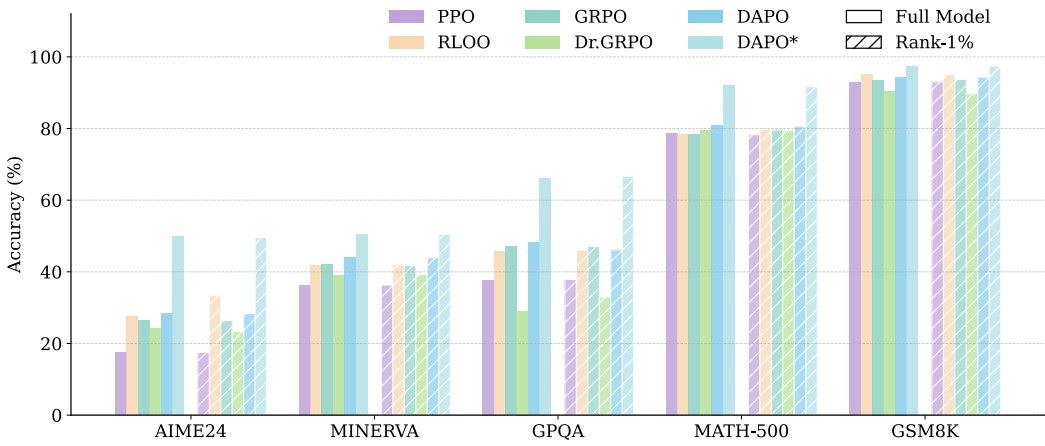

Figure 1: Comparison between RL-trained models and their Rank-1% parameter update counterparts across five reasoning benchmarks. The results demonstrate that retaining only the Top 1% of the parameter update matrix is sufficient to recover the reasoning gains achieved by RL-trained models. More detailed experimental settings and results are exhibited in Section 2. Best viewed in color.

---

*These authors contributed equally to this work.
†Corresponding author: gqliu@ustc.edu.cn, fjf@mail.ustc.edu.cn

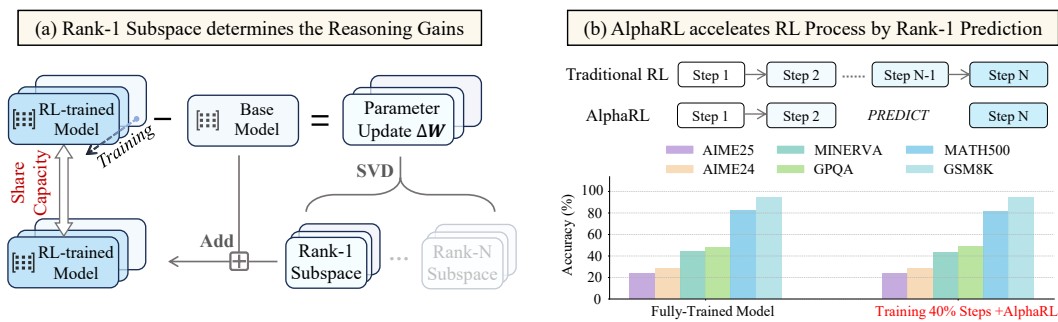

Figure 2: Overview of our key findings and method. (a) Rank-1 Dominance: The majority of reasoning improvements induced by RL can be captured by the Rank-1 Subspace of the parameter update $\Delta W$, which throughout the RL training process. (b) AlphaRL: Leveraging Rank-1 Linear Dynamics, AlphaRL predicts the trajectory of the Rank-1 Subspace, allowing models to reach final performance with fewer RL training steps. Best viewed in color.

# 1 INTRODUCTION

Large language models (LLMs) have witnessed rapid advances in reasoning, a development largely driven by reinforcement learning (RL) based training paradigm (OpenAI, 2025; Claude, 2025; Yang et al., 2025). These advances naturally motivate efforts to interpret RL-trained LLMs, leading to studies such as neuron attribution (Bogdan et al., 2025), circuit analysis (Qian et al., 2025), and sparse autoencoders (Galichin et al., 2025).

While effective, these researches mainly focus on post-hoc interpretability (*i.e.*, explaining the endpoints of training), leaving the RL process itself less explored (Wang et al., 2025b; Zhang et al., 2025; Feng et al., 2025). Understanding parameter dynamics during RL is essential: not only for optimizing RL paradigms, but also for shedding light on the emergence of reasoning capabilities. Hence, this work aims to reveal the black-box of RL process by addressing two fundamental questions: **Are RL-guided parameter updates governed by consistent principles, and how do these principles give rise to reasoning capabilities**?

To solve these, we conduct a step-wise analysis of the parameter update matrix $\Delta W$ (*i.e.*, the parameter difference between the RL-trained model and the base model). After applying mathematical tools such as orthogonal subspace projection (Cai & Cao, 2024), we uncover a striking phenomenon: performing singular value decomposition (SVD) (Koren et al., 2009) on $\Delta W$ reveals that the top singular subspace, defined as the *Rank-1 Subspace*, almost entirely determines the reasoning gains from RL. That is, adding only the Rank-1 component of $\Delta W$ to the base model is sufficient to recover nearly all of the RL-trained model's reasoning improvements. More remarkably, this property holds not just at convergence, but at any intermediate step of RL training, as shown in Figure 2 (a). We formalize this finding as **Property 1 (Rank-1 Dominance):** Rank-1 Subspace of $\Delta W$ determines the reasoning gains of LLMs throughout the RL training process.

This property inspired us to probe how the Rank-1 Subspace evolves during RL training. By applying partial least squares (PLS) to track the dimension-wise trajectory of the Rank-1 Subspace across training steps, we observe an almost strictly linear upward trend, with the linear rate metric $R^2$ (Geladi & Kowalski, 1986) exceeding 0.96. Consequently, the Rank-1 Subspace at a target step can be accurately forecast from a short early window of training. We formalize this as **Property 2 (Rank-1 Linear Dynamics):** Rank-1 Subspace evolves approximately linearly with RL training process, yielding high predictability from early-stage checkpoints of RL process.

To validate the generalizability, we conducted extensive experiments across 13 diverse LLMs (ranging from 7B to 32B parameters, *e.g.*, Qwen3 (Yang et al., 2025), Llama3 (Grattafiori et al., 2024), and GLM4 (GLM et al., 2024)) trained with 10 advanced training algorithms (*e.g.*, GRPO (Yu et al., 2025), Dr.GRPO (Liu et al., 2025b), and DAPO (Yu et al., 2025)). Our analysis shows that, for Property 1, the Rank-1 Subspace alone recovers an average of 99.17% of the reasoning capability. For Property 2, (1) the linearity of Rank-1 Subspace's evolution exhibits an average $R^2$ of 0.914, and (2) predictions of its later state based on early-stage states achieve an average error of less than

5%. Crucially, control experiments with alternative training paradigms like supervised fine-tuning and distillation on the same models yielded neither property, demonstrating that these phenomena are distinctive characteristics of the RL process for LLMs. Detailed experimental setups and results are presented in Sections 2 and 3.

These findings provide actionable interpretability for RL in LLMs: since the Rank-1 Subspace governs RL-induced gains (Property 1) and evolves almost linearly over training (Property 2), the trajectory becomes effectively predictable. We therefore introduce **AlphaRL**, a plug-in acceleration scheme. As shown in Figure 2 (b), for any given RL algorithm applied to any LLM, AlphaRL simply requires an early training window to calculate (1) the initial Rank-1 Subspace of $\Delta W$ and (2) its linear growth rate. It then directly predicts the final parameter update that attains the target reasoning performance without running the full schedule. Experiments on the aforementioned models and RL algorithms demonstrate that AlphaRL achieves up to 2.5× acceleration while retaining >96% of the final reasoning capability. Detailed implementation and results are presented in Section 4.

In summary, this paper uncover two laws of parameter dynamics in LLM training process, Rank-1 Dominance and Rank-1 Linear Dynamics, providing a predictive lens on how RL yields reasoning gains. These findings suggest that the complex, multi-step optimization of RL may be governed by a surprisingly simple and low-dimensional core mechanism. Hence, it not only challenges the black-box view of RL, but also opens new avenues for bridging the gap between empirical scaling laws and theoretical understandings of how capabilities emerge. Building on these properties, we introduce AlphaRL, a **"free lunch"** for RL acceleration: it requires no extra modules, hyperparameter tuning, or human intervention, and remains orthogonal to, thus multiplicatively compatible with, existing acceleration paradigms. This positions our finding as a versatile and practical tool for large-scale RL, opening a path toward **principled, interpretable, and efficient** training paradigm for LLMs.

## 2 DOMINANCE OF RANK-1 SUBSPACE (PROPERTY 1)

In this section, our objective is to analyze the effect of the Rank-1 Subspace of the parameter update matrix $\Delta W$ on reasoning gains. In Section 2.1, we first introduce the method for quantifying the contribution of the Rank-1 Subspace. Then, we exhibit experimental setups and main results in Section 2.2. Furthermore, the underlying causes of Rank-1 dominance and deeper analysis are investigated in Section 2.3.

### 2.1 RANK-1 SUBSPACE

We first describe the method for quantifying the contribution of the Rank-1 Subspace to reasoning gains of RL training process. Specifically, performing SVD on $\Delta W$, we have:

$$\Delta W = \sum_{i=1}^{r} \sigma_i \, \boldsymbol{u}_i \boldsymbol{v}_i^\top, \quad r = \text{rank}(\Delta W), \tag{1}$$

where $\sigma_i$ are singular values and $\boldsymbol{u}_i, \boldsymbol{v}_i$ are the corresponding singular vectors. The Rank-1 update matrix is then defined by retaining only the largest singular value $\sigma_1$ and its vectors:

$$\Delta W^{(1)} = \sigma_1 \, \boldsymbol{u}_1 \boldsymbol{v}_1^\top. \tag{2}$$

To ensure consistency in update strength, we rescale $\Delta W^{(1)}$ to match the L2 norm of $\Delta W$:

$$\Delta \hat{W}^{(1)} = \alpha \, \Delta W^{(1)}, \quad \alpha = \frac{\|\Delta W\|_2}{\|\Delta W^{(1)}\|_2}. \tag{3}$$

The evaluation model is then obtained by adding $\Delta \hat{W}^{(1)}$ to the base model. In addition, we also evaluate a Rank-$k$% Subspace strategy, in which only the leading the top $k$% of singular subspaces are retained, in order to consistently study the collective effect of multiple subspaces.

### 2.2 RANK-1 SUBSPACE & REASONING GAINS

**Experiment Seting.** Our experiments are conducted on eight models, covering five RL algorithms, including PPO (Schulman et al., 2017), RLOO (Ahmadian et al., 2024), GRPO (DeepSeek-AI et al.,

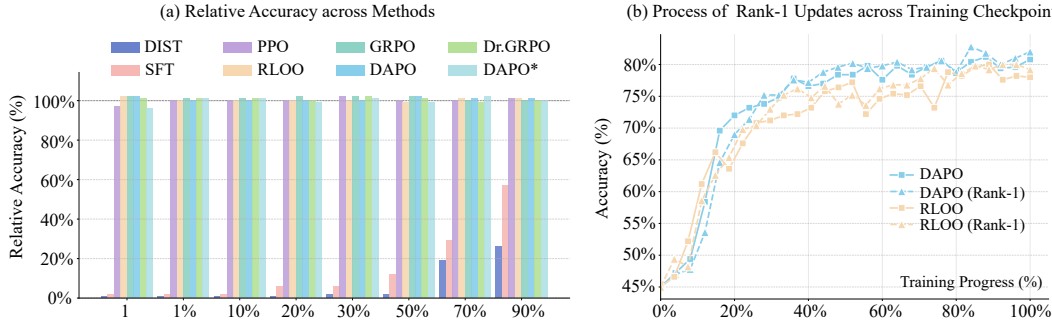

Figure 3: (a) Performance under Rank-1 and Rank-$k$% Subspaces on MATH-500; (b) Performance of the Rank-1 Subspace across training. Best viewed in color.

2025), Dr.GRPO (Liu et al., 2025b), and DAPO (Yu et al., 2025), as well as Distillation (DIST) (Hinton et al., 2015) and Supervised Fine-tuning (SFT) (Ye et al., 2025). We comprehensively evaluate the reasoning performance of these models on six reasoning benchmarks, including AIME24, AIME25 (Ye et al., 2025) and MATH-500 (Lightman et al., 2023), to verify the robustness and generality of our findings. More detailed settings are provided in Appendix B.

**Results on Fully Trained Models.** Figure 1 presents the comparison with Rank-1% Subspace, while Figure 3 shows the results for the Rank-1 Subspace. For clearer presentation, we report the *Relative Accuracy*, defined as the ratio between the accuracy of the Rank-1 reconstructed model and that of the fully trained model. As shown in Figure 3 (a), even a single Rank-1 Subspace is sufficient to recover performance close to that of the fully trained model; in RLOO, GRPO, and DAPO, it can even surpass the fully trained model. This indicates that RL updates are highly concentrated in a few directions, with a single Rank-1 Subspace capable of capturing and reproducing nearly all reasoning improvements. In contrast, SFT and DIST exhibit a strong dependence on subspace rank, requiring more subspaces to achieve performance gains. Notably, unlike methods such as LoRA in SFT, which predefine subspace dimensionality prior to training, our finding holds under a stricter condition: even after full-parameter RL training, reasoning improvements can still be almost entirely captured by only a few subspace directions. Additional results are provided in Appendix 7.

**Results across the RL Process.** We then examine the property of Rank-1 dominance throughout the RL training process, with results shown in Figure 3 (b). We observe that, during the early stages of training, the performance of the Rank-1 Subspace is slightly lower than that of the fully trained model; however, at later checkpoints, its performance can fully match the fully trained model. We hypothesize that this phenomenon arises because, in the early stages, update gradients are relatively dispersed and have not yet concentrated into stable subspace regions. As training progresses, the the RL update directions gradually converge and align with a unified reasoning-enhancement pattern, and the Rank-1 Subspace has already captured the principal components of this pattern, thereby exhibiting stronger effectiveness at later stages. In general, these results demonstrate that the Rank-1 Subspace of $\Delta W$ remains the key factor driving reasoning improvements throughout the RL process.

**Ablation Study.** After establishing the dominant role of the Rank-1 Subspace, we compare the relative contributions of different individual subspaces. As shown in Figure 4 (a), the Rank-1 Subspace significantly outperforms other subspaces, and its performance gradually decreases as the corresponding singular values decline, underscoring its central role in reasoning enhancement. Notably, several subspaces associated with relatively large singular values, although individually less effective than the Rank-1 Subspace, still contribute substantially to reasoning improvements. This indicates that, despite being orthogonal by construction, the functional contributions of these high-singular-value subspaces are largely aligned with the Rank-1 Subspace, collectively reflecting a unified reasoning-enhancement pattern.

**Scaling Experiment.** Finally, to better understand the impact of Rank-1 Subspace strength on reasoning improvements, we conducted an experiment with the scaling factor $\lambda$, where the update rule was modified as: $\Delta \hat{W}^{(1)} = \lambda \cdot \alpha \, \Delta W^{(1)}$. As shown in Figure 4 (b), performance increases rapidly with $\lambda$ and then saturates, peaking at $\lambda \approx 0.7$, slightly below the norm magnitude of the full update. This is consistent with the previous findings: the Rank-1 Subspace has captured the principal

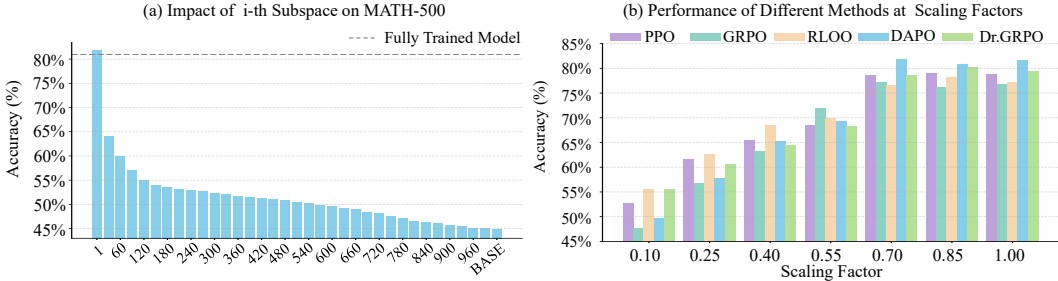

Figure 4: (a) Effect of different single subspaces on performance; (b) Effect of scaling the Rank-1 Subspace updates on performance. Best viewed in color.

components of the unified reasoning-enhancement pattern, with its core effect primarily determined by magnitude. At this level, the core contribution of the Rank-1 Subspace has been largely realized, and further increasing the scaling factor may result in diminishing returns.

## 2.3 Analysis of Rank-1 Dominance

In the previous part, we experimentally observed and analyzed the contribution of the Rank-1 Subspace to reasoning gains. In this section, we attempt to further investigate the underlying causes of this phenomenon. To this end, we begin by comparing the update characteristics of different methods. Specifically, we compute the average L2 norm of $\Delta W$ for RL and compare it with the average L2 norms of $\Delta W$ for SFT and DIST. Additionally, we examine the proportion of the norm of the unscaled Rank-1 Subspace and the Rank-1% subspace relative to the total update norm $\Delta W$.

As illustrated in Figure 5 (a), the update norms for DIST and SFT are found to be one to two orders of magnitude larger than those for RL, indicating that they involve much larger parameter changes during training. In contrast, RL updates show a higher degree of concentration, with the unscaled Rank-1 Subspace and Rank-1% Subspace occupying a larger fraction of the total update norm.

**Distribution Shifts of Embedding Space.** The previous experiments raise an intriguing question: *why is RL, compared to SFT and DIST, able to achieve substantial reasoning improvements with only 1% or even less of the parameter update?* To explore this, we analyze the impact of different training processes on token embeddings in LLMs. By applying PCA for dimensionality reduction and t-SNE for visualization, we observe that the embeddings of DIST and SFT exhibit noticeable global shifts, with DIST showing particularly large deviations for certain tokens, as shown in Figure 5 (b). In contrast, RL methods cause minimal distribution shift of the embedding space. This suggests that the updates in DIST and SFT are not merely adjustments in high-level reasoning pathways, but involve significant global modifications to the lower-level representation space. As a result, even when utilizing the all update information, these methods struggle to consistently improve reasoning performance. In contrast, RL maintains the embedding space largely unchanged, with its reasoning improvements primarily driven by the optimization and adjustment of high-level information flow.

**Approximate Low-rank of $\Delta W$.** It is worth mentioning that, in the above experiments, we discovered a universal *approximate low-rank* (Zhang, 2015) property of $\Delta W$ in RL, which is completely absent in SFT and DIST. Due to space limitations, we provide a detailed discussion of this phenomenon in Appendix C. Furthermore, we propose that the superior properties observed in RL-trained LLMs (*e.g.*, minimizing catastrophic forgetting and improving generalization) may fundamentally arise from this low-rank structure, which plays a pivotal role in the model's ability to effectively retain and adapt learned knowledge. Additionally, we also observed in our experiments the unique role of Rank-1 in guiding the reasoning chain, where modifying a small number of tokens achieves reasoning performance comparable to that of the fully trained model. We recommend interested readers to refer to the detailed results and analysis in Appendix D.

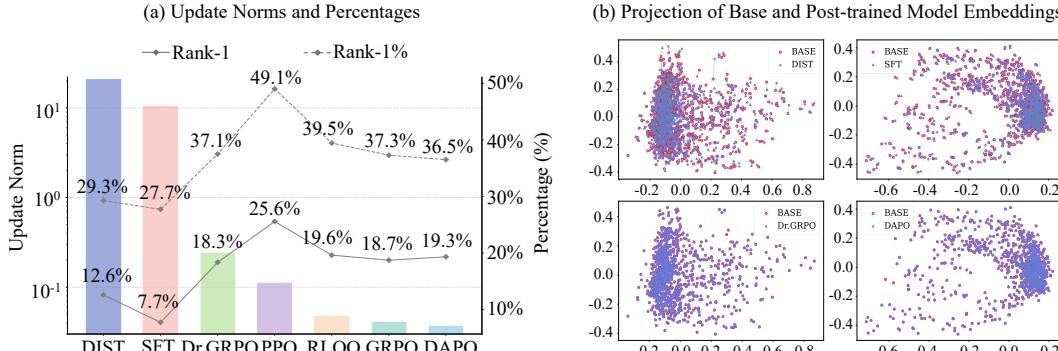

Figure 5: (a) L2 norm of updates across methods and the fraction of update information captured by the unscaled Rank-1 and Rank-1% Subspaces; (b) Effect of different update methods on the embedding layer, with the two embedding representations of the same token connected by gray lines. Best viewed in color.

# 3 Linear Dynamics of Rank-1 Subspace (Property 2)

In the section 2, we analyzed the contribution of the Rank-1 Subspace to reasoning gains and established its dominant role in RL training. Here, we investigate the evolution of the Rank-1 Subspace during RL training process.

## 3.1 Exploring the Dynamics of Rank-1 Subspace

To characterize the evolution of the Rank-1 Subspace during training, we collect the sequence of $\boldsymbol{u}_1$ vectors across $T$ checkpoints for each module: $\mathcal{U}_1 = \{\boldsymbol{u}_1^{(t)}\}_{t=1}^T$, which we refer to as the module's *update trajectory*. Since each $\boldsymbol{u}_1^{(t)}$ lies in a high-dimensional space, we apply PCA for dimensionality reduction and then t-SNE for visualization. As shown in Figure 6 (a), the trajectories exhibit smooth, nearly linear patterns, with color gradients aligned to training progress, indicating the existence of a stable update direction. Limited by space, we describe the perspective and interpretation underlying this procedure in Appendix E.

Furthermore, to quantify whether there is a similar linear relationship between this evolution and reasoning performance, we treat each module's Rank-1 trajectory $\boldsymbol{u}_1^{(t)}$ as the independent variable and the corresponding checkpoint's accuracy $y$ on the reasoning dataset as the dependent variable, performing linear fitting using Partial Least Squares (PLS) regression (Geladi & Kowalski, 1986) and using $R^2$ as a measure of linearity (calculation details can be found in Appendix E). As shown in Figure 6 (b), some modules even achieve $R^2$ values close to 1, indicating that the Rank-1 update directions are strongly correlated with reasoning performance and can be effectively modeled by a fixed linear relationship. These experiments reveal the unique role of Rank-1 Subspace, where it acts as a bridge during training, providing the foundation for the visual correlation between training steps and reasoning performance.

## 3.2 Rank-1 Linearity & Module Importance

Although many modules exhibit high linearity in their Rank-1 Subspace ($R^2$ close to 1), we still observe modules with relatively low linearity. These modules often display fragmented and irregular trajectories, with frequent directional shifts and unstable relations to accuracy. This naturally raises two key questions: (1) Does linearity systematically correspond to the functional roles of modules? (2) Can module contributions to reasoning gains be quantified based on their linearity?

To address the first question, we aggregate $R^2$ values across all modules and layers in Appendix 15. Results show that MLP modules, particularly those in mid-to-high layers, tend to achieve higher $R^2$. A possible reason is that higher-layer modules are closer to the source of the reward signal, allowing them to better retain and utilize reasoning-related update directions. In contrast, self-attention modules generally exhibit lower $R^2$, suggesting noisier or partially redundant update signals.

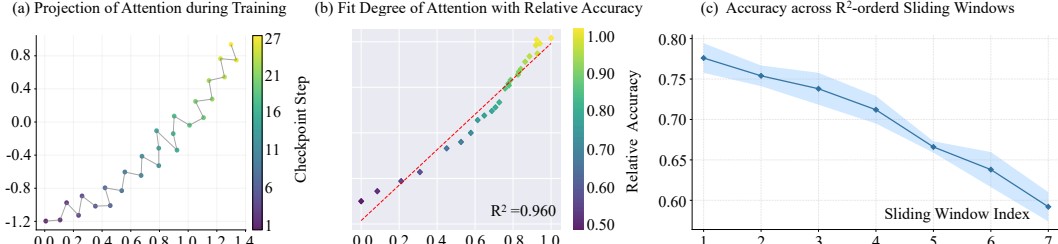

Figure 6: (a) Projection visualization of $\mathcal{U}_1$ trajectories; (b) PLS regression reveals a linear relationship between $\mathcal{U}_1$ and accuracy, with $R^2$ values indicating the strength of fit; (c) Sliding window analysis shows that as the window progresses from 1 to 7, the $R^2$ and prediction accuracy both exhibit a decreasing trend. Best viewed in color.

Based on this observation, we argue that the heterogeneity in linearity reflects differences in functional roles during reasoning. Modules with high $R^2$ and smooth monotonic trajectories are likely key regions where RL allocates effective capacity: after an initial exploratory phase, they converge around a reasoning-enhancing update direction. Conversely, modules with low $R^2$ and irregular trajectories may be only weakly influenced by reward signals and more strongly driven by noisy gradients, preventing them from forming stable performance-related update directions.

To validate the relation between $R^2$ and module contribution, we sort all modules in descending order of $R^2$ and select subsets using a sliding window (window size roughly one-third of all modules, step size about one-seventh). For each window, only the Rank-1 updates of the selected modules are injected into the base model, while other modules remain unchanged. As shown in Figure 6 (c), performance gradually declines as the minimum $R^2$ in the window decreases. This demonstrates that $R^2$ effectively quantifies the functional role of module updates, providing a reliable tool for systematically analyzing module-level contributions to performance during RL training.

## 4 ALPHARL: A FREE ACCELERATION FOR RL TRAINING

The dominance of the Rank-1 Subspace established in Section 2, combined with its linear dynamics demonstrated in Section 3, directly motivates the RL acceleration algorithm: AlphaRL. It leverages the early-training dynamics of Rank-1 Subspace to predict the final parameter update matrix, bypassing the need for full training. Here we provide the detailed acceleration process and results.

### 4.1 RANK-1 UPDATE PREDICTION

As noted in Section 3, we observed a linear relationship between the training trajectory and relative accuracy; however, since the $\boldsymbol{u}_1^{(t)}$ used in the construction are unit vectors, they do not capture the magnitude of the Rank-1 updates. To address this, we construct the scaled Rank-1 vectors, where each vector is scaled by the product of $\alpha^{(t)}$ and $\sigma_1^{(t)}$ to represent the corresponding column of the update matrix $\Delta \hat{\boldsymbol{W}}^{(1)}$. We record the relative accuracy of the corresponding checkpoints and fit the vectors with their relative accuracies using a single-component PLS regression, establishing a linear relationship between them. Given a target relative accuracy $y^*$, AlphaRL obtains the corresponding update vector through inversion. Finally, this update vector is combined with the left singular vector $\boldsymbol{v}_1$ to form the new Rank-1 update for each module.

In this manner, each module's Rank-1 update is guided by its AlphaRL-predicted linear relationship between the scaled Rank-1 trajectory and accuracy. We use the model's test accuracy on MATH-500 and set the target accuracy to $y^* = 1$. The update vectors obtained through inversion are then applied to all datasets for evaluation.

### 4.2 MAIN RESULTS

We use models trained with RLOO, GRPO, and DAPO on Qwen3-8B-Base (Qwen et al., 2025). We then evaluate on six standard mathematical reasoning benchmarks: AIME24, AIME25 (Ye et al.,

Table 1: Performance on reasoning benchmarks at different training stages and their AlphaRL-enhanced variants. The prediction is based on Rank-1 vectors and their corresponding accuracies extracted from checkpoints between the Base model and the current training stage. Within each method block, the highest score is highlighted in "**bold**", and the second highest score is indicated with "__".

| Stage | AIME24 | AIME25 | MATH | MINERVA | GPQA | GSM8K | Avg. |
|---|---|---|---|---|---|---|---|
| *DAPO for the Qwen3-8B Base Model* | | | | | | | |
| Fully Trained Model | **28.54** | **24.17** | **80.95** | **44.02** | 48.23 | 94.35 | **53.38** |
| Training 10% | 12.50 | 7.50 | 70.25 | 32.07 | 36.66 | 84.30 | 40.55 |
| Training 40% | 15.80 | 11.67 | 77.60 | 37.07 | 41.67 | 93.20 | 46.30 |
| Training 10%+AlphaRL | 15.00 | 11.67 | 76.45 | 40.46 | 41.54 | 93.75 | 46.47 |
| Training 40%+AlphaRL | 28.33 | 23.75 | 80.50 | 43.27 | **49.25** | **94.75** | 53.31 |
| *GRPO from the Qwen3-8B Base Model* | | | | | | | |
| Fully Trained Model | **26.40** | **21.67** | 78.25 | **42.19** | **47.10** | **93.50** | **51.52** |
| Training 10% | 9.17 | 8.33 | 64.65 | 31.89 | 36.74 | 85.35 | 39.36 |
| Training 40% | 15.83 | 14.17 | 72.25 | 37.30 | 41.16 | 91.25 | 45.30 |
| Training 10%+AlphaRL | 12.50 | 13.25 | 67.60 | 36.83 | 36.74 | 91.35 | 43.43 |
| Training 40%+AlphaRL | 22.25 | 18.13 | **78.45** | 40.12 | 43.13 | 91.75 | 49.42 |
| *RLOO from the Qwen3-8B Base Model* | | | | | | | |
| Fully Trained Model | **27.50** | 18.33 | **78.25** | **41.90** | **45.82** | **95.10** | **50.82** |
| Training 10% | 11.67 | 8.33 | 57.25 | 35.02 | 38.65 | 83.50 | 39.89 |
| Training 40% | 16.67 | 14.17 | 72.75 | 39.24 | 42.05 | 93.75 | 46.44 |
| Training 10%+AlphaRL | 11.67 | 14.17 | 60.45 | 37.46 | 44.95 | 93.75 | 43.74 |
| Training 40%+AlphaRL | 17.92 | **18.33** | 76.00 | 40.60 | 44.40 | 93.80 | 48.52 |

Table 2: Different model outputs on the centroid geometry problem.

| |
|---|
| **Question:** The medians $AD$, $BE$, and $CF$ of triangle $ABC$ intersect at the centroid $G$. The line through $G$ that is parallel to $BC$ intersects $AB$ and $AC$ at $M$ and $N$, respectively. If the area of triangle $ABC$ is 144, then find the area of triangle $ENG$. |
| **Base Model Answer:** To solve this problem, we need to understand the properties of medians and centroids in a triangle... Area of $ENG = 16 - \frac{16}{3} = \frac{48}{3} - \frac{16}{3} = \frac{32}{3}$. So, the area of triangle $ENG$ is $\boxed{\frac{32}{3}}$. |
| **Fully Trained Model Answer:** Alright, I have this geometry problem in front of me, and I need to find... Area of $ENG = \frac{1}{2} \times 16 = 8$. Final Answer: $\boxed{8}$. |
| **Rank-1 Answer:** Alright, I have this geometry problem in front of me, and I need to find... Area of $ENG = \frac{288}{36} = 8$. Final Answer: $\boxed{8}$. |

2025), MATH-500 (Lightman et al., 2023), Minerva, and GPQA (Rein et al., 2023), with 32 sampled responses per question under temperature $T = 0.6$, and we report average accuracy.

Table 1 presents the reasoning performance across six reasoning benchmarks (*i.e.*, AIME24, AIME25, MATH, MINERVA, GPQA, and GSM8K) at different training stages, with and without AlphaRL-enhanced updates. The results show that AlphaRL significantly improves the model's reasoning performance even at early stages (only 10% of the total training steps), achieving performance comparable to the non-AlphaRL model at 40% of training steps. By the 40% training stage, the AlphaRL-enhanced models almost reach the performance of fully trained models. For instance, RLOO and GRPO models achieve 96% of the reasoning performance of the fully trained model, surpassing the fully trained model on the MATH-500 dataset. For DAPO, AlphaRL reaches a relative accuracy of 102% on the GPQA dataset. Furthermore, we present an example after acceleration in Table 2, which shows that the AlphaRL-enhanced model not only remains consistent with the Fully Trained Model in evaluation metrics but also exhibits largely similar patterns and reasoning approaches in the responses. We present additional experimental results for more models in Appendix A.

Overall, the AlphaRL-enhanced LLMs exhibit significant improvements at all training stages. AlphaRL not only accelerates the training process but also maintains reasoning performance close to that of the fully trained model. On average, AlphaRL accelerates the training process by up to 2.5 times while retaining the vast majority of the reasoning performance, demonstrating its tremendous potential in improving both training efficiency and reasoning capability.

## 5 RELATED WORK

**Reinforcement Learning for LLMs.** Before the emergence of reasoning-capable models such as OpenAI's o1, RL was primarily employed in RLHF to improve instruction-following and alignment with human preferences (Ouyang et al., 2022). More recently, RL with Verifiable Rewards (RLVR) has been proposed as an effective strategy to enhance reasoning in domains such as mathematics and programming (Lambert et al., 2025). OpenAI's o1 was the first to demonstrate that RL can incentivize large-scale reasoning, inspiring subsequent models such as DeepSeek-R1 (DeepSeek-AI et al., 2025), Kimi-K2 (Team et al., 2025), and Qwen3 (Yang et al., 2025). Among these, DeepSeek-R1 stands out for achieving strong reasoning capabilities via the online RL algorithm GRPO and for introducing the "Zero RL" paradigm, showing that reasoning can emerge even without conventional RL fine-tuning. Building on these advances, later approaches, such as DAPO (Yu et al., 2025), VAPO (Yue et al., 2025b), GSPO (Zheng et al., 2025) and CISPO (MiniMax et al., 2025), have further broadened the landscape of RL-based reasoning.

**Emergent Behaviors of Reinforcement Learning.** Yue et al. (2025a) investigated the differences in sampling between base models and RL-fine-tuned models, showing that RL improves sampling efficiency for pass@1 but does not directly enhance reasoning ability. Cui et al. (2025) identified the phenomenon of "entropy collapse" in reinforcement learning, where rapid early convergence causes the model to become overly confident, prematurely degrading its exploratory capacity. Wang et al. (2025a) observed in chain-of-thought reasoning that high-entropy tokens often act as branching points defining multiple potential reasoning paths. Shenfeld et al. (2025) compared RL with supervised fine-tuning and found that RL better preserves the model's original knowledge and capabilities. Finally, Feng et al. (2025) demonstrated in an intent detection task that RL significantly improves generalization compared to SFT, underscoring the value of RL in more challenging scenarios, while Mukherjee et al. (2025) identified the sparse nature inherent in RL.

## 6 LIMITATIONS AND FUTURE WORKS

Despite revealing two simple yet generalizable laws of reinforcement learning in large language models, our study has certain limitations. The conclusions are primarily based on large-scale empirical observations, which uncover universal low-rank dynamics in RL training. However, these findings still lack rigorous theoretical foundations. Future work will incorporate techniques such as neuron attribution and causal tracing to build more formal theoretical models, thereby providing deeper insights into the underlying mechanisms.

Furthermore, while AlphaRL demonstrates the feasibility of predicting later updates from early checkpoints to accelerate RL training, its effectiveness remains constrained by the design and stability of RL algorithms. Future directions include exploring more sophisticated nonlinear forecasting methods, combined with robust reward modeling and optimization strategies, to further enhance acceleration. In addition, AlphaRL may find application in high-cost scenarios such as large-scale agents or multimodal training, where reducing computational overhead is especially critical.

Finally, the Rank-1 property can also be exploited for monitoring training dynamics and serving as a reward signal for reverse optimization of the training process. Future research may explore combining Rank-1 regularities with high-rank corrections to develop more flexible low-rank control methods, thereby advancing the development of efficient reasoning models.

## 7 CONCLUSION

In this work, we uncover two fundamental laws of RL in LLMs: (1) Rank-1 Dominance, where reasoning improvements concentrate in the top singular direction, and (2) Rank-1 Linear Dynamics,

where this direction evolves in a predictable linear manner throughout training. Building on these insights, we introduce AlphaRL, a plug-and-play acceleration method that leverages early checkpoints to forecast later updates, reducing computational cost while preserving reasoning performance close to full training. On average, AlphaRL accelerates the training process by up to 2.5 times while retaining the vast majority of the reasoning performance, demonstrating its potential in improving both training efficiency and reasoning capability.

## 8 ACKNOWLEDGEMENT

The Major Project of the National Social Science Fund of China: "Key Enabling Technologies for Agentic Document and Information Services in the Context of Digital and Intelligent Transformation" (Project No. 23&ZD228)

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

ETHICS STATEMENT

This research does not involve human subjects, personal data, or other sensitive information. All data used are derived from publicly available benchmark datasets and model parameters. The study adheres to principles of research integrity and strictly follows the ICLR Code of Ethics. We declare that there are no potential conflicts of interest or ethical risks.

REPRODUCIBILITY STATEMENT

To ensure the reproducibility of our results, we provide detailed descriptions of the experimental settings in the appendix, including the models used, training methods, hyperparameter configurations, and optimizer settings. All evaluations are conducted on publicly available datasets. We have already released the complete code, data processing scripts, and running instructions to facilitate verification and further exploration by the research community.

USE OF LLMs

We used large language models (LLMs) solely for language polishing and stylistic refinement of the manuscript. Specifically, LLMs were employed to improve clarity, grammar, and readability of the text, without altering the technical content, experimental results, or scientific claims. All ideas, methods, experiments, and conclusions presented in this paper are original contributions of the authors.

## A    ADDITIONAL EXPERIMENT

Table 3: AlphaRL's performance on Qwen3-8B-Base, Qwen3-14B-Base and GLM-4-9B-0414.

| Stage | AIME24 | AIME25 | MATH | MINERVA | GPQA | GSM8K | Avg. |
|---|---|---|---|---|---|---|---|
| *Qwen3-8B-Base* | | | | | | | |
| Training 5% | 7.50 | 5.50 | 62.25 | 27.25 | 31.50 | 78.47 | 35.45 |
| Training 5%+AlphaRL | 10.00 | 6.67 | 69.75 | 31.15 | 34.75 | 81.65 | 38.91 |
| Training 10% | 12.50 | 7.50 | 70.25 | 32.07 | 36.66 | 84.30 | 40.55 |
| Training 10%+AlphaRL | 15.00 | 11.67 | 76.45 | 40.46 | 41.54 | 93.75 | 46.47 |
| Training 20% | 13.17 | 7.50 | 72.25 | 35.07 | 38.50 | 87.54 | 42.34 |
| Training 20%+AlphaRL | 16.67 | 12.50 | 75.00 | 41.25 | 42.85 | 94.35 | 47.10 |
| Training 30% | 14.45 | 11.67 | 74.25 | 36.37 | 39.42 | 91.25 | 44.56 |
| Training 30%+AlphaRL | 23.33 | 20.00 | 78.75 | 41.85 | 44.13 | 93.20 | 50.21 |
| Training 40% | 15.80 | 11.67 | 77.60 | 37.07 | 41.67 | 93.20 | 46.30 |
| Training 40%+AlphaRL | 28.33 | 23.75 | 80.50 | 43.27 | 49.25 | 94.75 | 53.31 |
| Training 50% | 23.33 | 20.00 | 78.50 | 39.87 | 44.33 | 93.75 | 49.40 |
| Training 50%+AlphaRL | 27.67 | 23.33 | **82.00** | 42.85 | 47.75 | **94.50** | 53.01 |
| Fully Trained Model | **28.54** | **24.17** | 80.95 | **44.02** | **48.23** | 94.35 | **53.38** |
| *Qwen3-14B-Base* | | | | | | | |
| Training 5% | 12.50 | 8.33 | 74.30 | 40.44 | 41.04 | 90.39 | 44.50 |
| Training 5%+AlphaRL | 16.11 | 11.11 | 77.75 | 42.27 | 43.25 | 91.25 | 46.96 |
| Training 10% | 13.67 | 10.17 | 77.50 | 41.58 | 43.50 | 91.50 | 46.99 |
| Training 10%+AlphaRL | 21.72 | 15.05 | 81.25 | 42.37 | 44.76 | 92.75 | 49.98 |
| Training 20% | 22.00 | 15.33 | 81.25 | 42.37 | 45.67 | 92.75 | 49.89 |
| Training 20%+AlphaRL | 25.28 | 21.11 | 86.25 | 44.65 | 48.76 | 94.75 | 53.47 |
| Training 30% | 23.33 | 18.33 | 85.00 | 43.67 | 47.34 | 93.50 | 51.53 |
| Training 30%+AlphaRL | 30.69 | 23.89 | 89.50 | 46.50 | 51.73 | 96.75 | 56.84 |
| Training 40% | 28.33 | 20.00 | 88.25 | 44.30 | 49.75 | 94.75 | 54.56 |
| Training 40%+AlphaRL | 38.47 | 31.25 | 91.25 | 47.75 | 52.59 | 97.75 | 59.18 |
| Training 50% | 37.64 | 28.89 | 89.75 | 45.67 | 51.25 | 95.25 | 58.08 |
| Training 50%+AlphaRL | 40.00 | 31.80 | **92.25** | **48.33** | 53.75 | **97.50** | 60.27 |
| Fully Trained Model | **40.50** | **32.63** | 91.75 | 48.33 | **54.50** | 97.50 | **60.87** |
| *GLM-4-9B-0414* | | | | | | | |
| Training 5% | 4.17 | 1.67 | 64.20 | 35.29 | 40.40 | 87.34 | 38.84 |
| Training 5%+AlphaRL | 5.67 | 3.33 | 66.75 | 37.35 | 42.50 | 88.85 | 40.91 |
| Training 10% | 12.50 | 7.50 | 72.25 | 37.25 | 43.50 | 88.50 | 43.58 |
| Training 10%+AlphaRL | 16.67 | 11.67 | 79.50 | 40.25 | 53.25 | 90.75 | 48.68 |
| Training 20% | 15.00 | 13.33 | 79.50 | 39.75 | 44.50 | 90.25 | 47.06 |
| Training 20%+AlphaRL | 20.33 | 17.00 | 83.25 | 43.50 | 57.75 | 92.25 | 52.01 |
| Training 30% | 18.52 | 15.06 | 84.20 | 42.33 | 46.36 | 92.33 | 50.13 |
| Training 30%+AlphaRL | 24.54 | 19.33 | 88.50 | 48.25 | 53.33 | 95.55 | 54.92 |
| Training 40% | 22.67 | 21.50 | 86.50 | 45.50 | 49.35 | 94.00 | 53.92 |
| Training 40%+AlphaRL | 29.33 | 27.25 | 91.50 | 50.25 | 54.75 | 96.25 | 58.89 |
| Training 50% | 24.33 | 22.67 | 87.25 | 46.75 | 50.20 | 94.25 | 54.57 |
| Training 50%+AlphaRL | 30.60 | **30.00** | 91.50 | 50.87 | 55.33 | **97.00** | 59.72 |
| Fully Trained Model | **31.10** | 29.75 | **91.75** | **51.34** | **55.67** | 96.25 | **59.98** |

In Table 3, we report the full performance trajectories of the 9B and 14B models across training stages to demonstrate that the observed patterns persist at larger scales. As shown, AlphaRL exhibits similarly stable extrapolation behavior on these larger models, accurately reproducing the performance gains associated with later-stage updates even when applied at early checkpoints. Taken together, these results further support a central conclusion of this work: the predictability of RL dynamics is not limited to mid-sized models, but continues to hold as model scale increases.

Table 4 extends our analysis to a broader set of base models, including distilled variants and models that have undergone instruction tuning on mathematics-focused datasets. Despite these differences in pretraining objectives and data distributions, the trends observed in the main experiments remain consistent. In particular, both the Rank-1 structure of RL updates and the effectiveness of AlphaRL

Table 4: AlphaRL's performance on Deepseek-R1-Distill-Qwen-7B and Llama3-8B-Instruct.

| Stage | AIME24 | AIME25 | MATH | MINERVA | GPQA | GSM8K | Avg. |
|---|---|---|---|---|---|---|---|
| *Deepseek-R1-Distill-Qwen-7B (After Distillation)* | | | | | | | |
| Training 5% | 50.83 | 39.17 | 91.25 | 50.00 | 38.89 | 93.00 | 60.69 |
| Training 5%+AlphaRL | 51.92 | 40.08 | 92.75 | 51.35 | 40.75 | 93.00 | 61.64 |
| Training 10% | 52.50 | 41.75 | 92.75 | 51.50 | 41.17 | 94.25 | 62.32 |
| Training 10%+AlphaRL | 56.50 | 44.50 | 94.50 | 54.25 | 43.85 | 94.75 | 64.73 |
| Training 20% | 56.50 | 45.75 | 93.75 | 53.75 | 44.25 | 94.75 | 64.79 |
| Training 20%+AlphaRL | 60.00 | 52.25 | 95.50 | 57.85 | 48.25 | 95.50 | 48.23 |
| Training 30% | 59.50 | 51.50 | 95.20 | 56.22 | 47.25 | 95.50 | 67.52 |
| Training 30%+AlphaRL | 64.50 | 57.25 | 96.17 | 60.07 | 51.33 | 96.50 | 70.97 |
| Training 40% | 63.25 | 55.25 | 95.00 | 58.45 | 49.86 | 96.50 | 69.74 |
| Training 40%+AlphaRL | 67.50 | 61.50 | 97.25 | 62.35 | 53.67 | 97.75 | 73.34 |
| Training 50% | 65.75 | 58.85 | 96.50 | 60.65 | 52.27 | 97.25 | 71.87 |
| Training 50%+AlphaRL | 68.25 | **62.50** | 97.25 | **62.58** | **54.16** | 97.75 | 73.75 |
| Fully Trained Model | **68.75** | 62.00 | **97.50** | 61.74 | 54.16 | **98.25** | **73.65** |
| *Llama3-8B-Instruct (After SFT)* | | | | | | | |
| Training 5% | 9.58 | 6.67 | 68.75 | 26.44 | 29.75 | 85.25 | 37.74 |
| Training 5%+AlphaRL | 12.36 | 9.02 | 71.25 | 28.37 | 31.25 | 86.50 | 39.79 |
| Training 10% | 12.63 | 9.16 | 73.00 | 28.37 | 31.75 | 86.25 | 40.19 |
| Training 10%+AlphaRL | 15.97 | 13.06 | 77.75 | 31.76 | 34.54 | 89.25 | 43.72 |
| Training 20% | 20.00 | 15.00 | 78.00 | 31.07 | 34.75 | 89.50 | 45.06 |
| Training 20%+AlphaRL | 22.67 | 20.00 | 84.75 | 34.75 | 37.25 | 92.25 | 48.94 |
| Training 30% | 23.67 | 20.33 | 81.50 | 35.25 | 38.25 | 91.50 | 48.75 |
| Training 30%+AlphaRL | 30.64 | 27.02 | 85.75 | 38.85 | 42.13 | 93.50 | 53.98 |
| Training 40% | 28.33 | 23.67 | 84.25 | 37.50 | 40.75 | 92.25 | 51.46 |
| Training 40%+AlphaRL | 40.75 | 30.00 | 87.50 | 41.50 | 44.25 | 94.50 | 56.75 |
| Training 50% | 31.11 | 27.11 | 86.25 | 37.50 | 42.25 | 92.50 | 52.79 |
| Training 50%+AlphaRL | **39.12** | 30.75 | **87.75** | 42.25 | **43.75** | 94.25 | 56.98 |
| Fully Trained Model | 38.75 | **31.25** | 87.50 | **42.75** | 43.75 | **95.00** | **56.83** |

hold across these diverse model initializations. This consistency suggests that the predictable dynamics identified in this work are not tied to a specific base model configuration, but instead reflect a general property of RL training that persists under architectural variations.

Table 5 and 6 further extends our analysis to an adversarial self-play setting, allowing us to examine whether the regularities identified in this work persist beyond traditional mathematical and reasoning tasks. Specifically, we adopt the Self-Play framework proposed by Liu et al. (2025a), which is conceptually similar to AlphaZero (Silver et al., 2017) in Go and enables a single language model to play both competing roles within the same training process. In this setup, we use Qwen-4B-Base as the base model, train it for roughly 200 checkpoints (saving one checkpoint every two epochs), and evaluate its win rate against Gemini-2.0-Flash-Lite3 throughout training.

Despite the substantial differences from mathematical reasoning—such as the adversarial nature of the environment, the discrete and game-specific dynamics, and the inherently noisy win/loss reward signal—the results exhibit the same trends observed in our main experiments: RL updates consistently maintain a stable Rank-1 structure, and AlphaRL accurately extrapolates late-stage performance improvements using only early-stage checkpoints. These findings indicate that the predictive structure uncovered in this work is not tied to any particular task formulation. Instead, it generalizes beyond reasoning tasks and remains valid even in challenging RL settings characterized by high-variance rewards and strong adversarial interactions.

Table 5: AlphaRL's performance on Kuhn Poker Games.

| Stage | WinRate | Rank-1 WinRate | AlphaRL WinRate |
|---|---|---|---|
| *Qwen3-4B-Base* | | | |
| Training 5% | 3% | 2% | 4% |
| Training 10% | 5% | 5% | 12% |
| Training 20% | 13% | 15% | 28% |
| Training 30% | 27% | 25% | 41% |
| Training 40% | 43% | 44% | 59% |
| Training 60% | 52% | 52% | 64% |
| Training 80% | 61% | 59% | - |
| Training 100% | **65%** | 62% | - |

Table 6: AlphaRL's performance on MATH-500 (after training on Kuhn Poker Games).

| Stage | Full Model | Rank-1 | AlphaRL |
|---|---|---|---|
| *Qwen3-4B-Base* | | | |
| Training 5% | 45% | 45% | 51% |
| Training 10% | 52% | 51% | 60% |
| Training 20% | 56% | 58% | 65% |
| Training 30% | 61% | 61% | 69% |
| Training 40% | 64% | 64% | 75% |
| Training 60% | 68% | 68% | **81%** |
| Training 80% | 73% | 71% | - |
| Training 100% | 77% | 77% | - |

## B  EXPERIMENTAL SETUP

We begin by outlining our experimental setup and relevant definitions. Let $\theta_{\text{init}}$ denote the parameters of a pretrained base LLM. By applying a training method $M$, we obtain the updated parameters $\theta_{\text{full}}$.

In our experiments, we consider the following methods: Distillation (DIST) (Hinton et al., 2015), Supervised Fine-Tuning (SFT) (Ye et al., 2025; Wu et al., 2025), PPO (Schulman et al., 2017), RLOO (Ahmadian et al., 2024; DeepSeek-AI et al., 2025), GRPO (DeepSeek-AI et al., 2025), Dr.GRPO (Liu et al., 2025b), DAPO (Yu et al., 2025), On-Policy Distillation (Agarwal et al., 2024), DPO (Rafailov et al., 2024), and Spiral (Liu et al., 2025a).

For **DIST**, we adopt the distilled model *DeepSeek-R1-Distill-Qwen-7B* and its base model *Qwen2.5-Math-7B*. For **SFT**, we adopt *Qwen3-8B-Base* as the base model, trained on the *DeepMath-103K* dataset with the `LlamaFactory`[1] framework. For **PPO**, we adopt the open-sourced *Open-Reasoner-Zero-7B* and its base model *Qwen2.5-7B*. For **Dr.GRPO**, we adopt the open-sourced *Qwen2.5-Math-7B-Oat-Zero* and its base model *Qwen2.5-Math-7B*. For **RLOO** and **GRPO**, we adopt *Qwen3-8B-Base* as the base model, trained on the *DAPO-Math-17k* dataset with the `Verl`[2] framework. For **DAPO**, we evaluate the following models from small to large:

(1) *7B* — *DeepSeek-R1-Distill-Qwen-7B*, which is a distilled version of *Qwen2.5-Math-7B* , upon which we further trained on *DAPO-Math-17k* with `Verl`.

(2) *8B* — starting from *Qwen3-8B-Base*, trained on *DAPO-Math-17k* with `Verl`.

(3) We perform cold-start training on *Llama3.1-8B-Instruct* with the *DeepMath-103K* dataset as the initialization, and subsequently perform DAPO with `Verl`.

(3) *9B* — we additionally evaluate *GLM-9B*, trained on *DAPO-Math-17k* with `Verl`.

---

[1]https://github.com/hiyouga/LLaMA-Factory
[2]https://github.com/volcengine/verl

Table 7: L2 norm of updates across methods and the fraction of update information captured by the unscaled Rank-1 and Rank-1% Subspaces.

| Method | Offline DIST | Online DIST | Offline SFT | Offline DPO | Online RL DAPO |
|---|---|---|---|---|---|
| **On-Policy Sampling** | No | Yes | No | No | Yes |
| **Constraints on the distribution of $\pi_\theta$** | No | Yes | No | Yes | Yes |
| **Average Update Norm** | 21.24 | 1.46 | 10.75 | 0.79 | 0.01 |
| **Rank-1 fraction** | 12.6% | 10.45% | 7.7% | 11.42% | 19.3% |
| **Rank-1% fraction** | 29.3% | 24.78% | 27.7% | 24.13% | 36.5% |

Table 8: Performance under Rank-1 and Rank-k% Subspaces on MATH-500.

| Method | Offline DIST | Online DIST | Offline SFT | Offline DPO | Online RL DAPO |
|---|---|---|---|---|---|
| Rank-1 | 1.50% | 56.40% | 2.50% | 62.25% | 100.00% |
| Rank-1% | 2.50% | 64.50% | 3.00% | 71.50% | 100.00% |
| Rank-10% | 3.00% | 71.50% | 4.00% | 74.50% | 100.00% |
| Rank-20% | 3.00% | 72.50% | 7.50% | 79.00% | 100.00% |
| Rank-30% | 3.50% | 77.00% | 8.00% | 83.25% | 100.00% |
| Rank-50% | 3.50% | 80.25% | 12.25% | 88.25% | 100.00% |
| Rank-70% | 16.00% | 84.00% | 24.00% | 88.25% | 100.00% |
| Rank-90% | 23.25% | 89.25% | 46.75% | 93.25% | 100.00% |

(4) *14B* — we further evaluate *Qwen3-14B-Base*, trained on *DAPO-Math-17k* with `Verl`.

(5) *32B* — *DAPO-Qwen-32B* trained from the base *Qwen2.5-32B* using *DAPO-Math-17k*.

For **On-Policy Distillation**, We use *Qwen3-8B-Base-Open-Thoughts-On-Policy-Distillation*, whose base model is *Qwen3-8B-Base*.

For **DPO**, We use *Qwen3-8B-Base*, trained on *Math-Step-DPO-10K* with `Verl`.

For **Spiral**, we follow Liu et al. (2025a), and use *Qwen3-4B-Base* to perform adversarial training on Kuhn Poker and Simple Negotiation games.

All of our training runs are conducted on 8× H800 80GB or 16× H800 80GB GPUs until the reward/loss converges.

For **Supervised Fine-Tuning (SFT)**, we adapt our training codebase with `LlamaFactory` (Sheng et al., 2025). We employ full-parameter training in `Float16` precision, with the maximum sequence length set to 20,480 tokens. The training batch size is 1,024 and the mini-batch size is 4, corresponding to 512 gradient accumulation steps. The learning rate is set to $1 \times 10^{-5}$ with warmup, and gradient clipping of 1.0 is applied. We monitor the training loss and terminate training once the loss decreases by less than $2 \times 10^{-1}$ over five consecutive steps. We conduct the SFT training experiment on Qwen3-8B-Base models, using the DeepMath-107K (He et al., 2025) dataset. The chat template for SFT is specified as:

```
User:

{question}

Please reason step by step, and put your final answer within
boxed{}.

Assistant: {CoT}
```

with `<|endoftext|>` serving as the EOS token, where `{question}` is replaced with the specific problem instance and `{CoT}` denotes the chain-of-thought reasoning and final answer provided in the dataset. By training on nearly 100K examples, the model achieves stable convergence, and the final checkpoint is adopted for subsequent experiments.

For **RLOO**, **GRPO**, and **DAPO**, we adapt our training codebase with the `Verl` (Sheng et al., 2025) and follow the corresponding training setups. All methods share the same core configuration: the maximum prompt length is 2,048 tokens and the maximum response length is 20,480 tokens, yielding a total budget of 22,528 tokens. During training, each backward pass uses a mini-batch of 32 samples, and the gradients are accumulated for 16 iterations before a single optimization step is performed, resulting in an effective batch size of 512 under `Float16` precision. Each prompt generates $n=8$ outputs during rollout. The learning rate is set to $1 \times 10^{-6}$ with warmup, and gradient clipping of 1.0 is applied. We monitor the average reward per training batch and terminate training once the reward fails to improve for five consecutive steps.

In addition to the unified configuration described above, each method adopts specific hyperparameter settings in our experiments. For **RLOO**, we use a low-variance KL loss with coefficient 0.001 and disable entropy regularization. For **GRPO**, we set both the high and low clipping ratios to 0.2 and apply a KL loss with coefficient 0.001 following DeepSeek-AI et al. (2025). For **DAPO**, we employ techniques such as clip-higher, dynamic sampling, token-level policy gradient loss, and overlong reward shaping and apply the recommended hyperparameters from Yu et al. (2025): the clipping ratios are set to $\epsilon_{\text{low}} = 0.2$ and $\epsilon_{\text{high}} = 0.28$, KL divergence terms are removed entirely, and each training batch generates up to $512 \times 3$ candidate responses.

We perform RLVR experiments on Qwen3-8B-Base models, using the DAPO-Math-17K (Yu et al., 2025) dataset for training. For this dataset, we employ the built-in chat template, specified as:

```
User:  Solve the following math problem step by step.

The last line of your response should be of the form Answer:
$Answer (without quotes) where $Answer is the answer to the
problem.

{question} Remember to put your answer on its own line after
"Answer:".

Assistant:
```

As in the SFT setting, `<|endoftext|>` serves as the EOS token, where `{question}` is replaced with the corresponding problem instance. We save the checkpoint after each training batch to enable subsequent evaluation experiments.

## C  IN-DEPTH ANALYSIS OF THE LOW-RANK PHENOMENON

We propose that the "low-rank yet effective" update mechanism observed in reinforcement learning (RL) fine-tuning arises from several key factors.

First, most RL fine-tuning methods adopt an on-policy strategy, sampling training data directly from the model's own policy distribution. Shenfeld et al. (2025) suggest that this naturally biases the optimization process toward staying close to the base model in terms of KL divergence, favoring only minor corrections on top of its existing capabilities. Therefore, we argue that RL gradients do not introduce entirely new update directions, but rather reinforce signals already present during pretraining and instruction tuning. As a result, parameter updates concentrate in a few regions, exhibiting a sparse, low-rank structure.

Second, common stabilization mechanisms—such as KL regularization, logits clipping, and gradient clipping—further constrain the magnitude and spread of parameter updates, thereby limiting, to some extent, the enrichment of update information. Importantly, our norm-based analysis (Figure 3(b)) demonstrates that even under such strong constraints, RL achieves substantial improvements in reasoning performance through limited updates. This suggests that performance gains do not rely on large-scale parameter drift but emerge from focused adjustments within a small set of critical subspaces. Regarding the two points discussed above, Table 7 and 8 provide detailed empirical evidence. Both On-Policy Distillation and DPO exhibit clear low-rank structures in their update patterns, demonstrating that the phenomena identified in this work are not unique to RL. Notably, On-Policy RL—combining advantages from both approaches—shows an even more pronounced low-rank property.

Third, prior work shows that updating only about 20% of tokens suffices to match or even surpass full-token updates (Wang et al., 2025a), indicating that reasoning improvements primarily depend on a small set of critical tokens rather than broad, global parameter modifications. These sparse, high-impact token updates may constitute the microscopic origin of the unified reasoning-enhancement pattern in RL: low-rank, highly structured updates effectively concentrate on key tokens, forming dominant update directions in parameter space.

Finally, studies on RL generalization demonstrate that RL-fine-tuned models consistently outperform SFT-fine-tuned models in mitigating catastrophic forgetting and enhancing generalization (Shenfeld et al., 2025; Feng et al., 2025). Our analysis supports this view: RL leverages and reinforces existing gradient signals to activate latent model capabilities, with improvements primarily arising from concentrated adjustments in critical subspaces and minimal overall parameter drift. In contrast, SFT often requires learning task distributions that substantially deviate from the model's intrinsic capabilities, necessitating larger-scale training data and frequently inducing parameter shifts that may lead to catastrophic forgetting.

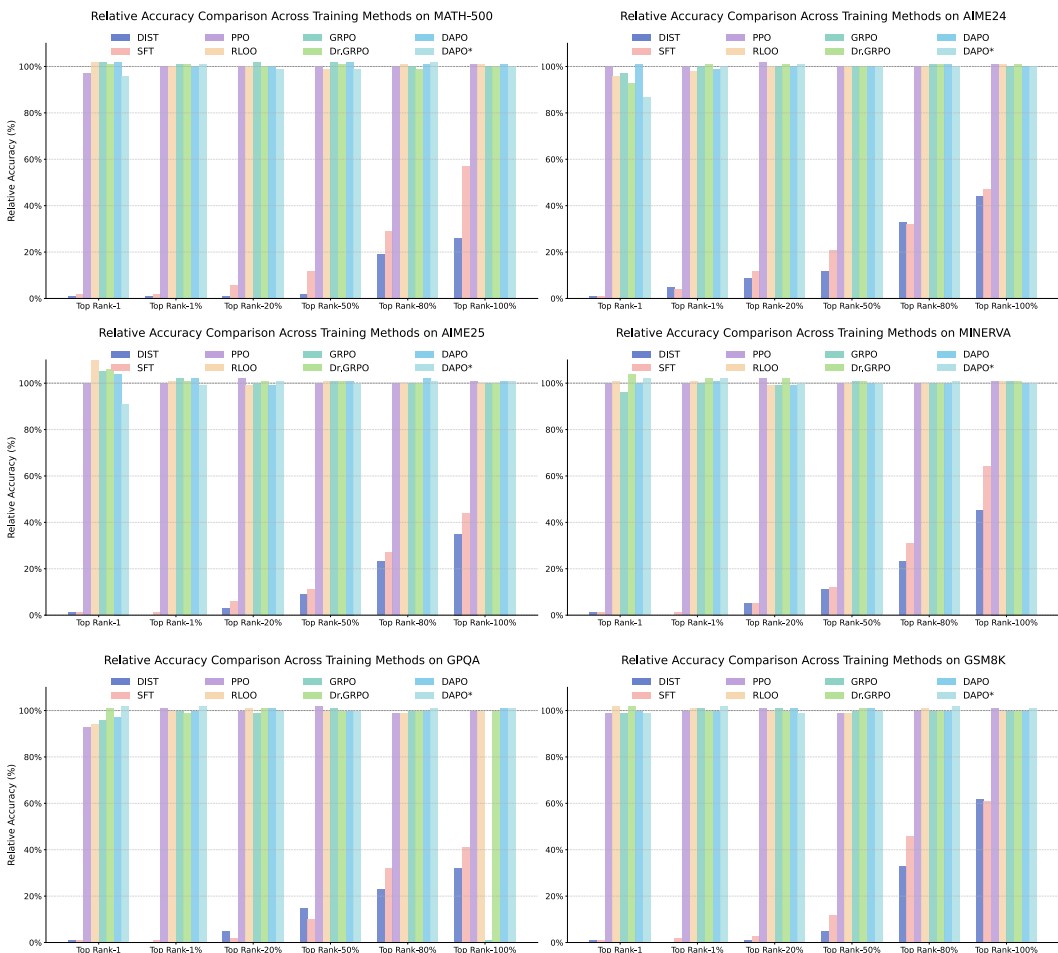

Figure 7: Performance under Rank-1 and Rank-$k$% Subspace on MATH-500, AIME24, AIME25, MINERVA, GSM8K, and GPQA datasets.

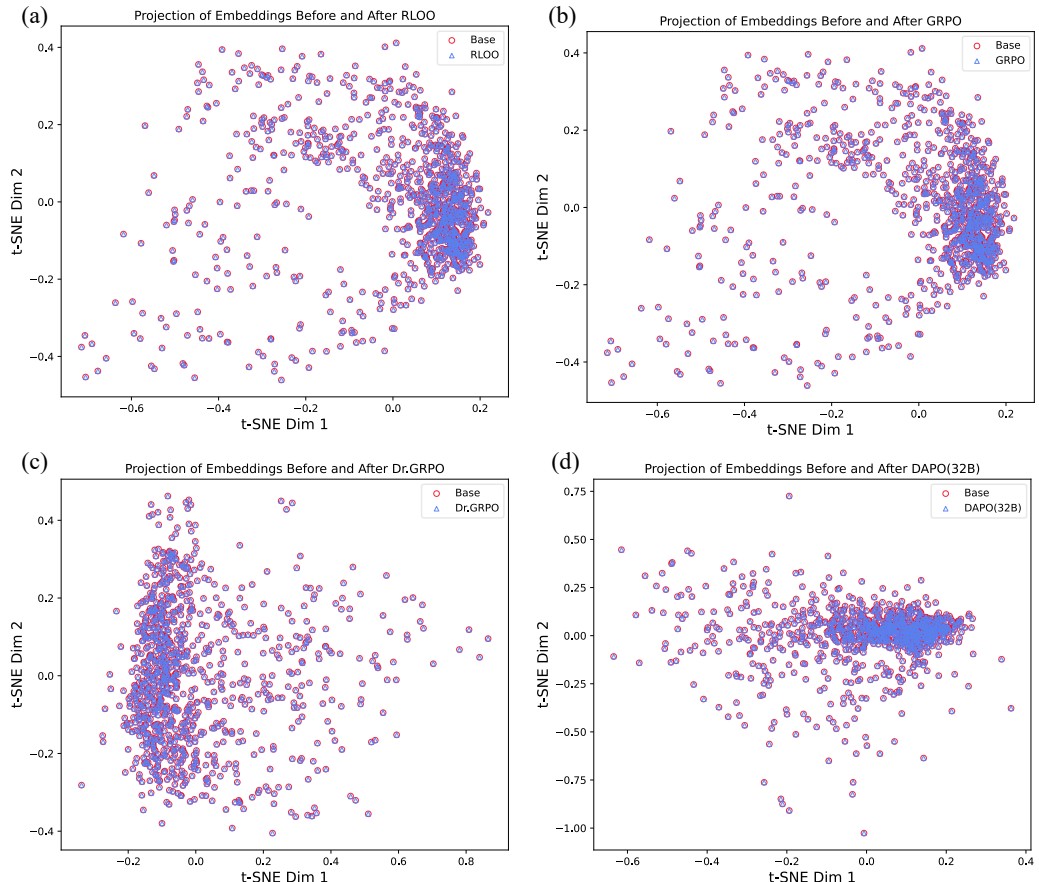

Figure 8: Effect of RLOO, GRPO, Dr.GRPO and DAPO(32B) on the embedding layer, the two representations of the same token are connected with gray lines.

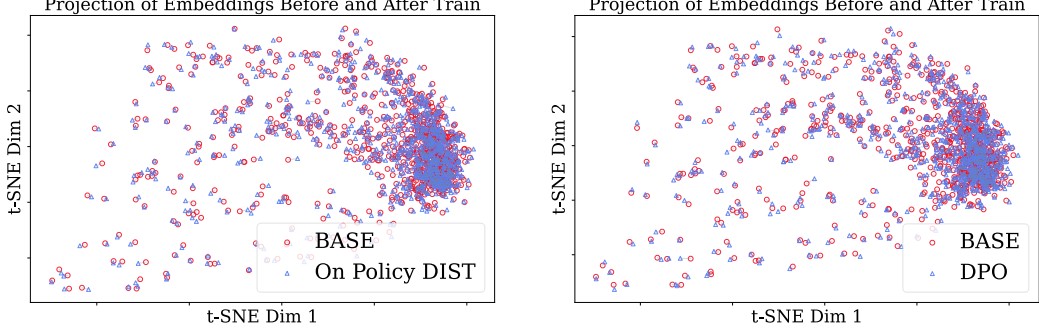

Figure 9: Effect of On Policy Distillation and DPO on the embedding layer, the two representations of the same token are connected with gray lines.

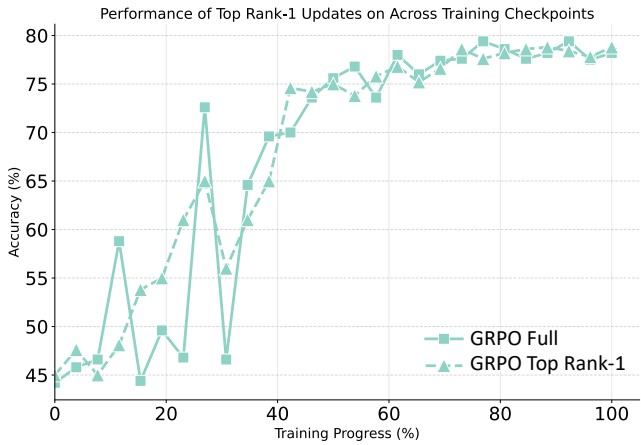

Figure 10: Performance of GRPO Rank-1 Subspace across different training checkpoints.

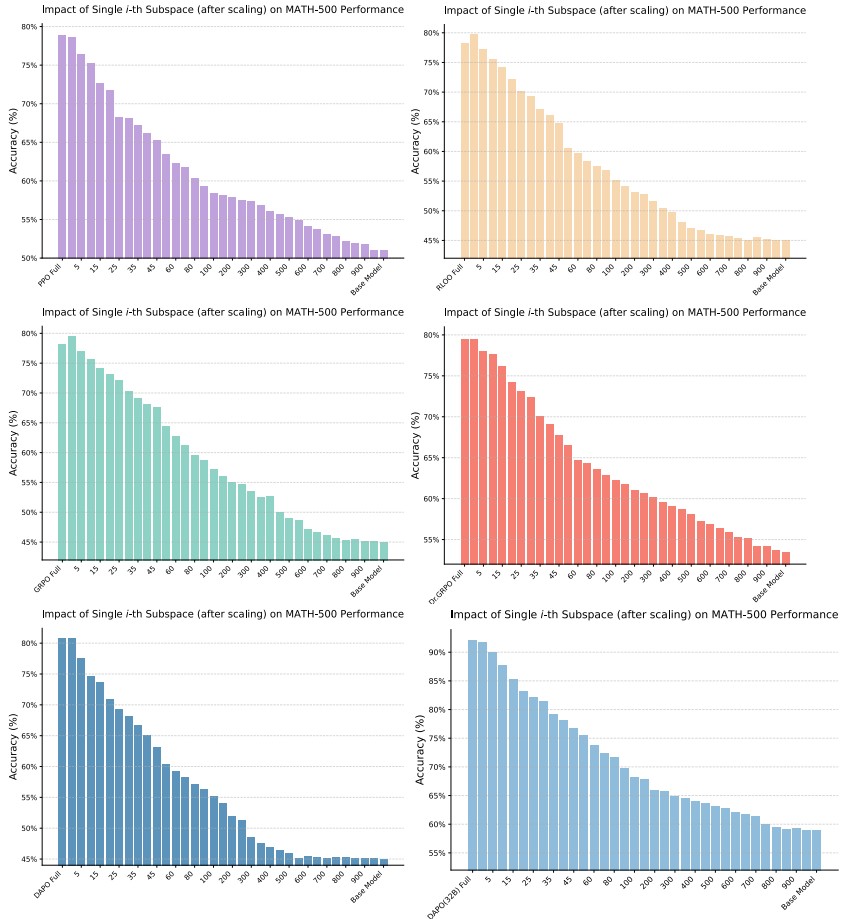

Figure 11: Effect of different single subspace on performance.

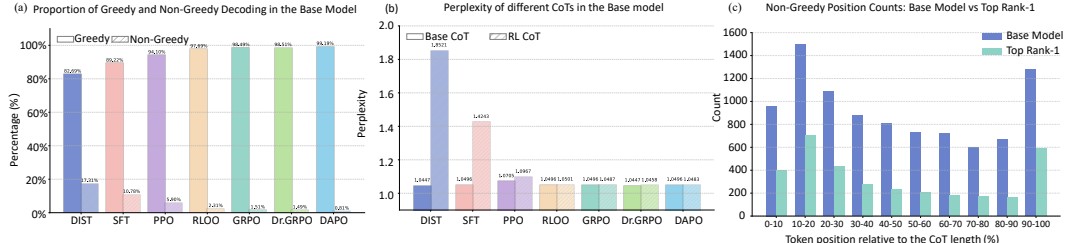

Figure 12: (a) Proportion of greedy versus non-greedy tokens in RL-generated reasoning chains (CoTs), evaluated with the Base Model; (b) Perplexity comparison of CoTs in the Base Model: RL-generated CoTs versus those generated by the Base Model itself; (c) Relative positional distribution of non-greedy tokens in RL-generated CoTs, evaluated under the Base Model and the Rank-1 model.

# D    EXTERNAL MANIFESTATIONS OF RANK-1 DOMINANCE

> **Takeaway**
>
> The Rank-1 Subspace captures key adjustments in the reasoning tokens, recovering the reasoning preferences of fully trained models.

In the previous section, we discovered the naturally emerging low-rank property in RL updates and discussed its potential causes. In this section, we further analyze external manifestations of the Rank-1 Subspace, focusing on how it shapes model behavior.

To investigate how RL training affects reasoning behavior, we conducted two experiments. For each problem, the RL-trained model first generated answers step by step using a greedy strategy, i.e., selecting the token with the highest predicted probability at each step, thereby producing a complete chain of thought. This chain was then fed token by token into the base model, and the base model's greedy predictions were recorded at each step. Positions where the base model's prediction matched the RL model were labeled as greedy, and all others as non-greedy.

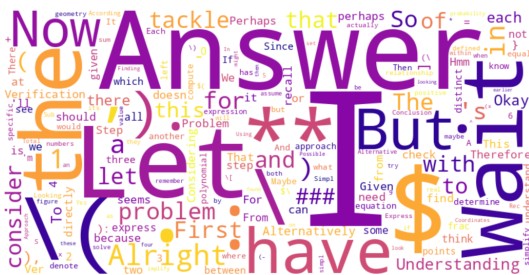

Figure 13: Word cloud of non-greedy tokens. These tokens appear in RL-generated reasoning chains but are not treated as greedily decoded tokens at the corresponding positions by the base model.

As shown in Figure 12 (a), the proportion of non-greedy positions is substantially higher for the DIST and SFT methods compared to RL, indicating that these methods significantly alter the base model's output distribution, whereas RL has a comparatively limited effect. We further measured the perplexity of the base model on the chain-of-thought reasoning generated before and after RL training (both using greedy decoding). The results, shown in Figure 12 (b), reveal that RL training leaves perplexity largely unchanged, while DIST and SFT training lead to a marked increase.

These observations suggest that, unlike DIST and SFT, the reasoning trajectories reinforced by RL are not entirely newly created; rather, they correspond to latent patterns already present in the base model that can be activated. In other words, RL training primarily introduces signals at a small number of critical positions, effectively activating and stabilizing these latent reasoning patterns, thereby enhancing reasoning capabilities without substantially altering the overall output distribution.

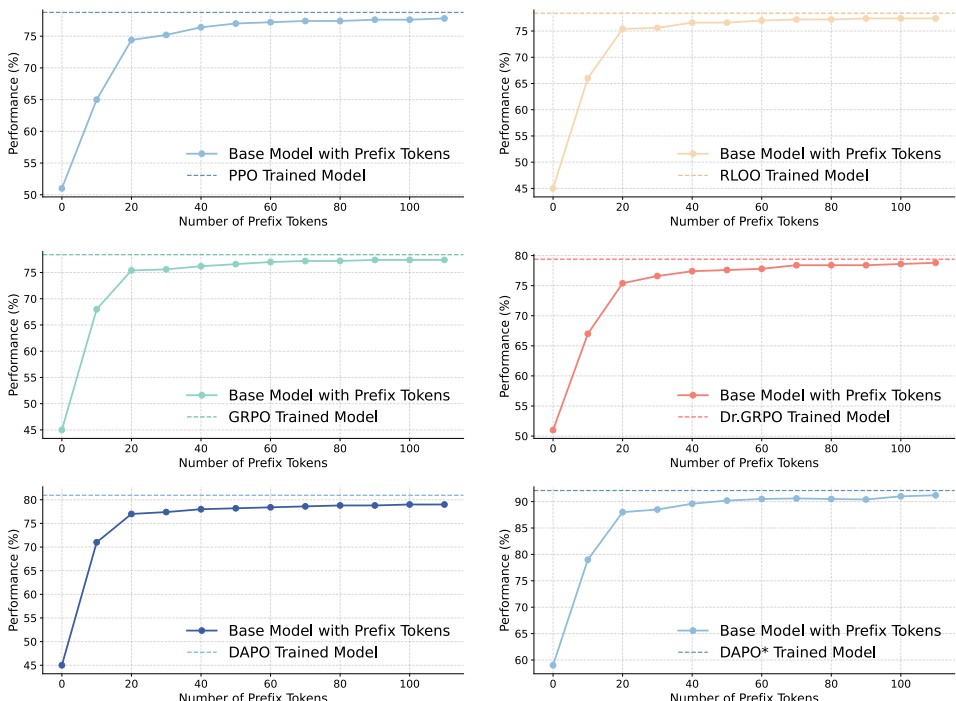

Figure 14: Using prefix tokens generated by the Rank-1 model to guide the base model's reasoning.

We analyzed the positions and characteristics of these non-greedy tokens and identified two predominant patterns in Figure 13. The first pattern corresponds to initial-answer tokens, such as `"Alright"` or `"Let"`, which typically occur at the beginning of the generation and reflect the model's initial understanding of the problem. The second pattern corresponds to reasoning-transition tokens, such as `"But"` or `"Wait"`, which often appear at critical reasoning junctures and indicate adjustments or corrections of intermediate steps.

Moreover, as shown in Figure 12 (c), injecting only the Rank-1 information significantly reduces the number of non-greedy tokens. This indicates that the Rank-1 Subspace reshapes token-level decoding preferences throughout the reasoning process. By adjusting a small set of critical tokens, the Rank-1 information can activate and stabilize latent reasoning trajectories already present in the base model, gradually aligning them with the RL-enhanced reasoning patterns.

To more precisely investigate where these critical adjustments exert their influence, we designed a prefix token experiment. Specifically, we truncated the reasoning chains produced by the Rank-1 model to their first *n* tokens and fed these prefixes into the base model to continue reasoning. As shown in Figure 14, using only the first 20 tokens - approximately the first sentence - allows the base model to approach the performance of the full parameter update.

This result indicates that performance gains primarily arise from the early stage of the reasoning chain, where the problem is represented and the solution strategy is established. They further suggest that the base model inherently possesses strong reasoning capabilities, but its potential is not fully realized, partly because it fails to sample the tokens most critical for understanding the problem. By capturing this key sampling capability, the Rank-1 Subspace effectively activates and stabilizes latent reasoning trajectories, aligning the base model's reasoning behavior more closely with that of the RL-trained model and thereby significantly enhancing performance.

Overall, compared to DIST and SFT, RL induces only limited modifications to the model's reasoning behavior. The Rank-1 Subspace accurately captures the reasoning preferences of the RL-trained model—particularly the critical token-level adjustments—thereby efficiently recovering reasoning capabilities that would otherwise require full-parameter RL training.

## Question

The medians $AD$, $BE$, and $CF$ of triangle $ABC$ intersect at the centroid $G$. The line through $G$ that is parallel to $BC$ intersects $AB$ and $AC$ at $M$ and $N$, respectively. If the area of triangle $ABC$ is 144, then find the area of triangle $ENG$.

## Base Model Answer:

To solve the problem, we need to find the area of triangle $ENG$ given that the area of triangle $ABC$ is 144. Let's break down the problem step by step.

**Identify Key Properties:** - The centroid $G$ divides each median into a ratio of $2 : 1$, with the longer segment being closer to the vertex. - The line through $G$ parallel to $BC$ intersects $AB$ at $M$ and $AC$ at $N$.

**Area Relationships:** - Since $G$ is the centroid, it divides the triangle into three smaller triangles of equal area. Each has area $\frac{144}{3} = 48$. - The line through $G$ parallel to $BC$ creates a smaller triangle $AMN$ that is similar to triangle $ABC$ with side ratio $\frac{1}{3}$.

**Area of Triangle $AMN$:** - Ratio of areas: $\left(\frac{1}{3}\right)^2 = \frac{1}{9}$. - So Area($AMN$) $= \frac{144}{9} = 16$.

**Area of Triangle $ENG$:** - By area decomposition, the final result is $\boxed{16}$.

## RL Trained Model Answer:

Alright, I have this geometry problem in front of me, and I need to find the area of triangle $ENG$ given that the area of triangle $ABC$ is 144. Let's break it down step by step.

**Given:**

- Triangle $ABC$ with medians $AD$, $BE$, and $CF$ intersecting at the centroid $G$.
- A line through $G$ parallel to $BC$ intersects $AB$ at $M$ and $AC$ at $N$.
- The area of triangle $ABC$ is 144.

**Find:**

- The area of triangle $ENG$.

Wait, I think there's a typo in the problem. It mentions triangle $ENG$, but I'm not sure what point $E$ is. Looking back at the problem, it says the medians are $AD$, $BE$, and $CF$, so $E$ is the midpoint of $AC$. But the triangle we're supposed to find the area of is $ENG$, which would involve points $E$, $N$, and $G$. Maybe $N$ is the intersection of the line through $G$ parallel to $BC$ with $AC$. Let's clarify that.

First, let's visualize the triangle and label all the points:

1. **Triangle $ABC$:** A triangle with vertices $A$, $B$, and $C$.
2. **Medians:**
   - $AD$: Median from $A$ to the midpoint $D$ of $BC$.
   - $BE$: Median from $B$ to the midpoint $E$ of $AC$.
   - $CF$: Median from $C$ to the midpoint $F$ of $AB$.
3. **Centroid $G$:** The point where the medians intersect. It divides each median in a 2:1 ratio.
4. **Line through $G$ parallel to $BC$:** This line intersects $AB$ at $M$ and $AC$ at $N$.
5. **Area of $ABC$:** 144.

We need to find the area of triangle $ENG$.

### STEP 1: UNDERSTANDING THE CENTROID AND MEDIANS

The centroid $G$ divides each median in a 2:1 ratio, with the longer part being between the vertex and the centroid. So, for median $AD$, $AG : GD = 2 : 1$. Similarly, $BG : GE = 2 : 1$ and $CG : GF = 2 : 1$.

STEP 2: LINE THROUGH $G$ PARALLEL TO $BC$

We have a line passing through $G$ that is parallel to $BC$. This line intersects $AB$ at $M$ and $AC$ at $N$.

Since the line is parallel to $BC$, triangles $AMN$ and $ABC$ are similar by the Basic Proportionality Theorem (Thales' theorem).

STEP 3: SIMILARITY OF TRIANGLES $AMN$ AND $ABC$

Because $MN \parallel BC$, triangle $AMN$ is similar to triangle $ABC$. The ratio of similarity can be determined by the position of $G$.

Since $G$ is the centroid, it divides the median in a 2:1 ratio. The distance from $A$ to $G$ is $\frac{2}{3}$ of the median $AD$. The line $MN$ is parallel to $BC$ and passes through $G$, so the height of triangle $AMN$ from $A$ is $\frac{2}{3}$ of the height of triangle $ABC$ from $A$.

Therefore, the ratio of similarity between $AMN$ and $ABC$ is $\frac{2}{3}$.

STEP 4: AREA OF TRIANGLE $AMN$

The area of similar triangles is the square of the ratio of their corresponding sides. So, the area of $AMN$ is $\left(\frac{2}{3}\right)^2$ times the area of $ABC$.

$$\text{Area of } AMN = \left(\frac{2}{3}\right)^2 \times 144 = \frac{4}{9} \times 144 = 64$$

STEP 5: LOCATING POINT $N$

Point $N$ is the intersection of the line $MN$ with $AC$. Since $MN \parallel BC$, and $M$ is on $AB$, $N$ divides $AC$ in the same ratio as $M$ divides $AB$.

But we need to find the area of triangle $ENG$. To do that, we need to understand the positions of points $E$, $N$, and $G$.

STEP 6: COORDINATES APPROACH

Maybe assigning coordinates to the points would help. Let's place triangle $ABC$ in a coordinate plane.
Let:
$$A(0,0), \quad B(2b,0), \quad C(2c,2h)$$

Then:
$$D = (b+c, h), \quad E = (c,h), \quad F = (b,0)$$

The centroid is:
$$G = \left(\frac{2b+2c}{3}, \frac{2h}{3}\right)$$

STEP 7: EQUATION OF THE LINE THROUGH $G$ PARALLEL TO $BC$

The slope of $BC$ is:
$$\frac{2h}{2(c-b)} = \frac{h}{c-b}$$

Equation of line through $G$ parallel to $BC$:
$$y - \frac{2h}{3} = \frac{h}{c-b}\left(x - \frac{2b+2c}{3}\right)$$

STEP 8: FINDING POINT $M$ ON $AB$

On $AB$, $y = 0$:
$$-\frac{2h}{3} = \frac{h}{c-b}\left(x - \frac{2b+2c}{3}\right)$$
$$x = \frac{4b}{3}, \quad M = \left(\frac{4b}{3}, 0\right)$$

STEP 9: FINDING POINT $N$ ON $AC$

Equation of $AC$:

$$y = \frac{h}{c}x$$

Intersect with line through $G$:

$$\frac{1}{c}x - \frac{2}{3} = \frac{1}{c-b}\left(x - \frac{2b+2c}{3}\right)$$

After manipulation, one finds contradictions. Let's switch method.

STEP 10: VECTOR GEOMETRY APPROACH

Let $\vec{A} = \vec{0}, \vec{B} = \vec{b}, \vec{C} = \vec{c}$.

Centroid:

$$\vec{G} = \frac{\vec{b}+\vec{c}}{3}$$

Equation of line through $G$ parallel to $BC$:

$$\vec{r} = \frac{\vec{b}+\vec{c}}{3} + t(\vec{c}-\vec{b})$$

Intersection with $AB : \vec{r}_{AB} = s\vec{b}$ gives:

$$t = -\tfrac{1}{3}, \; s = \tfrac{2}{3}, \quad M = \tfrac{2}{3}\vec{b}$$

Intersection with $AC : \vec{r}_{AC} = u\vec{c}$ gives:

$$t = \tfrac{1}{3}, \; u = \tfrac{2}{3}, \quad N = \tfrac{2}{3}\vec{c}$$

STEP 11: COORDINATES OF POINTS

$$M = \tfrac{2}{3}\vec{b}, \quad N = \tfrac{2}{3}\vec{c}, \quad E = \tfrac{1}{2}\vec{c}, \quad G = \tfrac{\vec{b}+\vec{c}}{3}$$

STEP 12: AREA OF TRIANGLE $ENG$

$$\vec{EN} = \tfrac{1}{6}\vec{c}, \quad \vec{EG} = \tfrac{2\vec{b}-\vec{c}}{6}$$

$$\vec{EN} \times \vec{EG} = \tfrac{1}{18}(\vec{c}\times\vec{b})$$

But $\text{Area}_{ABC} = \tfrac{1}{2}|\vec{b}\times\vec{c}| = 144 \implies |\vec{b}\times\vec{c}| = 288$

So:

$$|\vec{EN}\times\vec{EG}| = \tfrac{1}{18}\times 288 = 16$$

Area of $ENG$:

$$\frac{1}{2}\times 16 = 8$$

Final Answer:

$$\boxed{8}$$

---

## Top Rank-1 Answer:

Alright, I have this geometry problem in front of me, and I need to find the area of triangle $ENG$ given that the area of triangle $ABC$ is 144. Let's break it down step by step.

UNDERSTANDING THE PROBLEM

First, let's make sure I understand all the components:

- **Triangle** $ABC$: A triangle with vertices $A$, $B$, and $C$.

- **Medians $AD$, $BE$, and $CF$**: A median of a triangle is a line segment joining a vertex to the midpoint of the opposite side. So:
  - $D$ is the midpoint of $BC$.
  - $E$ is the midpoint of $AC$.
  - $F$ is the midpoint of $AB$.
- **Centroid $G$**: The point where the three medians intersect. It's also the "center of mass" of the triangle.
- **Line through $G$ parallel to $BC$**: This line is parallel to side $BC$ and passes through the centroid $G$.
- **Points $M$ and $N$**: The intersections of this parallel line with sides $AB$ and $AC$, respectively.
- **Area of triangle $ABC$**: Given as 144.
- **Goal**: Find the area of triangle $ENG$.

Note that point $E$ is already defined as the midpoint of $AC$, so triangle $ENG$ has vertices $E$, $N$, and $G$.

### VISUALIZING THE PROBLEM

It's helpful to draw a diagram. Sketch triangle $ABC$, mark midpoints $D$, $E$, and $F$, draw the medians, locate the centroid $G$, and then draw the line through $G$ parallel to $BC$, intersecting $AB$ at $M$ and $AC$ at $N$.

(Diagram representation omitted for clarity)

### PROPERTIES OF THE CENTROID

The centroid divides each median in a 2:1 ratio:

$$AG : GD = 2 : 1, \quad BG : GE = 2 : 1, \quad CG : GF = 2 : 1.$$

### LINE PARALLEL TO $BC$ THROUGH $G$

Since the line through $G$ is parallel to $BC$, triangles $AMN$ and $ABC$ are similar. The ratio of similarity is determined by the position of $G$ along the median. As $G$ divides the median in a $2 : 1$ ratio, the height from $A$ to $MN$ is $\frac{2}{3}$ of the height from $A$ to $BC$. Therefore, the ratio of similarity is $\frac{2}{3}$.

### AREA OF TRIANGLE $AMN$

The area of similar triangles scales with the square of the ratio of corresponding sides:

$$\text{Area}(AMN) = \left(\frac{2}{3}\right)^2 \times \text{Area}(ABC) = \frac{4}{9} \times 144 = 64.$$

### COORDINATES APPROACH

To find the area of triangle $ENG$, we assign coordinates:

$$B(0,0), \quad C(c,0), \quad A(a,b)$$

Midpoints:

$$D\left(\frac{c}{2}, 0\right), \quad E\left(\frac{a+c}{2}, \frac{b}{2}\right), \quad F\left(\frac{a}{2}, \frac{b}{2}\right)$$

Centroid:

$$G\left(\frac{a+c}{3}, \frac{b}{3}\right)$$

Line through $G$ parallel to $BC$ (horizontal) intersects $AB$ at $M$ and $AC$ at $N$.
Equation of $AB$: $y = \frac{b}{a}x$

Intersection with $y = \frac{b}{3}$:

$$M\left(\frac{a}{3}, \frac{b}{3}\right)$$

Equation of $AC$: $y = \frac{-b}{c-a}x + \frac{bc}{c-a}$

Intersection with $y = \frac{b}{3}$:

$$N\left(\frac{2c+a}{3}, \frac{b}{3}\right)$$

Thus, points of interest:

$$E\left(\frac{a+c}{2}, \frac{b}{2}\right), \quad N\left(\frac{2c+a}{3}, \frac{b}{3}\right), \quad G\left(\frac{a+c}{3}, \frac{b}{3}\right)$$

AREA OF TRIANGLE $ENG$

Using the shoelace formula:

$$\text{Area} = \frac{1}{2}\left|x_1(y_2 - y_3) + x_2(y_3 - y_1) + x_3(y_1 - y_2)\right|$$

Plugging in coordinates:

$$x_1 = \frac{a+c}{2}, y_1 = \frac{b}{2}, \quad x_2 = \frac{2c+a}{3}, y_2 = \frac{b}{3}, \quad x_3 = \frac{a+c}{3}, y_3 = \frac{b}{3}$$

Compute:

$$\text{Area} = \frac{1}{2}\left|0 + \frac{-(2c+a)b}{18} + \frac{(a+c)b}{18}\right| = \frac{1}{2} \cdot \frac{cb}{18} = \frac{cb}{36}$$

Given Area$(ABC) = 144$:

$$\frac{1}{2}|cb| = 144 \implies cb = 288$$

Therefore:

$$\text{Area}(ENG) = \frac{288}{36} = 8$$

Final Answer:

$$\boxed{8}$$

## E  LINEAR PROJECTION METHODS

**Constructing Update Trajectories**

Motivated by prior interpretability studies (Geva et al., 2021; Meng et al., 2023), we interpret the tuple $(\boldsymbol{u}_1, \alpha, \sigma_1, \boldsymbol{v}_1)$ of the Rank-1 update $\Delta \hat{\boldsymbol{W}}^{(1)}$ as a *key–value operator*. For any input $\boldsymbol{h}$, the Rank-1 update induces:

$$\Delta \boldsymbol{y}^{(1)} = \Delta \hat{\boldsymbol{W}}^{(1)} \boldsymbol{h} = \alpha \, \sigma_1 \, \boldsymbol{u}_1 \, \langle \boldsymbol{v}_1, \boldsymbol{h} \rangle, \tag{4}$$

where $\boldsymbol{v}_1$ serves as the *key*, selecting the relevant input directions, $\boldsymbol{u}_1$ defines the *value* direction injected into the output space, and $\alpha \sigma_1$ controls the magnitude of the update.

To characterize the evolution of the dominant update direction during training, we collect the sequence of $\boldsymbol{u}_1$ vectors across $T$ checkpoints for each module:

$$\mathcal{U}_1 = \{\boldsymbol{u}_1^{(t)}\}_{t=1}^T, \tag{5}$$

which we refer to as the module's *update trajectory*.

Since each $\boldsymbol{u}_1^{(t)}$ resides in a high-dimensional space, we first apply Principal Component Analysis (PCA) to capture the top 50 principal components, retaining the most significant directions of variation. The vectors are then projected onto this 50-dimensional subspace, and t-SNE is subsequently applied to these projections to obtain a two-dimensional, geometry-aware visualization of the trajectory. This procedure provides an interpretable representation of how the Rank-1 update direction evolves over the course of training.

**Details of PLS regression**

For each module, we collect checkpoint-wise pairs, forming the set:

$$\mathcal{D} = \{(\boldsymbol{u}^{(t)}, y^{(t)})\}_{t=1}^T, \tag{6}$$

where $\boldsymbol{u}^{(t)} \in \mathbb{R}^d$ is the Rank-1 left singular ("value") vector extracted at checkpoint $t$, and $y^{(t)} \in \mathbb{R}$ is the corresponding reasoning accuracy. The vectors are stacked row-wise into $\mathcal{U}_1 \in \mathbb{R}^{T \times d}$, and each feature is standardized to zero mean and unit variance, yielding the design matrix $\tilde{\mathcal{U}}_1$.

We then perform Partial Least Squares (PLS) regression with a single latent component. PLS regression can be viewed as Ordinary Least Squares (OLS) applied in a latent low-dimensional space: it first extracts the most predictive direction by maximizing the covariance with the response variable, and then fits the target values on this component using OLS. The resulting score vector is defined as:

$$z_1 = \tilde{\mathcal{U}}_1 \, \boldsymbol{w}_1, \tag{7}$$

where $\boldsymbol{w}_1$ identifies the direction in the standardized value space that is maximally predictive of accuracy. Accuracy is then regressed on this component via:

$$y^{(t)} = \alpha \, z_1^{(t)} + \beta + \varepsilon^{(t)}, \tag{8}$$

with $(\hat{\alpha}, \hat{\beta})$ estimated by OLS, i.e., by minimizing the sum of squared residuals:

$$(\hat{\alpha}, \hat{\beta}) = \arg \min_{\alpha, \beta} \sum_{t=1}^T \left(y^{(t)} - (\alpha z_1^{(t)} + \beta)\right)^2. \tag{9}$$

The coefficient of determination is computed as:

$$R^2 = 1 - \frac{\sum_{t=1}^T \left(y^{(t)} - \hat{y}^{(t)}\right)^2}{\sum_{t=1}^T \left(y^{(t)} - \bar{y}\right)^2}, \quad \hat{y}^{(t)} = \hat{\alpha} z_1^{(t)} + \hat{\beta}. \tag{10}$$

Here, $R^2$ quantifies the strength of the approximately linear coupling between the module's value trajectory and performance variation across checkpoints. In Section 4, AlphaRL perform the same computation but with the scaled vectors $\hat{\boldsymbol{u}}^{(t)} = \alpha^{(t)} \sigma_1^{(t)} \boldsymbol{u}^{(t)}$ instead of the raw vectors $\boldsymbol{u}^{(t)}$.

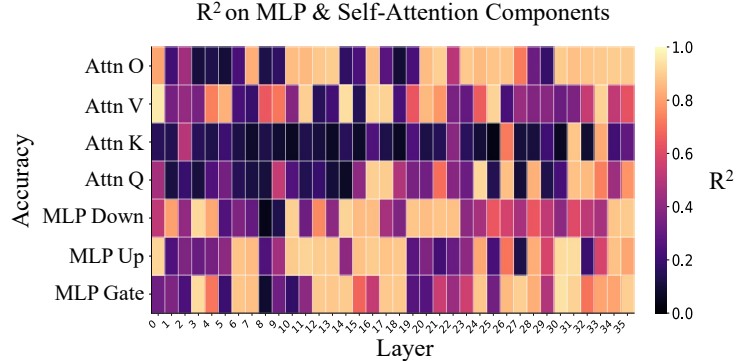

Figure 15: Heatmap of $R^2$ across MLP and self-attention components.

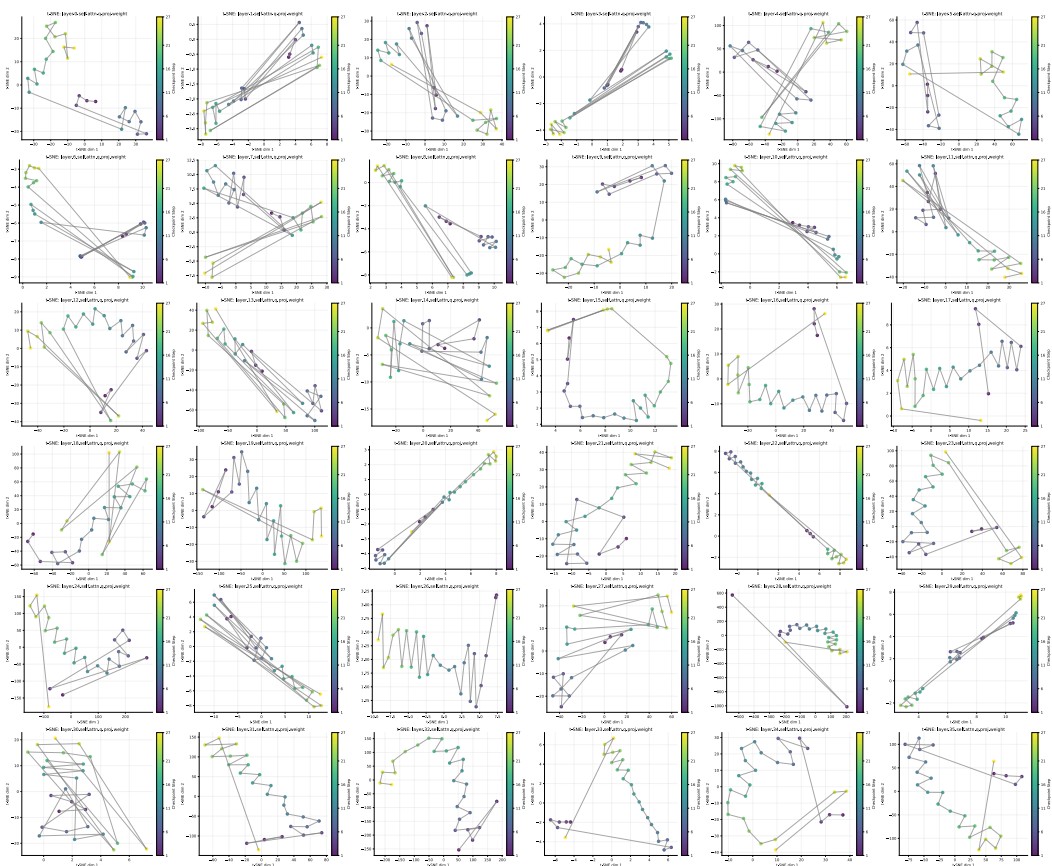

Figure 16: t-SNE visualization of $\mathcal{U}_1$ trajectories under DAPO for Attn Q modules.

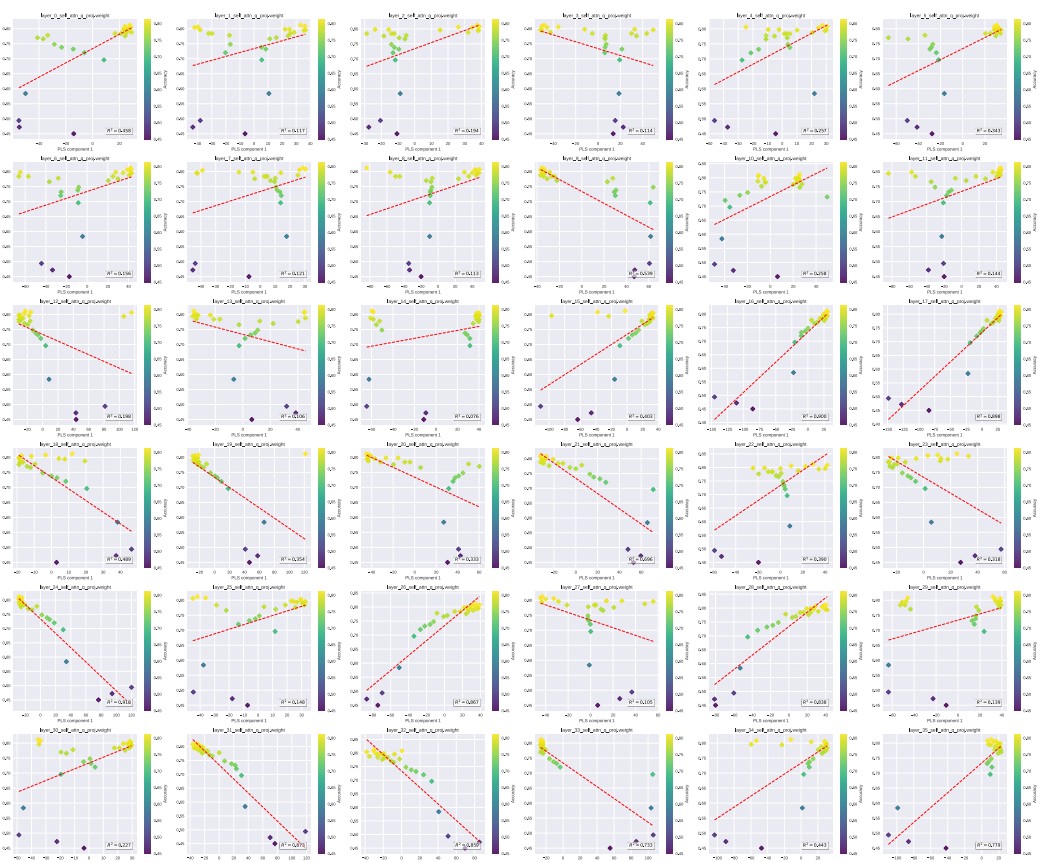

Figure 17: PLS regression visualization of $\mathcal{U}_1$ trajectories under DAPO for Attn Q modules.

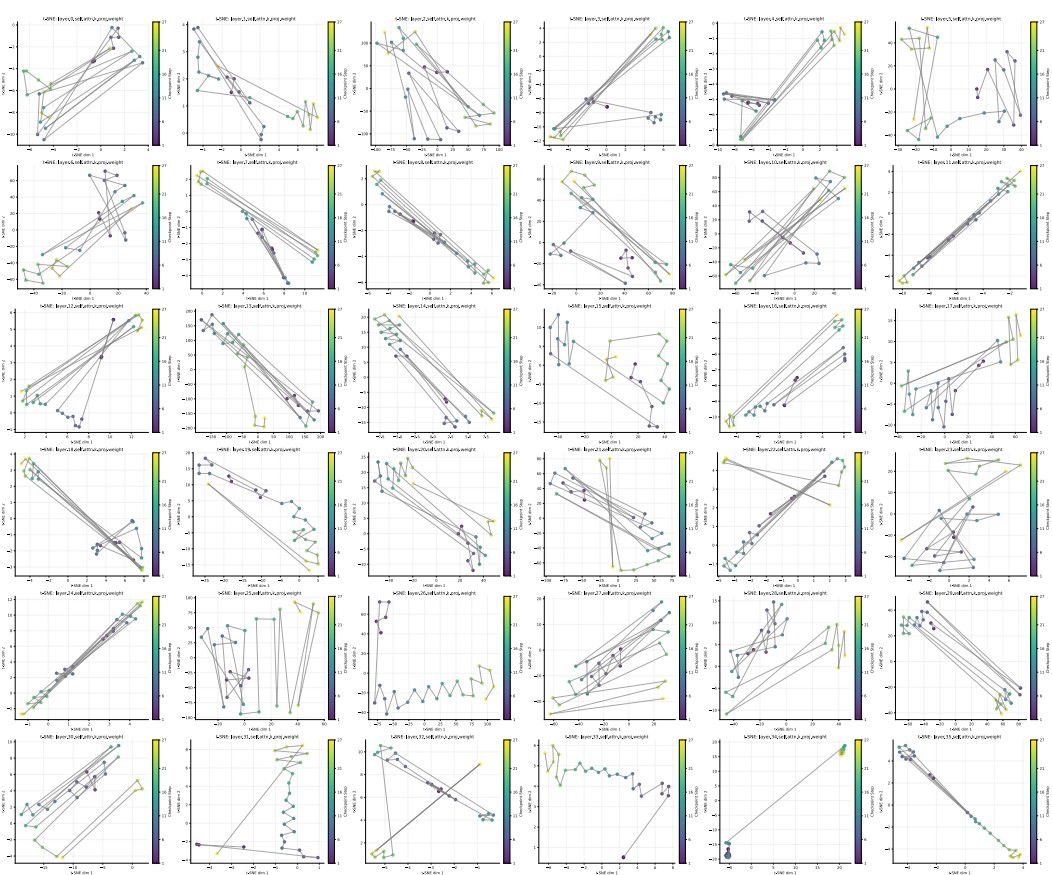

Figure 18: t-SNE visualization of $\mathcal{U}_1$ trajectories under DAPO for Attn K modules.

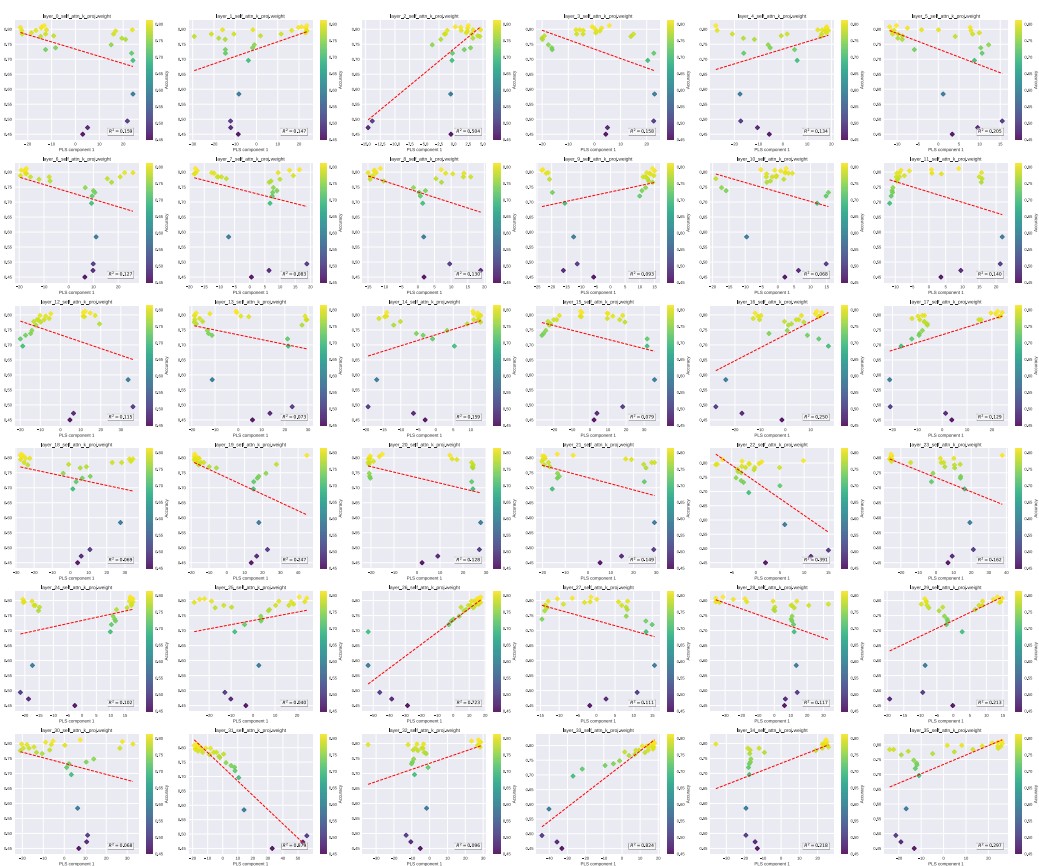

Figure 19: PLS regression visualization of $\mathcal{U}_1$ trajectories under DAPO for Attn K modules.

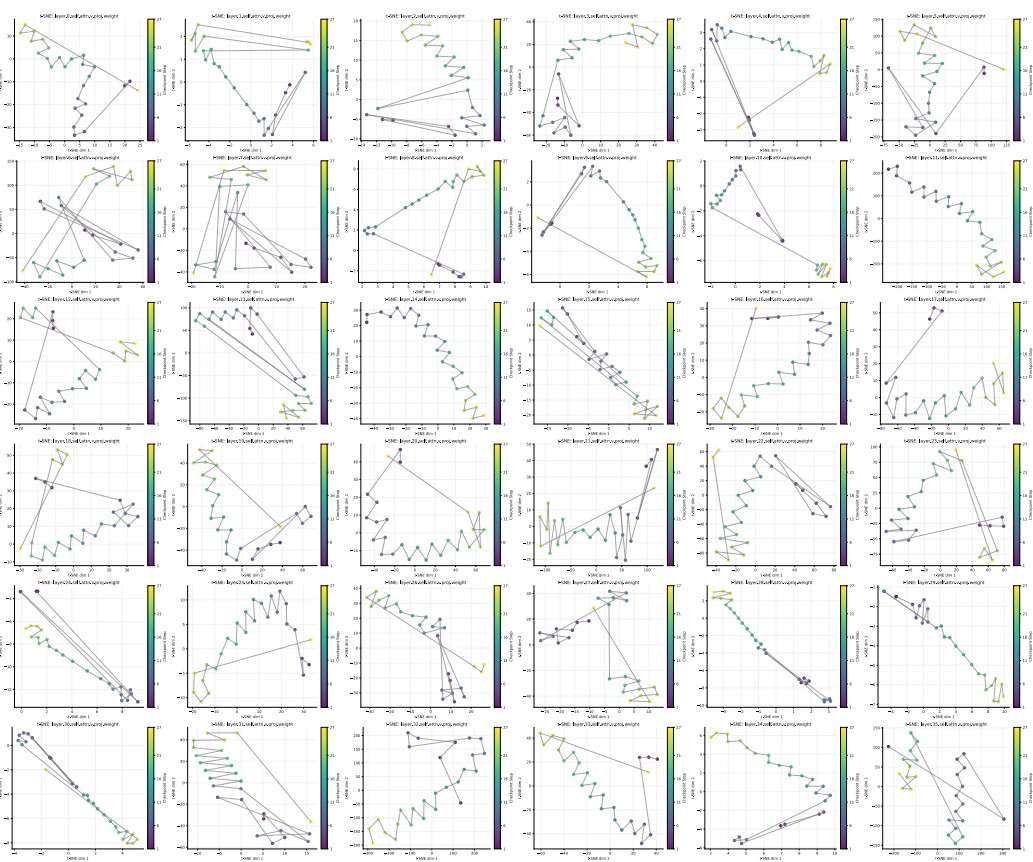

Figure 20: t-SNE visualization of $\mathcal{U}_1$ trajectories under DAPO for Attn V modules.

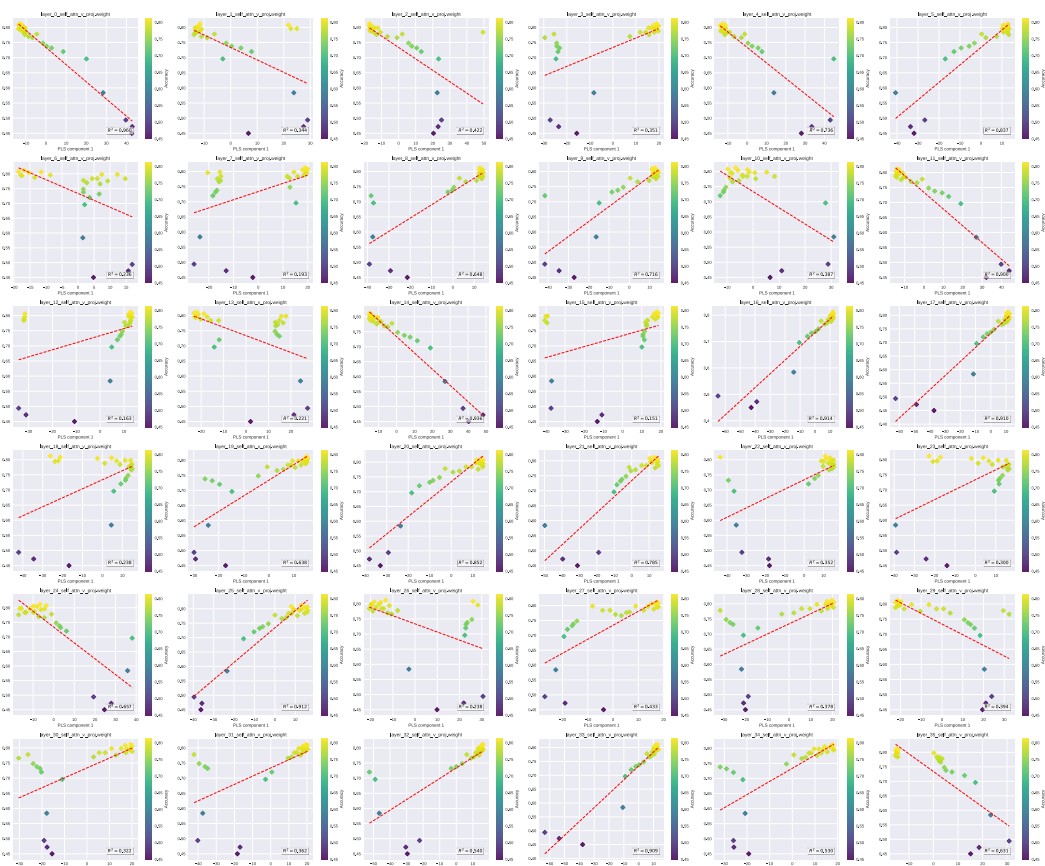

Figure 21: PLS regression visualization of $\mathcal{U}_1$ trajectories under DAPO for Attn V modules.

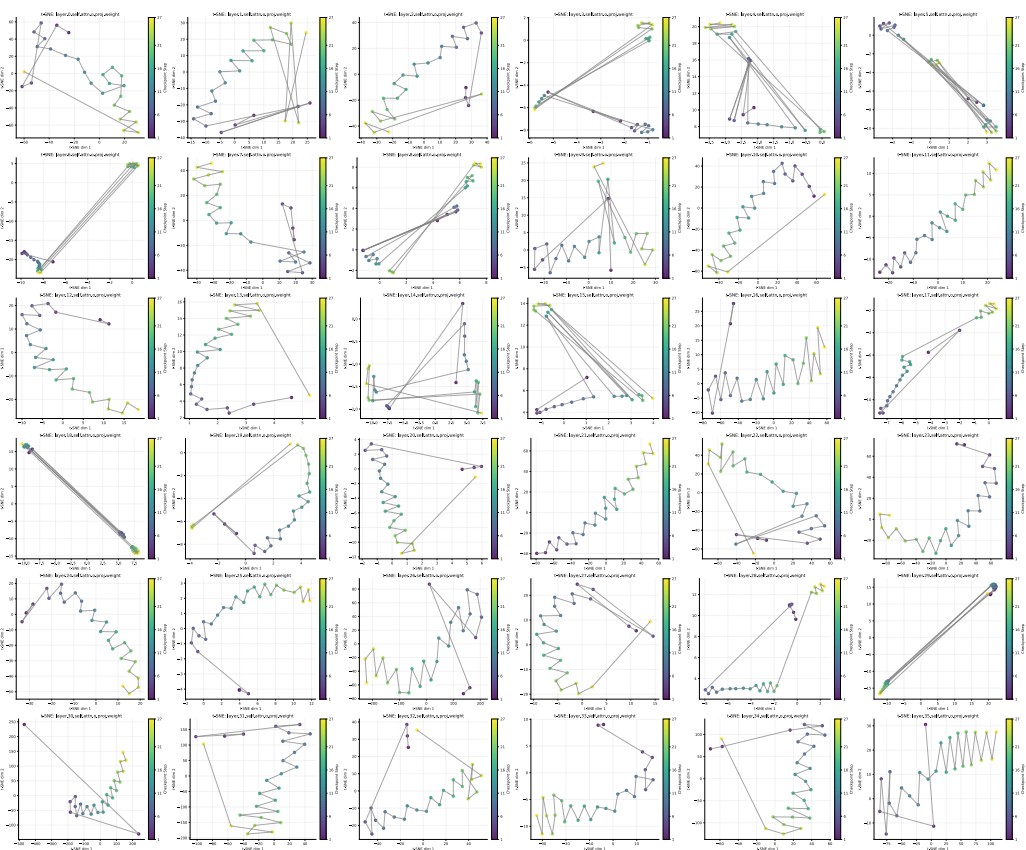

Figure 22: t-SNE visualization of $\mathcal{U}_1$ trajectories under DAPO for Attn O modules.

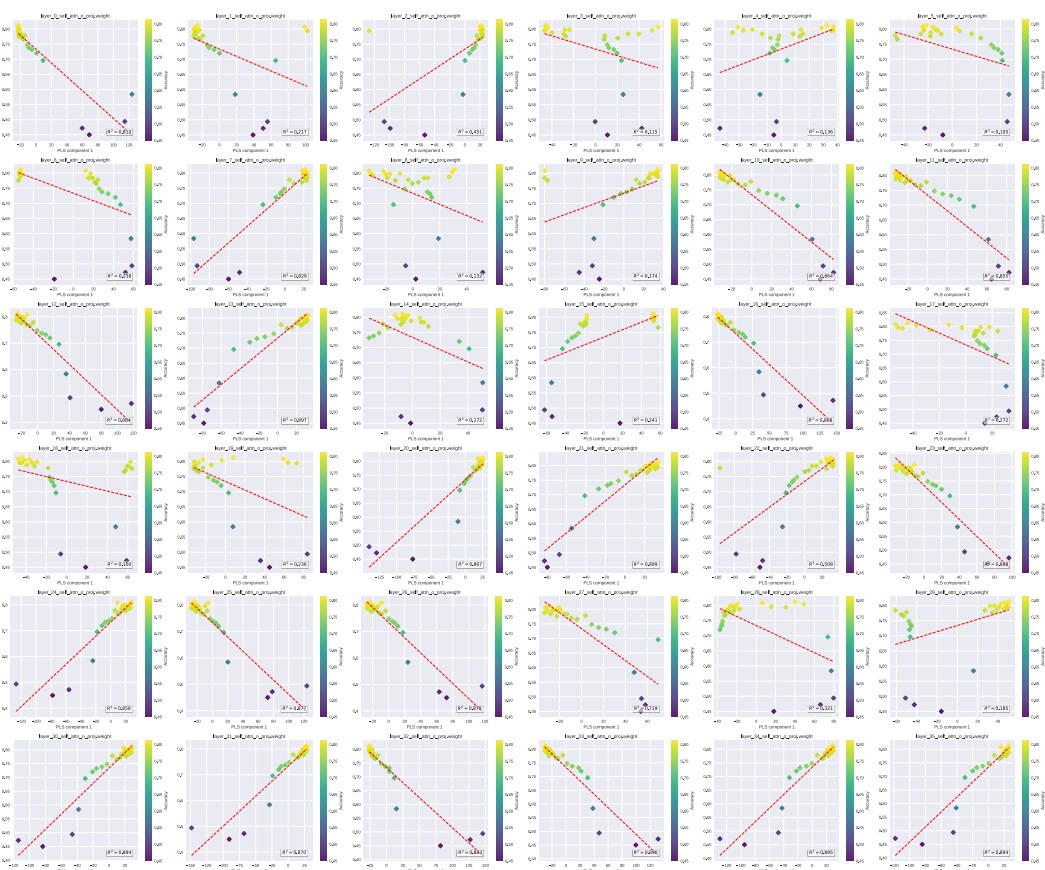

Figure 23: PLS regression visualization of $\mathcal{U}_1$ trajectories under DAPO for Attn O modules.

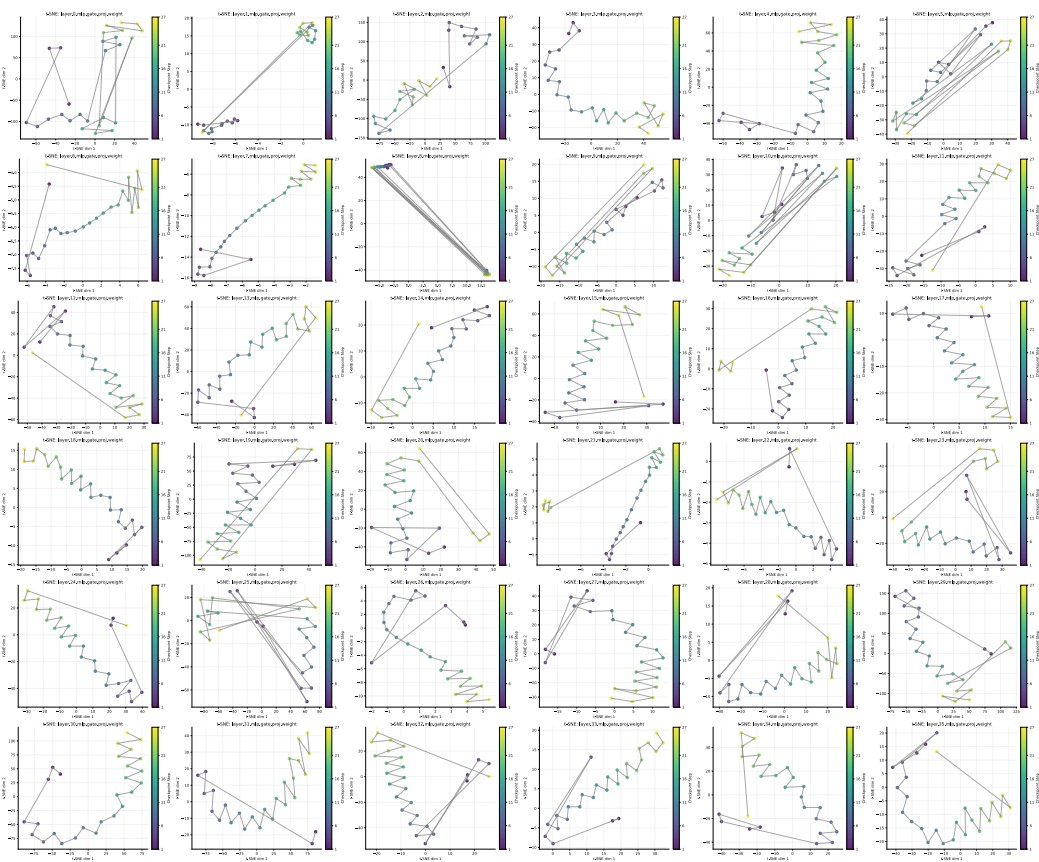

Figure 24: t-SNE visualization of $\mathcal{U}_1$ trajectories under DAPO for MLP GATE modules.

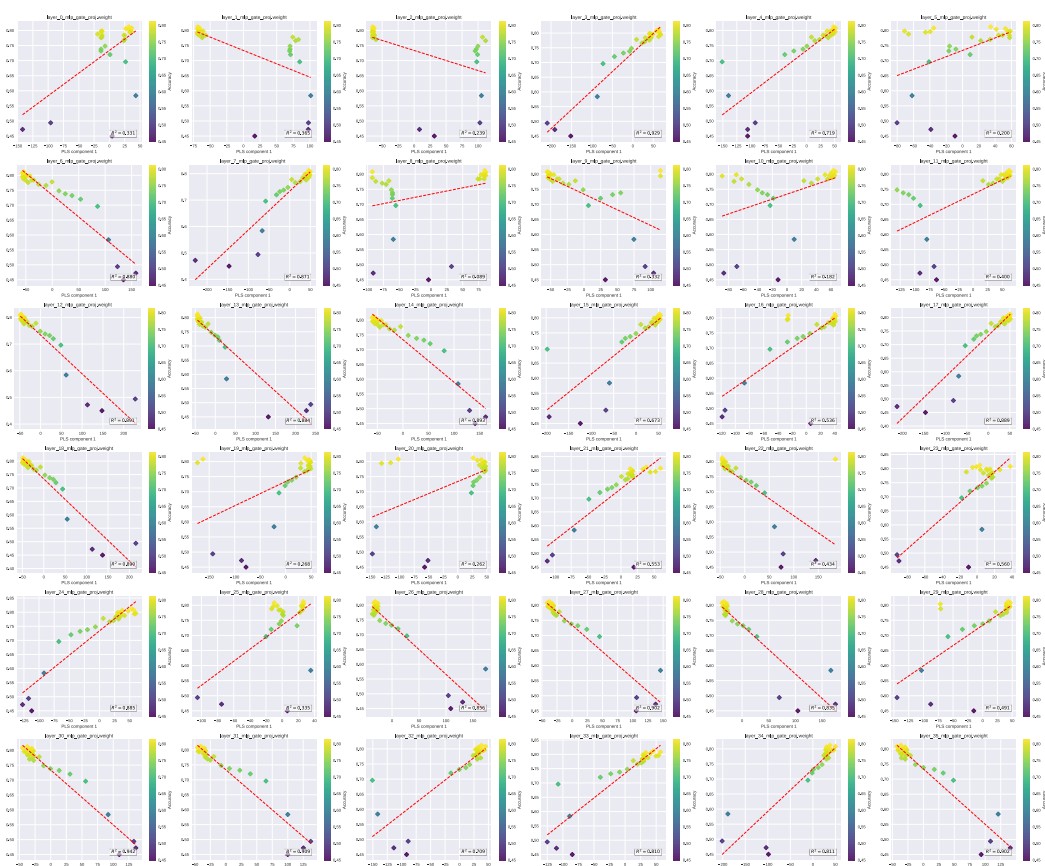

Figure 25: PLS regression visualization of $\mathcal{U}_1$ trajectories under DAPO for MLP GATE modules.

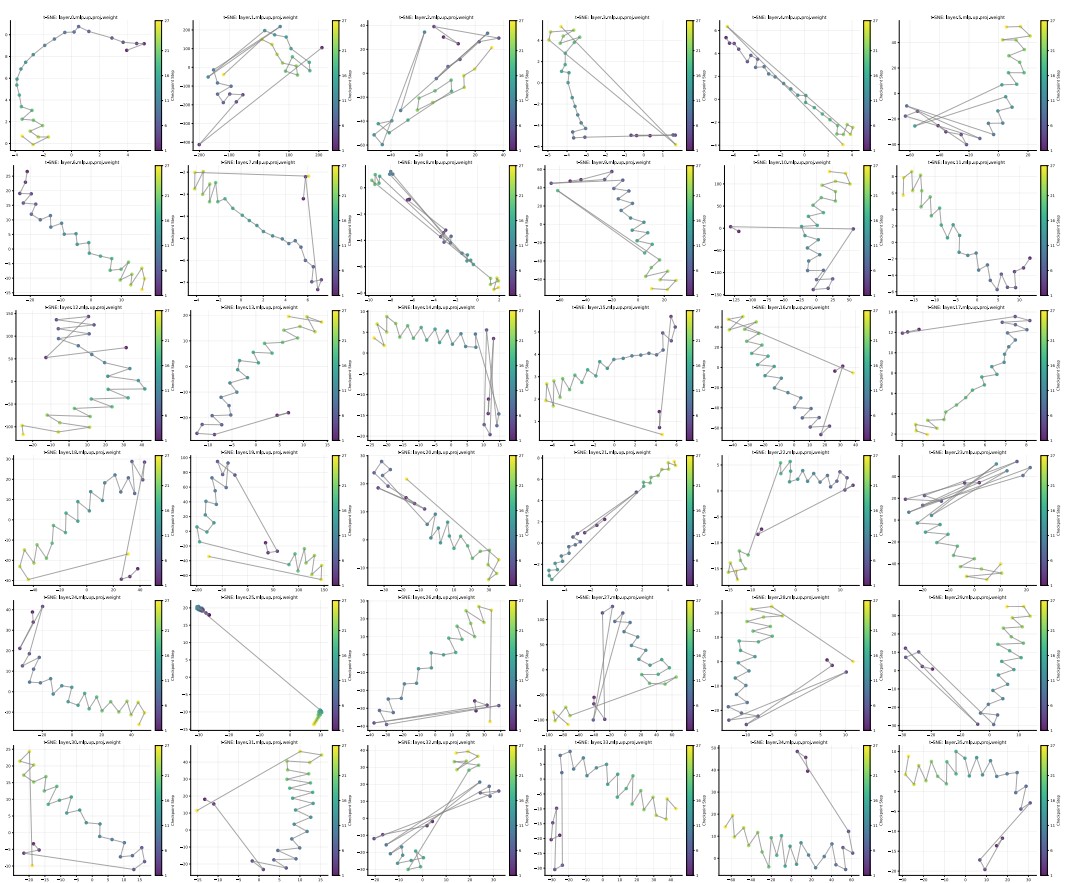

Figure 26: t-SNE visualization of $\mathcal{U}_1$ trajectories under DAPO for MLP UP modules.

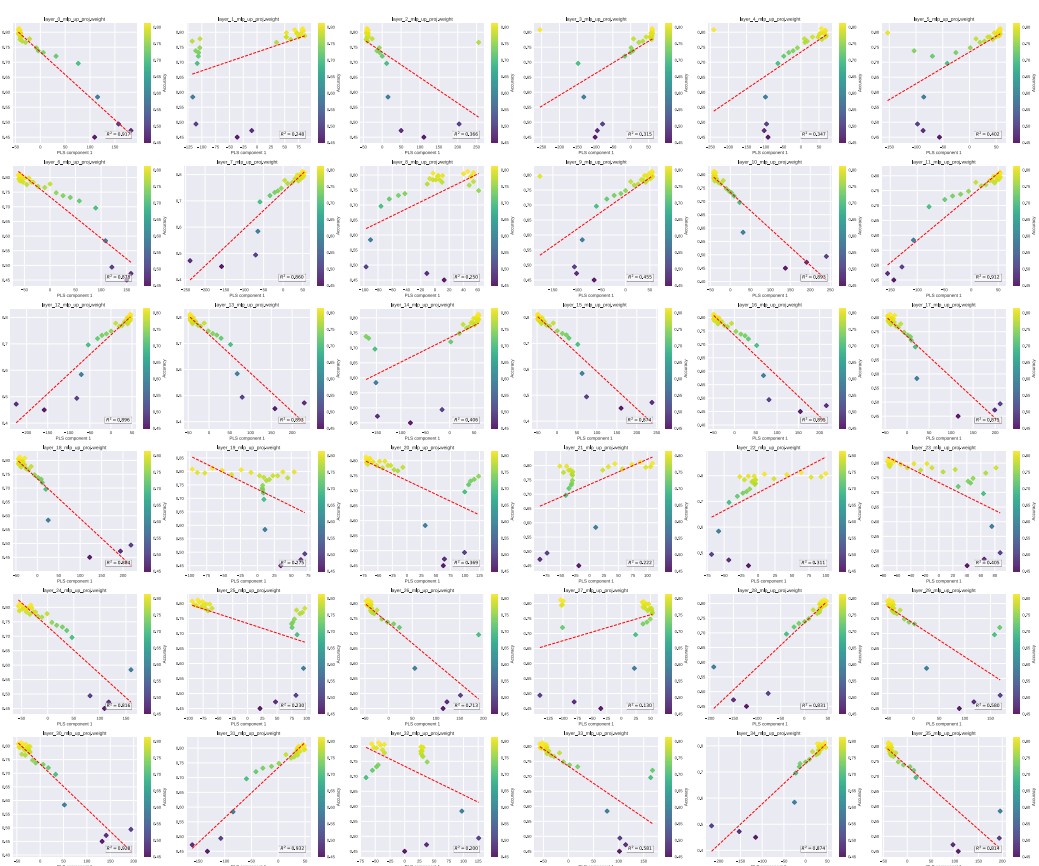

Figure 27: PLS regression visualization of $\mathcal{U}_1$ trajectories under DAPO for MLP UP modules.

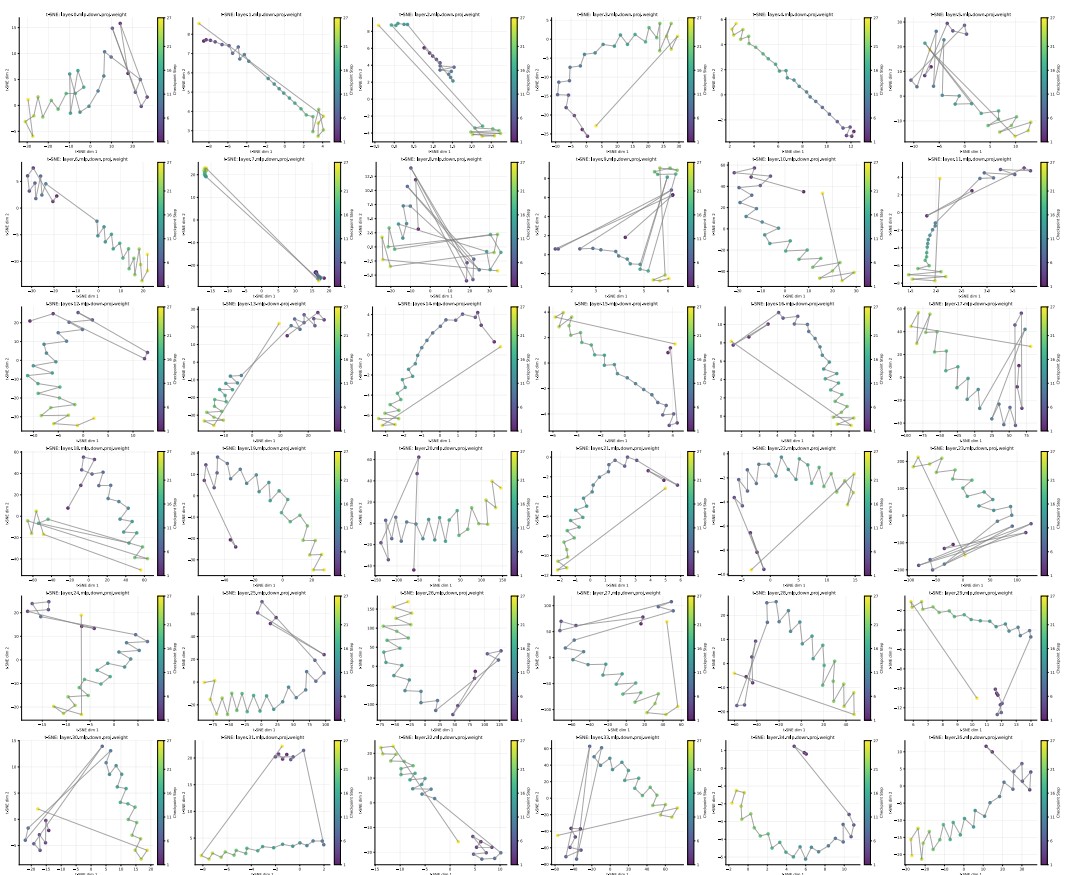

Figure 28: t-SNE visualization of $\mathcal{U}_1$ trajectories under DAPO for MLP DOWN modules.

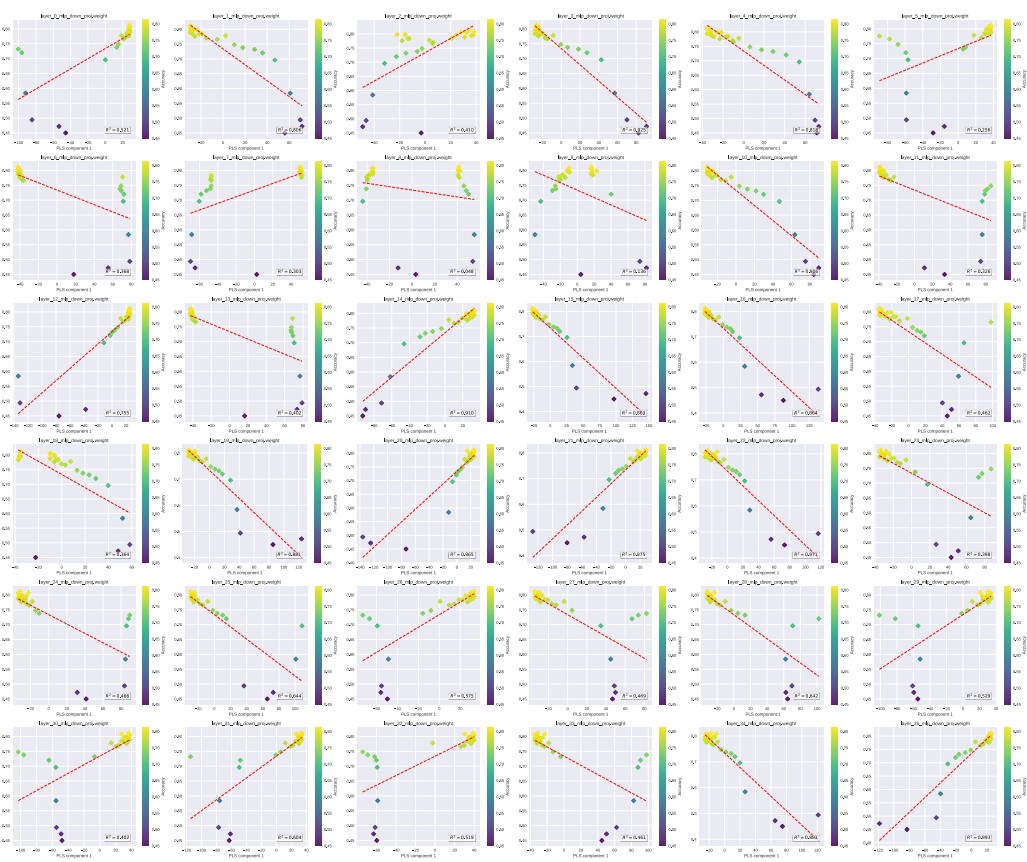

Figure 29: PLS regression visualization of $\mathcal{U}_1$ trajectories under DAPO for MLP DOWN modules.

