# OpenReview forum: "On Predictability of Reinforcement Learning Dynamics for Large Language Models"
_ICLR.cc/2026/Conference — ICLR 2026 Poster_

### Official Review · Reviewer_h22J · 2025-10-30

**Soundness:** 3
**Presentation:** 3
**Contribution:** 4
**Rating:** 4
**Confidence:** 4

**Summary:**

This paper analyzes the parameter dynamics of large language models during reinforcement learning (RL) training. The authors find that RL-induced parameter updates are rank-1 dominant, meaning that most reasoning improvements come from changes along a single dominant direction in parameter space. Moreover, they show that this direction evolves linearly across training checkpoints. Using these observations, the paper proposes AlphaRL, a method that predicts the final update trajectory using only 10-40 % of the training process and extrapolates the remaining steps.

**Strengths:**

- The paper evaluates several RL algorithms (e.g., PPO, GRPO, RLOO, DAPO), providing a comprehensive view of RL dynamics across diverse optimization paradigms.
- The Rank-1 phenomenon is demonstrated on models of different sizes (8B-32B), suggesting that the observed behavior is not limited to a specific scale.
- Results are reported using avg@32 across multiple reasoning benchmarks, ensuring statistical reliability and mitigating sampling variance.
-  The paper clearly motivates the need to understand RL dynamics in LLMs and shows the practical relevance with its AlphaRL method

**Weaknesses:**

- While the paper analyses important RL algorithms, there exist newer and more efficient policy-gradient methods like CISPO[1]. Including such methods would further strengthen the analysis. However, the current coverage is already broad and sufficient for the paper’s main claims.
- Generalisation of Rank-1 and Rank-1 Linear Dynamics
   - (a) in Table 1, the reported accuracies for the fully trained Qwen3-8B model appear notably low. For instance, the paper reports 28.5 % on AIME’24, whereas Qwen3-4B-Instruct achieves roughly 73%[2]. While these numbers are not directly comparable, as Qwen3-4B-Instruct benefits from distillation (which typically yields stronger models) the size of the gap is large. Replicating the entire industrial post-training pipeline of a model like Qwen3 is understandably beyond the paper’s scope, however, this discrepancy raises a concern: are the observed Rank-1 updates and Rank-1 Linear Dynamics an inherent property of RL training or an artifact of under-optimized or unstable training regimes. The same question holds for larger scale models: In Figure 1, DAPO* (a 32B model) only achieves 50% on AIME24 which is also far below Qwen-4B-Instruct.
   - (b) The analysis of the Rank-1 Linear Dynamics in Table 1 is limited to only 8B models, which are known to be less efficient to train under RL[3]. It remains unclear whether the reported phenomena would persist for larger models (e.g., 32B+), where RL tends to be more effective. Adding a table with AlphaRL on a 32B model would strengthen the result.
   - (c) Experiments are only on Qwen models on a single dataset.
- The conclusions of this work seem to contradict some prior findings [4], which report that RL updates are sparse but full-rank. The paper would benefit from a discussion reconciling these differences.

1: MiniMax-M1: Scaling Test-Time Compute Efficiently with Lightning Attention (https://arxiv.org/abs/2506.13585)
2: https://qwen.ai/blog?id=1e3fa5c2d4662af2855586055ad037ed9e555125&from=research.research-list
3: DeepSeek-R1: Incentivizing Reasoning Capability in LLMs via Reinforcement Learning (https://arxiv.org/abs/2501.12948)
4: Reinforcement Learning Finetunes Small Subnetworks in Large Language Models (https://arxiv.org/abs/2505.11711)

**Questions:**

- **Q1:** The paper starts RL training from the Qwen3-Base model, whereas most recent RL works typically apply RL on top of an instruction-tuned model (i.e., following the standard SFT -> RL pipeline).
   - (a) Can the authors elaborate why they started from the base model?
   - (b) Are there experiments or results starting from an instruction-tuned checkpoint? If not, do the authors expect the Rank-1 properties to still hold under the standard SFT -> RL setup?
- **Q2:** For the evaluation on mathematical reasoning benchmarks, was an automated expression evaluation system (e.g., math-verify) or any symbolic equivalence checker used?
- **Q3:** Given the accuracy of Qwen3-8B after training is low compared to vanilla Qwen3-4B-Instruct, could the observed Rank-1 phenomenon & Rank-1 Linear Dynamics be specific to under-optimized or unstable training regimes rather than an inherent property of RL training?
- **Q4:** The experiments are limited to Qwen models trained on the same dataset. How dependent do the authors believe their findings are on the specific base model architecture or training data?
- **Q5:** Could the authors elaborate why they think their results differ from the conclusions in _"Reinforcement Learning Finetunes Small Subnetworks in Large Language Models"_?

---

> ### Author Response · Authors · 2025-11-20
> **# Q1: Can the authors elaborate why they started from the base model? Are there experiments or results starting from an instruction-tuned checkpoint?**
>
> Dear Reviewer h22J,
>
> Thank you for your kind words and positive feedback regarding the novelty, presentation, and effectiveness of our work! Your approval is the great encouragement for us and motivates us to continue advancing our research.
>
> We also appreciate your insightful suggestion to validate the effectiveness of AlphaRL on more models and conduct a deeper comparison with [11].
>
> Below, we meticulously provide responses to each of your comments and outline the modifications made to the manuscript.
>
> -----
>
>
> ### R1:
>
> Thank you for your insight question! The reason this paper uses the Base model is that we have referenced the experimental setups of Shanghai AI Laboratory [1], Qwen Team [2,3], and Seed Lab [4], as they have extensively discussed work on base models in their RL experiments.
>
> We evaluated the predictive performance of **Deepseek-R1-Distill-Qwen-7B** [2] after **off-policy distillation**  and **Llama3-8B-Instruct** [3] **fine-tuned with mathematical instructions**. The results are shown in Table 1:
>
>
> ## Table 1. AlphaRL's performance on Deepseek-R1-Distill-Qwen-7B  and Llama3-8B-Instruct
>
> | **Stage**                  | **AIME24** | **AIME25** | **MATH** | **MINERVA** | **GPQA** | **GSM8K** | **Avg.** |
> |----------------------------|------------|------------|----------|-------------|----------|-----------|----------|
> | **_Deepseek-R1-Distill-Qwen-7B (Distillation + RL)_** | | | | | | | |
> | Training 5%                | 50.83      | 39.17      | 91.25    | 50.00       | 38.89    | 93.00     | 60.69    |
> | Training 5%+AlphaRL        | 51.92      | 40.08      | 92.75    | 51.35       | 40.75    | 93.00     | 61.64    |
> | Training 10%               | 52.50      | 41.75      | 92.75    | 51.50       | 41.17    | 94.25     | 62.32    |
> | Training 10%+AlphaRL       | 56.50      | 44.50      | 94.50    | 54.25       | 43.85    | 94.75     | 64.73    |
> | Training 20%               | 56.50      | 45.75      | 93.75    | 53.75       | 44.25    | 94.75     | 64.79    |
> | Training 20%+AlphaRL       | 60.00      | 52.25      | 95.50    | 57.85       | 48.25    | 95.50     | 68.23    |
> | Training 30%               | 59.50      | 51.50      | 95.20    | 56.22       | 47.25    | 95.50     | 67.52    |
> | Training 30%+AlphaRL       | 64.50      | 57.25      | 96.17    | 60.07       | 51.33    | 96.50     | 70.97    |
> | Training 40%               | 63.25      | 55.25      | 95.00    | 58.45       | 49.86    | 96.50     | 69.74    |
> | Training 40%+AlphaRL       | 67.50      | 61.50      | 97.25    | 62.35       | 53.67    | 97.75     | 73.34    |
> | Training 50%               | 65.75      | 58.85      | 96.50    | 60.65       | 52.27    | 97.25     | 71.87    |
> | Training 50%+AlphaRL       | 68.25      | **62.50**  | 97.25    | **62.58**   | **54.16**| 97.75     | 73.75    |
> | Fully Trained Model        | **68.75**  | 62.00      | **97.50**| 61.74       | 54.16    | **98.25** | **73.65**|
> | **_Llama3-8B-Instruct (SFT + RL)_** | | | | | | | |
> | Training 5%                | 9.58       | 6.67       | 68.75    | 26.44       | 29.75    | 85.25     | 37.74    |
> | Training 5%+AlphaRL        | 12.36      | 9.02       | 71.25    | 28.37       | 31.25    | 86.50     | 39.79    |
> | Training 10%               | 12.63      | 9.16       | 73.00    | 28.37       | 31.75    | 86.25     | 40.19    |
> | Training 10%+AlphaRL       | 15.97      | 13.06      | 77.75    | 31.76       | 34.54    | 89.25     | 43.72    |
> | Training 20%               | 21.11      | 16.25      | 78.00    | 31.07       | 34.75    | 89.50     | 45.11    |
> | Training 20%+AlphaRL       | 26.67      | 23.33      | 84.75    | 34.75       | 37.25    | 92.25     | 49.67    |
> | Training 30%               | 27.78      | 23.33      | 81.50    | 35.25       | 38.25    | 91.50     | 49.26    |
> | Training 30%+AlphaRL       | 34.64      | 31.02      | 85.75    | 38.85       | 42.13    | 93.50     | 53.64    |
> | Training 40%               | 33.33      | 26.67      | 84.25    | 37.50       | 40.75    | 92.25     | 52.45    |
> | Training 40%+AlphaRL       | 44.75      | 34.02      | 87.50    | 41.50       | 44.25    | 94.50     | 57.74    |
> | Training 50%               | 36.11      | 31.11      | 86.25    | 37.50       | 42.25    | 92.50     | 54.28    |
> | Training 50%+AlphaRL       | **45.12**  | 34.75      | **87.75**| 42.25       | **43.75**| 94.25     | 58.00    |
> | Fully Trained Model        | 44.75      | **35.25**  | 87.50    | **42.75**   | 43.75    | **95.00** | **58.17**|
>
> 1. **From experimental perspective:** The performance of AlphaRL on these models is highly consistent with that of the Base model which demonstrate that AlphaRL generalizes well across different compute budgets and training stages.
>
> 2. **From theoretical perspective:** According to Deepseek-R1 [5], the main purposes of SFT is to enhance the model's learning efficiency during the reinforcement process. That means if a model already has some sampling ability, then RL can effectively its effect.

---

> ### Author Response · Authors · 2025-11-20
> **Q2: For the evaluation on mathematical reasoning benchmarks, was an automated expression evaluation system (e.g., math-verify) or any symbolic equivalence checker used?**
>
> ### R2：
>
> Thank you for your question! The evaluation framework we use is LIMO [6] (commonly adopted by Qwen and Deepseek), which employs mathematical expressions to validate the system and checker functions. The checker can be found at line 212 in the ` AlphaRL/eval/reasoning_eval.py` file, making it easier for you to reproduce the results.

---

> ### Author Response · Authors · 2025-11-20
> **Q3: The experiments are limited to Qwen models trained on the same dataset. How dependent do the authors believe their findings are on the specific base model architecture or training data?**
>
> ### R3:
>
> Thank you for your suggestions! We have supplemented the experiments from two perspectives: a broader range of models and training data. The results are as follows:
> 1. During the rebuttal period, we were unable to complete the training of the 32B model. However, we conducted evaluations on the 4B, 9B and 14B models to validate the scalability of AlphaRL. The results are shown in Table 2 and Table 3.
>
>
>  ## Table 2. AlphaRL's performance on Qwen3-14B-Base and GLM-4-9B-0414.
>
> | **Stage**                  | **AIME24** | **AIME25** | **MATH** | **MINERVA** | **GPQA** | **GSM8K** | **Avg.** |
> |----------------------------|------------|------------|----------|-------------|----------|-----------|----------|
> | **_Qwen3-14B-Base_**        |            |            |          |             |          |           |          |
> | Training 5%                | 12.50      | 8.33       | 74.30    | 40.44       | 41.04    | 90.39     | 44.50    |
> | Training 5%+AlphaRL        | 16.11      | 11.11      | 77.75    | 42.27       | 43.25    | 91.25     | 46.96    |
> | Training 10%               | 16.67      | 13.17      | 77.50    | 41.58       | 43.50    | 91.50     | 47.32    |
> | Training 10%+AlphaRL       | 24.72      | 18.05      | 81.25    | 42.37       | 44.76    | 92.75     | 50.65    |
> | Training 20%               | 25.00      | 18.33      | 81.25    | 42.37       | 45.67    | 92.75     | 50.90    |
> | Training 20%+AlphaRL       | 30.28      | 26.11      | 86.25    | 44.65       | 48.76    | 94.75     | 55.13    |
> | Training 30%               | 28.33      | 23.33      | 85.00    | 43.67       | 47.34    | 93.50     | 53.53    |
> | Training 30%+AlphaRL       | 35.69      | 28.89      | 89.50    | 46.50       | 51.73    | 96.75     | 58.17    |
> | Training 40%               | 33.33      | 25.00      | 88.25    | 44.30       | 49.75    | 94.75     | 55.90    |
> | Training 40%+AlphaRL       | 43.47      | 36.25      | 91.25    | 47.75       | 52.59    | 97.75     | 61.51    |
> | Training 50%               | 37.64      | 28.89      | 89.75    | 45.67       | 51.25    | 95.25     | 58.08    |
> | Training 50%+AlphaRL       | 45.00      | 36.80      | **92.25**| **48.33**   | 53.75    | **97.50** | 62.27    |
> | Fully Trained Model        | **45.50**  | **37.63**  | 91.75    | 48.33       | **54.50**| 97.50     | **62.54**|
> | **_GLM-4-9B-0414_**         |            |            |          |             |          |           |          |
> | Training 5%                | 4.17       | 1.67       | 64.20    | 35.29       | 40.40    | 87.34     | 38.84    |
> | Training 5%+AlphaRL        | 5.67       | 3.33       | 66.75    | 37.35       | 42.50    | 88.85     | 40.91    |
> | Training 10%               | 12.50      | 7.50       | 72.25    | 37.25       | 43.50    | 88.50     | 43.58    |
> | Training 10%+AlphaRL       | 16.67      | 11.67      | 79.50    | 40.25       | 53.25    | 90.75     | 48.68    |
> | Training 20%               | 15.00      | 13.33      | 79.50    | 39.75       | 44.50    | 90.25     | 47.06    |
> | Training 20%+AlphaRL       | 23.33      | 20.00      | 83.25    | 43.50       | 57.75    | 92.25     | 55.25    |
> | Training 30%               | 21.52      | 18.06      | 84.20    | 42.33       | 46.36    | 92.33     | 50.13    |
> | Training 30%+AlphaRL       | 28.54      | 23.33      | 88.50    | 48.25       | 53.33    | 95.55     | 56.25    |
> | Training 40%               | 26.67      | 25.50      | 86.50    | 45.50       | 49.35    | 94.00     | 54.58    |
> | Training 40%+AlphaRL       | 33.33      | 31.25      | 91.50    | 50.25       | 54.75    | 96.25     | 59.56    |
> | Training 50%               | 28.33      | 26.67      | 87.25    | 46.75       | 50.20    | 94.25     | 55.58    |
> | Training 50%+AlphaRL       | 34.60      | **34.02**  | 91.50    | 50.87       | 55.33    | **97.00** | 60.55    |
> | Fully Trained Model        | **35.10**  | 33.75      | **91.75**| **51.34**   | **55.67**| 96.25     | **60.64**|

---

> ### Author Response · Authors · 2025-11-20
>
> 2. To validate the predictive capability of AlphaRL on other task distributions, we followed the Self-Play framework used by Liu et al. [7], similar to AlphaZero [8] in Go, with the aim of using a large language model LLM to play two adversarial roles in a self-play setting. We chose this framework for the following reasons:
>
>     1. The reward function is entirely determined by whether the role played wins in the game, which means the rewards are highly variable and random in nature.
>     2. The environment is fundamentally different from mathematical reasoning, as it is fixed to a specific game problem, such as poker or mutual negotiation.
>
>      We used Qwen-4-Base as the base model, training it for approximately 200 checkpoints, saving a checkpoint every 2 epochs, and measuring its win rate against Gemini-2.0-Flash-Lite3. Our experimental results are as follows:
>
>
> ## Table 3. AlphaRL's performance on Kuhn Poker Games.
> | **Stage**           | **WinRate** | **Rank-1 WinRate** | **AlphaRL WinRate** |
> |---------------------|-------------|--------------------|---------------------|
> | **_Qwen3-4B-Base_** |             |                    |                     |
> | **Training 5%**     | 3%          | 2%                 | 4%                  |
> | **Training 10%**    | 5%          | 5%                 | 10%                 |
> | **Training 20%**    | 13%         | 15%                | 28%                 |
> | **Training 30%**    | 27%         | 25%                | 41%                 |
> | **Training 40%**    | 43%         | 44%                | 59%                 |
> | **Training 60%**    | 52%         | 52%                | 64%                 |
> | **Training 80%**    | 61%         | 59%                | -                   |
> | **Training 100%**   | **65%**     | 62%                | -                   |
>
>
> As shown in Table 3, we find that AlphaRL still demonstrates strong predictive ability in this domain of adversarial reinforcement, which suggests that the Rank-1 property and linearity are not only present in the mathematical aspect.
>
> Overall, these experimental results demonstrate the strong versatility of AlphaRL.
>
> Hope that these updates could address your concerns!

---

> ### Author Response · Authors · 2025-11-20
> **Q4: Given the accuracy of Qwen3-8B after training is low compared to vanilla Qwen3-4B-Instruct, could the observed Rank-1 phenomenon & Rank-1 Linear Dynamics be specific to under-optimized or unstable training regimes rather than an inherent property of RL training?**
>
> ### R4：
>
> Thank you again for your question! We have ruled out the concerns you raised based on the following four points:
>
> 1. According to the technical report of Qwen3 [9], the **Qwen3-Base** series represents the checkpoint after the pretraining stage. On the other hand, models like **Qwen3-4B** and **Qwen3-8B** are distilled from **Qwen3-235B-A22B**, rather than being directly RL learned from the Base model. This means that the capabilities of Qwen3-4B come from a 235B model, so it is reasonable that its performance surpasses that of DAPO-32B [4], which was trained solely using DAPO.
>
> 2. As for the teacher model, **Qwen3-235B-A22B**, the process it underwent includes:
>    1. Long Thought Chain Cold Start
>    2. Long Thought Chain Reinforcement Learning
>    3. Thought Pattern Fusion
>    4. General Reinforcement Learning
>
>    which involves two stages of **SFT (Supervised Fine-Tuning)** and **RL (Reinforcement Learning)**.
>
> 3. Our results on AIME24 are in agreement with those presented in the CISPO [10] and DAPO [4] papers for DAPO-32B, and we cited the work of CISPO [10] in the Related Work section of the new version of the PDF.
>
>
>
> 4. The training steps we used for Qwen3-8B-Base and Qwen3-14B-Base are highly consistent with those in [2], and the evaluation results remain highly consistent.
>
> These results out the possibility of incorrect conclusions due to insufficient optimization or instability.
>
> Hope our response could meet your concerns!

---

> ### Author Response · Authors · 2025-11-20
> **Q5: Could the authors elaborate why they think their results differ from the conclusions in "Reinforcement Learning Finetunes Small Subnetworks in Large Language Models"?**
>
> ### R5:
> **We sincerely appreciate your thoughts and suggestions!**  We believe that the conclusions of our paper are **not in conflict with [11], but rather complementary**, for the following reasons:
>
> 1. **From the Research Objective Perspective**:
>
>    1. They [11] focus on the impact of small values in the update matrix on RL performance. Specifically, they show that setting values below a certain threshold to zero does not affect the RL results. As for the rank, they only compute the rank of the update matrix, but they do not investigate how this rank information influences the performance.
>    2. In contrast, we also observe the full rank property (Page 5 Fig 4 and Page 25 Fig 11), but the most important aspect is that **we demonstrate that much of this rank information is redundant**. This, in essence, further proves the sparsity of the RL updates from a different perspective.
>
> 2. **From the Research Task Perspective**: They only focus on the update matrix after full model training. In contrast, we are the first to investigate the dynamic changes in the RL training process, specifically focusing on the matrix changes at each training checkpoint.
>
> Hope that these comparisons can help address your understanding of the paper!

---

> ### Author Response · Authors · 2025-11-20
>
> Once again, we deeply appreciate your thoughtful and encouraging feedback. Your suggestions have not only enhanced the current work but have also inspired us to to keep moving forward and contributing to the community!
>
> Best,
>
> Authors of Paper 10568

---

> ### Author Response · Authors · 2025-11-20
>
> ## Reference
>
> [1] The Entropy Mechanism of Reinforcement Learning for Reasoning Language Models
>
> [2] Beyond the 80/20 Rule: High-Entropy Minority Tokens Drive Effective Reinforcement Learning for LLM Reasoning
>
> [3] Group Sequence Policy Optimization
>
> [4] DAPO: An Open-Source LLM Reinforcement Learning System at Scale
>
> [5] DeepSeek-R1: Incentivizing Reasoning Capability in LLMs via Reinforcement Learning
>
> [6] https://github.com/GAIR-NLP/LIMO
>
> [7] Direct Preference Optimization: Your Language Model is Secretly a Reward Model
>
> [8] SPIRAL: Self-Play on Zero-Sum Games Incentivizes Reasoning via Multi-Agent Multi-Turn Reinforcement Learning
>
> [9] https://qwen.ai/blog?id=qwen3
>
> [10] MiniMax-M1: Scaling Test-Time Compute Efficiently with Lightning Attention
>
> [11] Reinforcement Learning Finetunes Small Subnetworks in Large Language Models

---

> ### Comment · Reviewer_h22J · 2025-11-25
> **Response to Author Rebuttal**
>
> All of my concerns have been addressed. I thank the authors for their thoughtful responses and for conducting additional experiments. I am now convinced that the reported results generalize across model architectures, model sizes and datasets, and that the observed rank-1 phenomenon is indeed a consequence of the RL training procedure. I believe the proposed AlphaRL method represents a simple yet excellent contribution that I expect will attract strong interest from the LLM post-training community at ICLR. I am therefore updating my score to 10.

---

> ### Author Response · Authors · 2025-11-26
> **Expressing Our Sincere Gratitude to the Reviewer h22J**
>
> Dear Reviewer h22J,
>
> Thank you very much for providing constructive insights throughout the review process, which greatly helped us improve the work, as well as for your final recognition. Your feedback means a great deal to us and further strengthens our determination to continue exploring the directions of reinforcement learning and LLM training.
>
> For future research, we believe the following directions merit more systematic investigation:
>
> **1. Evaluating whether AlphaRL achieves better predictive performance when applied to trajectories from LoRA-Adapter RL training.**
> Given that LoRA itself imposes an explicit low-rank structure, we are curious whether the rank-1 direction identified by AlphaRL may become more salient or easier to model under this setting. This could not only improve predictive capability but also deepen our understanding of how different parameterizations influence RL dynamics.
>
> **2. Extending AlphaRL from a post-hoc analysis tool to an integrated component of the training process.**
> A natural extension is to periodically insert AlphaRL updates during training (e.g., applying a rank-1 correction every few steps) to examine whether this can stabilize the optimization trajectory and enable more efficient and robust convergence.
>
> **3. Investigating whether the linear and low-rank structures observed in representation engineering share common mechanisms with low-rank directions in RL.**
> Some studies in representation engineering also reveal pronounced linearity and low-rank structure [1, 2], raising an intriguing question: are these phenomena driven by similar optimization dynamics? Establishing such a connection may point toward a more universal mechanism governing model adaptation.
>
> **4. Exploring whether AlphaRL's properties can help mitigate catastrophic forgetting.**
> Full-parameter updates often lead to catastrophic forgetting across multiple subtasks [3]. Although RL provides partial relief from this issue, tasks still share the same parameter space, leading to representational interference. We are curious whether AlphaRL could improve stability in continual or multi-task learning and reduce cross-task interference.
>
> Once again, we sincerely appreciate your patience, understanding, and support throughout the entire review process!
>
> With best wishes,
>
> Authors of Paper 10568
>
> [1] Representation Engineering: A Top-Down Approach to AI Transparency
>
> [2] Refusal in Language Models Is Mediated by a Single Direction
>
> [3] Gradient Projection Memory for Continual Learning

---

### Official Review · Reviewer_kA6K · 2025-10-31

**Soundness:** 3
**Presentation:** 3
**Contribution:** 3
**Rating:** 6
**Confidence:** 4

**Summary:**

This work identifies two properties of RL-induced parameter updates in LLMs: (1) Rank1 Dominance, where the top singular subspace of the parameter update matrix nearly fully determines reasoning improvements, recovering over 99% of performance gains; and (2) Rank-1 Linear Dynamics, where this dominant subspace evolves linearly throughout training, enabling accurate prediction from early checkpoints. Extensive experiments across 8 LLMs and 7 algorithms validate the generalizability of these properties.

**Strengths:**

1. This paper is clearly written and easy to follow.
2. The findings are interesting and novel for me.

**Weaknesses:**

1. Lack of theoretical justification on the observations. The conclusions are primarily based on large-scale empirical observations, which uncover universal low-rank dynamics in RL training. However, these findings lack rigorous theoretical foundations. In particular, why does RL possess these characteristics while SFT does not?
2. Experiments are mainly on reasoning tasks. Investigating whether the findings hold for reasoning tasks only or not may help figure out the reason why the findings hold.

**Questions:**

1. How does RL methods like GRPO/PPO with learned reward models on alignment tasks perform? Do the observed phenomenon still occur? And what about DPO?

---

> ### Author Response · Authors · 2025-11-20
> **Q1：Lack of theoretical justification on the observations. In particular, why does RL possess these characteristics while SFT does not?**
>
> Thank you for in-depth consideration of our paper! This will motivate us to further refine our claims. In Appendix C, Page 22, we have discussed the reasons behind this phenomenon:
>
> 1. **On-Policy Sampling Strategy**
> 2. **Constraints on the distribution of $\pi_{\theta}$**
>
> To further explore these hypotheses, we have added additional experiments. Specifically:
>
> 1. We have included **On-Policy Distillation [1]**  (a distillation algorithm that incorporates the On-Policy sampling strategy) to validate Reason 1.
> 2. We have added **Direct Preference Optimization (DPO) [2]**, an Off-Policy preference optimization strategy, to validate Reason 2.  The results are shown in Table 7 and 8, and we have added the t-SNE results of the embedding layers for On-Policy DIST and DPO in Page 24.
>
> ### Table1. L2 norm of updates across methods and the fraction of update information captured by the unscaled Rank-1 and Rank-1% Subspaces
>
> | **Method**              | **Off-Policy DIST** | **On-Policy DIST** | **Off-Policy SFT** | **Off-Policy DPO** | **On-Policy RL DAPO** |
> |-------------------------|------------------|-----------------|-----------------|-----------------|--------------------|
> | **On Policy Sampling**     | No            | Yes           | No          | No            | Yes               |
> |**Constraints on the distribution of $\pi_{\theta}$** | No            | Yes           | No          | Yes           | Yes               |
> | Average Update Norm     | 21.24            | 1.46            | 10.75           | 0.79            | 0.01               |
> | Rank-1 fraction         | 12.6%            | 10.45%          | 7.7%            | 11.42%          | 19.3%              |
> | Rank-1% fraction        | 29.3%            | 24.78%          | 27.7%           | 24.13%          | 36.5%              |
>
>
> ## Table 2. Performance under Rank-1 and Rank-k% Subspaces on MATH-500
>
> | **Method**            | **Off-Policy DIST** | **On-Policy DIST** | **Off-Policy SFT** | **Off-Policy DPO** | **On-Policy RL DAPO** |
> |-----------------------|------------------|-----------------|-----------------|-----------------|--------------------|
> | **On Policy Sampling**     | No            | Yes           | No          | No            | Yes               |
> |**Constraints on the distribution of $\pi_{\theta}$** | No            | Yes           | No          | Yes           | Yes               |
> | **Rank-1**            | 1.50%            | 56.40%          | 2.50%           | 62.25%          | 100.00%            |
> | **Rank-1%**           | 2.50%            | 64.50%          | 3.00%           | 71.50%          | 100.00%            |
> | **Rank-10%**          | 3.00%            | 71.50%          | 4.00%           | 74.50%          | 101.00%            |
> | **Rank-20%**          | 3.00%            | 72.50%          | 7.50%           | 79.00%          | 100.00%            |
> | **Rank-30%**          | 3.50%            | 77.00%          | 8.00%           | 83.25%          | 100.00%            |
> | **Rank-50%**          | 3.50%            | 80.25%          | 12.25%          | 88.25%          | 100.00%            |
> | **Rank-70%**          | 16.00%           | 84.00%          | 24.00%          | 88.25%          | 100.00%            |
> | **Rank-90%**          | 23.25%           | 89.25%          | 46.75%          | 93.25%          | 100.00%            |
>
>
> 1. From Table 1, we observe that compared to Off-Policy Distillation and Off-Policy SFT, **On-Policy DIST** successfully reduces the norm size of the required updates by using the on-policy sampling strategy. In Table 2, the low-rank property of Off-Policy Distillation is better than that of Off-Policy SFT and On-Policy DIST.
>
> 2. Similarly, compared to Off-Policy Distillation and Off-Policy SFT, our experiment with **DPO**, though off-policy, reduces the norm size of the required updates by about an order of magnitude through the constraint on the variation between the update strategy and the reference strategy. In Table 2, DPO's low-rank property is also better than Off-Policy SFT and On-Policy DIST.
>
> 3. The above results suggest that both **on-policy sampling** and **Constraints on the distribution of $ \pi_{\theta}$** improve the low-rank property, and their effect is even more pronounced when used together.

---

> ### Author Response · Authors · 2025-11-20
> **Q2：How do RL methods perform on other tasks?**
>
> ### R2:
> Thank you for your thoughtful consideration of our method. To this end, we followed the Self-Play framework used by Liu et al. [3], similar to AlphaZero [4] in Go, with the aim of using a large language model to play two adversarial roles in a self-play setting. We chose this framework for the following reasons:
>
> 1. The reward function is entirely determined by whether the role played wins in the game, which means the rewards are highly variable and random in nature.
> 2. The environment is fundamentally different from mathematical reasoning, as it is fixed to a specific game problem, such as poker or mutual negotiation.
>
> We used Qwen-4-Base as the base model, training it for approximately 200 checkpoints, saving a checkpoint every 2 epochs, and measuring its win rate against Gemini-2.0-Flash-Lite3. Our experimental results are as follows:
>
>
> ## Table 3. AlphaRL's performance on Kuhn Poker Games.
> | **Stage**           | **WinRate** | **Rank-1 WinRate** | **AlphaRL WinRate** |
> |---------------------|-------------|--------------------|---------------------|
> | **_Qwen3-4B-Base_** |             |                    |                     |
> | **Training 5%**     | 3%          | 2%                 | 4%                  |
> | **Training 10%**    | 5%          | 5%                 | 10%                 |
> | **Training 20%**    | 13%         | 15%                | 28%                 |
> | **Training 30%**    | 27%         | 25%                | 41%                 |
> | **Training 40%**    | 43%         | 44%                | 59%                 |
> | **Training 60%**    | 52%         | 52%                | 64%                 |
> | **Training 80%**    | 61%         | 59%                | -                   |
> | **Training 100%**   | **65%**     | 62%                | -                   |
>
>
> As shown in Table 3, we find that AlphaRL still demonstrates strong predictive ability in this domain of adversarial reinforcement, which suggests that the Rank-1 property and linearity are not only present in the mathematical aspect.

---

> ### Author Response · Authors · 2025-11-20
>
> Furthermore, we conducted a deeper investigation into the scope of AlphaRL, the properties of the base model, and the theoretical supporting methods.
> 1. We performed AlphaRL experiments on larger models (GLM4-9B [5] and Qwen 14B) and with more checkpoints. The experimental results are shown in Table 4. The predictive quality steadily improves and reaches its optimal performance around 40–50% of training, but remains robust even at very early stages (e.g., 5–10%).
>
> ## Table 4. AlphaRL's performance on Qwen3-14B-Base and GLM-4-9B-0414.
>
> | **Stage**                  | **AIME24** | **AIME25** | **MATH** | **MINERVA** | **GPQA** | **GSM8K** | **Avg.** |
> |----------------------------|------------|------------|----------|-------------|----------|-----------|----------|
> **_Qwen3-14B-Base_**        |            |            |          |             |          |           |          |
> | Training 5%                | 12.50      | 8.33       | 74.30    | 40.44       | 41.04    | 90.39     | 44.50    |
> | Training 5%+AlphaRL        | 16.11      | 11.11      | 77.75    | 42.27       | 43.25    | 91.25     | 46.96    |
> | Training 10%               | 16.67      | 13.17      | 77.50    | 41.58       | 43.50    | 91.50     | 47.32    |
> | Training 10%+AlphaRL       | 24.72      | 18.05      | 81.25    | 42.37       | 44.76    | 92.75     | 50.65    |
> | Training 20%               | 25.00      | 18.33      | 81.25    | 42.37       | 45.67    | 92.75     | 50.90    |
> | Training 20%+AlphaRL       | 30.28      | 26.11      | 86.25    | 44.65       | 48.76    | 94.75     | 55.13    |
> | Training 30%               | 28.33      | 23.33      | 85.00    | 43.67       | 47.34    | 93.50     | 53.53    |
> | Training 30%+AlphaRL       | 35.69      | 28.89      | 89.50    | 46.50       | 51.73    | 96.75     | 58.17    |
> | Training 40%               | 33.33      | 25.00      | 88.25    | 44.30       | 49.75    | 94.75     | 55.90    |
> | Training 40%+AlphaRL       | 43.47      | 36.25      | 91.25    | 47.75       | 52.59    | 97.75     | 61.51    |
> | Training 50%               | 37.64      | 28.89      | 89.75    | 45.67       | 51.25    | 95.25     | 58.08    |
> | Training 50%+AlphaRL       | 45.00      | 36.80      | **92.25**| **48.33**   | 53.75    | **97.50** | 62.27    |
> | Fully Trained Model        | **45.50**  | **37.63**  | 91.75    | 48.33       | **54.50**| 97.50     | **62.54**|
> | **_GLM-4-9B-0414_**         |            |            |          |             |          |           |          |
> | Training 5%                | 4.17       | 1.67       | 64.20    | 35.29       | 40.40    | 87.34     | 38.84    |
> | Training 5%+AlphaRL        | 5.67       | 3.33       | 66.75    | 37.35       | 42.50    | 88.85     | 40.91    |
> | Training 10%               | 12.50      | 7.50       | 72.25    | 37.25       | 43.50    | 88.50     | 43.58    |
> | Training 10%+AlphaRL       | 16.67      | 11.67      | 79.50    | 40.25       | 53.25    | 90.75     | 48.68    |
> | Training 20%               | 15.00      | 13.33      | 79.50    | 39.75       | 44.50    | 90.25     | 47.06    |
> | Training 20%+AlphaRL       | 23.33      | 20.00      | 83.25    | 43.50       | 57.75    | 92.25     | 55.25    |
> | Training 30%               | 21.52      | 18.06      | 84.20    | 42.33       | 46.36    | 92.33     | 50.13    |
> | Training 30%+AlphaRL       | 28.54      | 23.33      | 88.50    | 48.25       | 53.33    | 95.55     | 56.25    |
> | Training 40%               | 26.67      | 25.50      | 86.50    | 45.50       | 49.35    | 94.00     | 54.58    |
> | Training 40%+AlphaRL       | 33.33      | 31.25      | 91.50    | 50.25       | 54.75    | 96.25     | 59.56    |
> | Training 50%               | 28.33      | 26.67      | 87.25    | 46.75       | 50.20    | 94.25     | 55.58    |
> | Training 50%+AlphaRL       | 34.60      | **34.02**  | 91.50    | 50.87       | 55.33    | **97.00** | 60.55    |
> | Fully Trained Model        | **35.10**  | 33.75      | **91.75**| **51.34**   | **55.67**| 96.25     | **60.64**|

---

> ### Author Response · Authors · 2025-11-20
>
> 2. We investigated whether AlphaRL is related to the properties of the base model. To do this, we performed reinforcement learning on distilled models (Deepseek-R1-Distill-Qwen-7B [6]) and SFT models (Llama3-8B-Instruct [7]) to validate the effect of AlphaRL. The results are shown in Table 5, which indicates that AlphaRL can be widely applied to different base models.
>
> ## Table 5. AlphaRL's performance on Deepseek-R1-Distill-Qwen-7B  and Llama3-8B-Instruct
>
> | **Stage**                  | **AIME24** | **AIME25** | **MATH** | **MINERVA** | **GPQA** | **GSM8K** | **Avg.** |
> |----------------------------|------------|------------|----------|-------------|----------|-----------|----------|
> | **_Deepseek-R1-Distill-Qwen-7B (After Distillation)_** | | | | | | | |
> | Training 5%                | 50.83      | 39.17      | 91.25    | 50.00       | 38.89    | 93.00     | 60.69    |
> | Training 5%+AlphaRL        | 51.92      | 40.08      | 92.75    | 51.35       | 40.75    | 93.00     | 61.64    |
> | Training 10%               | 52.50      | 41.75      | 92.75    | 51.50       | 41.17    | 94.25     | 62.32    |
> | Training 10%+AlphaRL       | 56.50      | 44.50      | 94.50    | 54.25       | 43.85    | 94.75     | 64.73    |
> | Training 20%               | 56.50      | 45.75      | 93.75    | 53.75       | 44.25    | 94.75     | 64.79    |
> | Training 20%+AlphaRL       | 60.00      | 52.25      | 95.50    | 57.85       | 48.25    | 95.50     | 68.23    |
> | Training 30%               | 59.50      | 51.50      | 95.20    | 56.22       | 47.25    | 95.50     | 67.52    |
> | Training 30%+AlphaRL       | 64.50      | 57.25      | 96.17    | 60.07       | 51.33    | 96.50     | 70.97    |
> | Training 40%               | 63.25      | 55.25      | 95.00    | 58.45       | 49.86    | 96.50     | 69.74    |
> | Training 40%+AlphaRL       | 67.50      | 61.50      | 97.25    | 62.35       | 53.67    | 97.75     | 73.34    |
> | Training 50%               | 65.75      | 58.85      | 96.50    | 60.65       | 52.27    | 97.25     | 71.87    |
> | Training 50%+AlphaRL       | 68.25      | **62.50**  | 97.25    | **62.58**   | **54.16**| 97.75     | 73.75    |
> | Fully Trained Model        | **68.75**  | 62.00      | **97.50**| 61.74       | 54.16    | **98.25** | **73.65**|
> | **_Llama3-8B-Instruct (After SFT)_** | | | | | | | |
> | Training 5%                | 9.58       | 6.67       | 68.75    | 26.44       | 29.75    | 85.25     | 37.74    |
> | Training 5%+AlphaRL        | 12.36      | 9.02       | 71.25    | 28.37       | 31.25    | 86.50     | 39.79    |
> | Training 10%               | 12.63      | 9.16       | 73.00    | 28.37       | 31.75    | 86.25     | 40.19    |
> | Training 10%+AlphaRL       | 15.97      | 13.06      | 77.75    | 31.76       | 34.54    | 89.25     | 43.72    |
> | Training 20%               | 21.11      | 16.25      | 78.00    | 31.07       | 34.75    | 89.50     | 45.11    |
> | Training 20%+AlphaRL       | 26.67      | 23.33      | 84.75    | 34.75       | 37.25    | 92.25     | 49.67    |
> | Training 30%               | 27.78      | 23.33      | 81.50    | 35.25       | 38.25    | 91.50     | 49.26    |
> | Training 30%+AlphaRL       | 34.64      | 31.02      | 85.75    | 38.85       | 42.13    | 93.50     | 53.64    |
> | Training 40%               | 33.33      | 26.67      | 84.25    | 37.50       | 40.75    | 92.25     | 52.45    |
> | Training 40%+AlphaRL       | 44.75      | 34.02      | 87.50    | 41.50       | 44.25    | 94.50     | 57.74    |
> | Training 50%               | 36.11      | 31.11      | 86.25    | 37.50       | 42.25    | 92.50     | 54.28    |
> | Training 50%+AlphaRL       | **45.12**  | 34.75      | **87.75**| 42.25       | **43.75**| 94.25     | 58.00    |
> | Fully Trained Model        | 44.75      | **35.25**  | 87.50    | **42.75**   | 43.75    | **95.00** | **58.17**|

---

> > ### Author Response · Authors · 2025-11-20
> > **Q3 What phenomena were observed with DPO, and how does it differ from GRPO?**
> >
> > ### R3:
> >
> > We discussed the differences between DPO and other methods in Q1. Overall, DPO achieves a better low-rank phenomenon by imposing **Constraints on the distribution of $ \pi_{\theta} $**. However, due to the use of off-policy training data, DPO does not improve sampling efficiency like on-policy RL (GRPO) [8]. Instead, it learns additional new knowledge, which leads to the need for some extra rank information.

---

> > > ### Author Response · Authors · 2025-11-20
> > >
> > > Finally,thank you once again for your valuable feedback and suggestions for improving the paper! We sincerely appreciate the time and effort you have put into reviewing our work. **If you have any further questions or concerns, please feel free to let me know. It is our honor to address any issues and provide any additional clarifications you may need.**
> > >
> > >
> > > ## Reference
> > >
> > > [1] https://huggingface.co/gbcfchc/Qwen3-8B-Base-Open-Thoughts-On-Policy-Distillation
> > >
> > > [2] Direct Preference Optimization: Your Language Model is Secretly a Reward Model
> > >
> > > [3] SPIRAL: Self-Play on Zero-Sum Games Incentivizes Reasoning via Multi-Agent Multi-Turn Reinforcement Learning
> > >
> > > [4] Mastering Chess and Shogi by Self-Play with a General Reinforcement Learning Algorithm
> > >
> > > [5] https://huggingface.co/zai-org/GLM-4-9B-0414
> > >
> > > [6] https://huggingface.co/deepseek-ai/DeepSeek-R1-Distill-Qwen-7B
> > >
> > > [7] https://huggingface.co/meta-llama/Llama-3.1-8B-Instruct
> > >
> > > [8] Does Reinforcement Learning Really Incentivize Reasoning
> > > Capacity in LLMs Beyond the Base Model?

---

### Official Review · Reviewer_E5rk · 2025-10-31

**Soundness:** 3
**Presentation:** 3
**Contribution:** 3
**Rating:** 6
**Confidence:** 3

**Summary:**

The paper studies parameter-update dynamics during the RL procedure of LLMs. The authors identify two general properties when inspecting the parameter difference between the post-trained model and the base model: 1) the top singular subspace of the parameter difference almost fully characterizes the "essence" of reasoning gains; 2) the dominant subspace evolves approximately linear during RL, which enables simple prediction rules given early checkpoints. Leveraging these properties, the authors also propose AlphaRL, which extrapolates the rank-1 trajectory and brings significant training speedup. Experiments across models from 7B to 32B and various RL algorithms validate the effectiveness of the method.

**Strengths:**

1. The identified phenomenon is indeed new (at least to me) and striking and further leaves room for further exploration.
2. The work includes results spanning multiple base models and RL algorithms, with controls showing the phenomena are specific to RL rather than SFT/distillation. It also demonstrates near-linear evolution of rank-1 subspaces links training dynamics to performance, offering a compact lens to monitor or forecast progress.
3. The work goes beyond observation to deliver a practical speedup recipe (up to 2.5x, >96% retention), potentially saving significant compute during RL runs.

**Weaknesses:**

1. The paper itself notes a lack of rigorous theory explaining why RL updates should be rank-1-dominant and linearly evolving. The true underlying mechanisms remain speculative.
2. Most of the evaluated tasks are math-based. It is unclear if the claimed properties generalized across different domains, such as coding, long-context planning, and tool use. The same question holds for completely different model families and datasets used during RL.
3. If I understand correctly, AlphaRL assumes (and hence requires) stable linear trends in early checkpoints and relies on a number of early checkpoints to work. In practice, effectiveness is limited by the design and stability of the RL algorithm, as the paper acknowledges. It remains uncertain whether the proposed approach encounters failure modes.
4. Computing per-module SVDs of the parameter difference across all layers for large models can be memory-/I/O-heavy; the paper doesn't quantify overheads or detail engineering tricks needed for 10B–30B+ models (or even larger ones). I assume this should be doable for small models but am not sure for larger ones.

**Questions:**

1. Is SVD performed per parameter block or per layer (or in a different way)?
2. How does AlphaRL behave under changing reward models, curricula, or exploration regimes that violate linearity (e.g., entropy collapse or under non-verifiable reward)?
3. Is there a criterion to decide when rank-1 is insufficient and a small rank-k should be used?
4. The paper reports that SFT/distillation lack both properties. Can we attribute this to a specific ingredient in RL (on-policy sampling, verifiable rewards, credit assignment) that leads to the low-rank/linearity behavior?

---

> ### Author Response · Authors · 2025-11-20
> **Q1: Is SVD performed per parameter block or per layer (or in a different way)?**
>
> ### R1:
>
> Thank you for your interest in our method. We perform SVD decomposition on any update matrix, defined as the difference between the matrix before and after the update ($\Delta W$).
>
> For example, in the case of Qwen3-8B-Base, which has 36 layers, each layer's attention block contains 4 matrices (attn_q, attn_k, attn_v, attn_o), and the MLP block contains 3 matrices (mlp_proj, mlp_up, mlp_down). This means we perform 36 * (4 + 3) SVD decompositions each model.
>
> The detailed implementation code is provided in the Supplementary Material AlphaRL/eval/svd.py. Additionally, the code for Rank-1 Update can be found in Supplementary Material AlphaRL/eval/upd_rank.py.

---

> ### Author Response · Authors · 2025-11-20
> **Q2: How does AlphaRL behave under changing reward models, curricula, or exploration regimes that violate linearity?**
>
> ### R2:
>
> Thank you for your thoughtful consideration of our method. To this end, we followed the Self-Play framework used by Liu et al. [1], similar to AlphaZero [2] in Go, with the aim of using a large language model to play two adversarial roles in a self-play setting. We chose this framework for the following reasons:
>
> 1. The reward function is entirely determined by whether the role played wins in the game, which means the rewards are highly variable and random in nature.
> 2. The environment is fundamentally different from mathematical reasoning, as it is fixed to a specific game problem, such as poker or mutual negotiation.
>
> We used Qwen-4-Base as the base model, training it for approximately 200 checkpoints, saving a checkpoint every 2 epochs, and measuring its win rate against Gemini-2.0-Flash-Lite3. Our experimental results are as follows:
>
>
> ## Table 1. AlphaRL's performance on Kuhn Poker Games.
> | **Stage**           | **WinRate** | **Rank-1 WinRate** | **AlphaRL WinRate** |
> |---------------------|-------------|--------------------|---------------------|
> | **_Qwen3-4B-Base_** |             |                    |                     |
> | **Training 5%**     | 3%          | 2%                 | 4%                  |
> | **Training 10%**    | 5%          | 5%                 | 10%                 |
> | **Training 20%**    | 13%         | 15%                | 28%                 |
> | **Training 30%**    | 27%         | 25%                | 41%                 |
> | **Training 40%**    | 43%         | 44%                | 59%                 |
> | **Training 60%**    | 52%         | 52%                | 64%                 |
> | **Training 80%**    | 61%         | 59%                | -                   |
> | **Training 100%**   | **65%**     | 62%                | -                   |
>
>
>
> As shown in Table 1, we find that AlphaRL still demonstrates strong predictive ability in this domain of adversarial reinforcement, which suggests that the Rank-1 property and linearity are not only present in the mathematical aspect.

---

> ### Author Response · Authors · 2025-11-20
> **Q3: AlphaRL may be limited by the stability of the learning method. It is currently uncertain whether the proposed method will encounter failure modes.**
>
> ### R3:
>
> Building upon Q2, to explore the adaptability and generalization of AlphaRL, we conducted a deeper investigation into the scope of AlphaRL, the properties of the base model, and the theoretical supporting methods.
>
> 1. We performed AlphaRL experiments on larger models (GLM4-9B [3] and Qwen 14B) and with more checkpoints. The experimental results are shown in **Table 2**. The predictive quality steadily improves and reaches its optimal performance around 40–50% of training, but remains robust even at very early stages (e.g., 5–10%).
>
> ## Table 2. AlphaRL's performance on Qwen3-8B-Base, Qwen3-14B-Base and GLM-4-9B-0414.
>
> | **Stage**                  | **AIME24** | **AIME25** | **MATH** | **MINERVA** | **GPQA** | **GSM8K** | **Avg.** |
> |----------------------------|------------|------------|----------|-------------|----------|-----------|----------|
> | **_Qwen3-8B-Base_**         |            |            |          |             |          |           |          |
> | Training 5%                | 7.50       | 5.50       | 62.25    | 27.25       | 31.50    | 78.47     | 35.45    |
> | Training 5%+AlphaRL        | 10.00      | 6.67       | 69.75    | 31.15       | 34.75    | 81.65     | 38.91    |
> | Training 10%               | 12.50      | 7.50       | 70.25    | 32.07       | 36.66    | 84.30     | 40.55    |
> | Training 10%+AlphaRL       | 15.00      | 11.67      | 76.45    | 40.46       | 41.54    | 93.75     | 46.47    |
> | Training 20%               | 13.17      | 7.50       | 72.25    | 35.07       | 38.50    | 87.54     | 42.34    |
> | Training 20%+AlphaRL       | 16.67      | 12.50      | 75.00    | 41.25       | 42.85    | 94.35     | 47.10    |
> | Training 30%               | 14.45      | 11.67      | 74.25    | 36.37       | 39.42    | 91.25     | 44.56    |
> | Training 30%+AlphaRL       | 23.33      | 20.00      | 78.75    | 41.85       | 44.13    | 93.20     | 50.21    |
> | Training 40%               | 15.80      | 11.67      | 77.60    | 37.07       | 41.67    | 93.20     | 46.30    |
> | Training 40%+AlphaRL       | 28.33      | 23.75      | 80.50    | 43.27       | 49.25    | 94.75     | 53.31    |
> | Training 50%               | 23.33      | 20.00      | 78.50    | 39.87       | 44.33    | 93.75     | 49.40    |
> | Training 50%+AlphaRL       | 27.67      | 23.33      | **82.00**| 42.85       | 47.75    | **94.50** | 53.01    |
> | Fully Trained Model        | **28.54**  | **24.17**  | 80.95    | **44.02**   | **48.23**| 94.35     | **53.38**|
> | **_Qwen3-14B-Base_**        |            |            |          |             |          |           |          |
> | Training 5%                | 12.50      | 8.33       | 74.30    | 40.44       | 41.04    | 90.39     | 44.50    |
> | Training 5%+AlphaRL        | 16.11      | 11.11      | 77.75    | 42.27       | 43.25    | 91.25     | 46.96    |
> | Training 10%               | 16.67      | 13.17      | 77.50    | 41.58       | 43.50    | 91.50     | 47.32    |
> | Training 10%+AlphaRL       | 24.72      | 18.05      | 81.25    | 42.37       | 44.76    | 92.75     | 50.65    |
> | Training 20%               | 25.00      | 18.33      | 81.25    | 42.37       | 45.67    | 92.75     | 50.90    |
> | Training 20%+AlphaRL       | 30.28      | 26.11      | 86.25    | 44.65       | 48.76    | 94.75     | 55.13    |
> | Training 30%               | 28.33      | 23.33      | 85.00    | 43.67       | 47.34    | 93.50     | 53.53    |
> | Training 30%+AlphaRL       | 35.69      | 28.89      | 89.50    | 46.50       | 51.73    | 96.75     | 58.17    |
> | Training 40%               | 33.33      | 25.00      | 88.25    | 44.30       | 49.75    | 94.75     | 55.90    |
> | Training 40%+AlphaRL       | 43.47      | 36.25      | 91.25    | 47.75       | 52.59    | 97.75     | 61.51    |
> | Training 50%               | 37.64      | 28.89      | 89.75    | 45.67       | 51.25    | 95.25     | 58.08    |
> | Training 50%+AlphaRL       | 45.00      | 36.80      | **92.25**| **48.33**   | 53.75    | **97.50** | 62.27    |
> | Fully Trained Model        | **45.50**  | **37.63**  | 91.75    | 48.33       | **54.50**| 97.50     | **62.54**|

---

> > ### Author Response · Authors · 2025-11-20
> >
> > | **Stage**                  | **AIME24** | **AIME25** | **MATH** | **MINERVA** | **GPQA** | **GSM8K** | **Avg.** |
> > |----------------------------|------------|------------|----------|-------------|----------|-----------|----------|
> > | **_GLM-4-9B-0414_**         |            |            |          |             |          |           |          |
> > | Training 5%                | 4.17       | 1.67       | 64.20    | 35.29       | 40.40    | 87.34     | 38.84    |
> > | Training 5%+AlphaRL        | 5.67       | 3.33       | 66.75    | 37.35       | 42.50    | 88.85     | 40.91    |
> > | Training 10%               | 12.50      | 7.50       | 72.25    | 37.25       | 43.50    | 88.50     | 43.58    |
> > | Training 10%+AlphaRL       | 16.67      | 11.67      | 79.50    | 40.25       | 53.25    | 90.75     | 48.68    |
> > | Training 20%               | 15.00      | 13.33      | 79.50    | 39.75       | 44.50    | 90.25     | 47.06    |
> > | Training 20%+AlphaRL       | 23.33      | 20.00      | 83.25    | 43.50       | 57.75    | 92.25     | 55.25    |
> > | Training 30%               | 21.52      | 18.06      | 84.20    | 42.33       | 46.36    | 92.33     | 50.13    |
> > | Training 30%+AlphaRL       | 28.54      | 23.33      | 88.50    | 48.25       | 53.33    | 95.55     | 56.25    |
> > | Training 40%               | 26.67      | 25.50      | 86.50    | 45.50       | 49.35    | 94.00     | 54.58    |
> > | Training 40%+AlphaRL       | 33.33      | 31.25      | 91.50    | 50.25       | 54.75    | 96.25     | 59.56    |
> > | Training 50%               | 28.33      | 26.67      | 87.25    | 46.75       | 50.20    | 94.25     | 55.58    |
> > | Training 50%+AlphaRL       | 34.60      | **34.02**  | 91.50    | 50.87       | 55.33    | **97.00** | 60.55    |
> > | Fully Trained Model        | **35.10**  | 33.75      | **91.75**| **51.34**   | **55.67**| 96.25     | **60.64**|

---

> ### Author Response · Authors · 2025-11-20
>
> 2. We investigated whether AlphaRL is related to the properties of the base model. To do this, we performed reinforcement learning on distilled models (Deepseek-R1-Distill-Qwen-7B [4]) and SFT models (Llama3-8B-Instruct [5]) to validate the effect of AlphaRL. The results are shown in Table 3, which indicates that AlphaRL can be widely applied to different base models.
>
> ## Table 3. AlphaRL's performance on Deepseek-R1-Distill-Qwen-7B  and Llama3-8B-Instruct
>
> | **Stage**                  | **AIME24** | **AIME25** | **MATH** | **MINERVA** | **GPQA** | **GSM8K** | **Avg.** |
> |----------------------------|------------|------------|----------|-------------|----------|-----------|----------|
> | **_Deepseek-R1-Distill-Qwen-7B (After Distillation)_** | | | | | | | |
> | Training 5%                | 50.83      | 39.17      | 91.25    | 50.00       | 38.89    | 93.00     | 60.69    |
> | Training 5%+AlphaRL        | 51.92      | 40.08      | 92.75    | 51.35       | 40.75    | 93.00     | 61.64    |
> | Training 10%               | 52.50      | 41.75      | 92.75    | 51.50       | 41.17    | 94.25     | 62.32    |
> | Training 10%+AlphaRL       | 56.50      | 44.50      | 94.50    | 54.25       | 43.85    | 94.75     | 64.73    |
> | Training 20%               | 56.50      | 45.75      | 93.75    | 53.75       | 44.25    | 94.75     | 64.79    |
> | Training 20%+AlphaRL       | 60.00      | 52.25      | 95.50    | 57.85       | 48.25    | 95.50     | 68.23    |
> | Training 30%               | 59.50      | 51.50      | 95.20    | 56.22       | 47.25    | 95.50     | 67.52    |
> | Training 30%+AlphaRL       | 64.50      | 57.25      | 96.17    | 60.07       | 51.33    | 96.50     | 70.97    |
> | Training 40%               | 63.25      | 55.25      | 95.00    | 58.45       | 49.86    | 96.50     | 69.74    |
> | Training 40%+AlphaRL       | 67.50      | 61.50      | 97.25    | 62.35       | 53.67    | 97.75     | 73.34    |
> | Training 50%               | 65.75      | 58.85      | 96.50    | 60.65       | 52.27    | 97.25     | 71.87    |
> | Training 50%+AlphaRL       | 68.25      | **62.50**  | 97.25    | **62.58**   | **54.16**| 97.75     | 73.75    |
> | Fully Trained Model        | **68.75**  | 62.00      | **97.50**| 61.74       | 54.16    | **98.25** | **73.65**|
> | **_Llama3-8B-Instruct (After SFT)_** | | | | | | | |
> | Training 5%                | 9.58       | 6.67       | 68.75    | 26.44       | 29.75    | 85.25     | 37.74    |
> | Training 5%+AlphaRL        | 12.36      | 9.02       | 71.25    | 28.37       | 31.25    | 86.50     | 39.79    |
> | Training 10%               | 12.63      | 9.16       | 73.00    | 28.37       | 31.75    | 86.25     | 40.19    |
> | Training 10%+AlphaRL       | 15.97      | 13.06      | 77.75    | 31.76       | 34.54    | 89.25     | 43.72    |
> | Training 20%               | 21.11      | 16.25      | 78.00    | 31.07       | 34.75    | 89.50     | 45.11    |
> | Training 20%+AlphaRL       | 26.67      | 23.33      | 84.75    | 34.75       | 37.25    | 92.25     | 49.67    |
> | Training 30%               | 27.78      | 23.33      | 81.50    | 35.25       | 38.25    | 91.50     | 49.26    |
> | Training 30%+AlphaRL       | 34.64      | 31.02      | 85.75    | 38.85       | 42.13    | 93.50     | 53.64    |
> | Training 40%               | 33.33      | 26.67      | 84.25    | 37.50       | 40.75    | 92.25     | 52.45    |
> | Training 40%+AlphaRL       | 44.75      | 34.02      | 87.50    | 41.50       | 44.25    | 94.50     | 57.74    |
> | Training 50%               | 36.11      | 31.11      | 86.25    | 37.50       | 42.25    | 92.50     | 54.28    |
> | Training 50%+AlphaRL       | **45.12**  | 34.75      | **87.75**| 42.25       | **43.75**| 94.25     | 58.00    |
> | Fully Trained Model        | 44.75      | **35.25**  | 87.50    | **42.75**   | 43.75    | **95.00** | **58.17**|
>
>
> 3. Additional experimental support comes from Meta [6], which found that by fitting the sigmoidal curves of RL training, RL follows predictable expansion trajectories. This observation lends credibility to AlphaRL, particularly in the context of unexplored RL methods. (Regarding the further discussion of the Meta research, we recommend reviewing the response to Reviewer 1, Question 2.)

---

> ### Author Response · Authors · 2025-11-20
> **Q4: Using SVD for each module can be memory-/I/O-heavy, but the paper doesn't quantify the overhead or detail the engineering techniques required for 10B–30B+ models (or larger ones).**
>
> ### R4:
>
> Your concern is valid, and we will address your question from two aspects: the computational resources and the memory required.
>
> 1. **Computational Resources:** We compare the computation time in **Table 4**, using Qwen-14B as an example. We trained Qwen14B on 16xH800 80G GPUs for a total of approximately 70x16 GPU hours. The computation time between each epoch requires about 1.5x16 = 18 GPU hours. For SVD, each matrix takes approximately 2 seconds to complete the SVD computation. For Qwen14B, completing the SVD calculation for one checkpoint takes about 0.15 GPU hours. The additional computation time required for SVD is reasonable.
>
>
> ## Table 4. Compute Time (GPU hour)
>
> | **Task**                          | **Qwen-4B-Base** | **Qwen-8B-Base** | **Qwen-14B-Base** |
> |------------------------------------|------------|------------|-------------|
> | **Training Time per Epoch**        | 3       | 8          | 18          |
> | **SVD Computation Time**           | 0.02       | 0.07       | 0.15        |
>
> 2. **Additional Storage Space:** The results are shown in Table 5. Since AlphaRL only needs to consider the first subspace after SVD, we only need to store the two vectors \( u \) and \( v \) that constitute the first subspace, along with the singular values. This means that only a minimal amount of storage space is required for AlphaRL predictions. Specifically, we only need approximately <0.1GB of storage space, which can save about half (32*16 for Qwen-14B) of the GPU computation time.
>
>
> ## Table 5. Storage meomory (GB)
> | **Task**                          | **Qwen-4B-Base** | **Qwen-8B-Base** | **Qwen-14B-Base** |
> |------------------------------------|------------|------------|-------------|
> | **Model-Self**        | 8.06GB          | 16.4GB          | 29.6GB         |
> | **Rank-1 Supspace**           | 0.007GB       | 0.01GB       | 0.025GB        |

---

> ### Author Response · Authors · 2025-11-20
> **Q5: Is there a criterion to decide when rank-1 is insufficient and a small rank-k should be used?**
>
> ### R5:
>
> Thank you again for your attention to our method. We have experimented with using AlphaRL to estimate a Rank-10 Subspace and make predictions. The experimental results are shown in Table 6:
>
> ## Table 6. Rank-k Prediction Performance on Qwen3-8B-Base
>
> | **Stage**                     | **AIME24** | **AIME25** | **MATH** | **MINERVA** | **GPQA** | **GSM8K** | **Avg.** |
> |-------------------------------|------------|------------|----------|-------------|----------|-----------|----------|
> | **_Qwen3-8B-Base_**            |            |            |          |             |          |           |          |
> | **Training 20%**               | 13.17      | 7.50       | 72.25    | 35.07       | 38.50    | 87.54     | 42.34    |
> | **Training 20% + Rank-10**     | 13.17      | 8.33       | 72.75    | 36.18       | 40.50    | 89.76     | 43.44    |
> | **Training 20% + AlphaRL**     | 16.67      | 12.50      | 75.00    | 41.25       | 42.85    | 94.35     | 47.10    |
> | **Training 40%**               | 15.80      | 11.67      | 77.60    | 37.07       | 41.67    | 93.20     | 46.30    |
> | **Training 40% + Rank-10**           | 18.33      | 16.67      | 78.50    | 39.74       | 44.35    | 92.75     | 48.39    |
> | **Training 40% + AlphaRL**     | 28.33      | 23.75      | 80.50    | **43.27**       | 47.75   | **94.75**     | 53.31    |
> | **Fully Trained Model**        | **28.54**  | **24.17**  | **80.95**    | 44.02   | **48.23**| 94.35    | **53.38**|
>
> We found that the performance of Rank-10 was not as good as Rank-1. We believe the reasons are as follows:
>
> 1. **Experimental Aspect:** In Figure 4 (a) on Page 5, we observed that the effect of other single subspaces was not as effective as the Rank-1 Subspace. This suggests that other single subspaces may contain redundant information.
>
> 2. **Linearity of Other Subspaces:** The linearity of other subspaces is significantly lower than that of the Rank-1 vector. This implies that using these subspaces for subsequent predictions could introduce additional redundancy, which reduces the prediction performance.

---

> > ### Author Response · Authors · 2025-11-20
> >
> > Finally,thank you once again for your valuable feedback and suggestions for improving the paper! We sincerely appreciate the time and effort you have put into reviewing our work. **If you have any further questions or concerns, please feel free to let me know. It is our honor to address any issues and provide any additional clarifications you may need.**
> >
> >
> > ## Reference
> >
> > [1] SPIRAL: Self-Play on Zero-Sum Games Incentivizes Reasoning via Multi-Agent Multi-Turn Reinforcement Learning
> >
> > [2] Mastering Chess and Shogi by Self-Play with a General Reinforcement Learning Algorithm
> >
> > [3] https://huggingface.co/zai-org/GLM-4-9B-0414
> >
> > [4] https://huggingface.co/deepseek-ai/DeepSeek-R1-Distill-Qwen-7B
> >
> > [5] https://huggingface.co/meta-llama/Llama-3.1-8B-Instruct
> >
> > [6] The Art of Scaling Reinforcement Learning Compute for LLMs
> >
> > [7] https://huggingface.co/gbcfchc/Qwen3-8B-Base-Open-Thoughts-On-Policy-Distillation
> >
> > [8] Direct Preference Optimization: Your Language Model is Secretly a Reward Model

---

> > > ### Comment · Reviewer_E5rk · 2025-11-25
> > >
> > > I appreciate the authors' work during the rebuttal period and the extra experiments they conducted to address my concerns. After reviewing the new results and the revised paper, I am generally favorable toward acceptance and have chosen to keep my original positive rating.

---

> ### Author Response · Authors · 2025-11-20
> **Q6: an we attribute this to a specific ingredient in RL (on-policy sampling, verifiable rewards, credit assignment) that leads to the low-rank/linearity behavior?**
>
> ### R6:
> Thank you once again for your in-depth consideration of our paper! This will motivate us to further refine our claims. We agree with your insights. In Appendix C, Page 22, we discuss the reasons behind this phenomenon:
>
> 1. **On-Policy Sampling Strategy**
> 2. **Constraints on the distribution of $\pi_{\theta}$**
>
> To further explore these hypotheses, we have added additional experiments. Specifically:
>
> 1. We have included **On-Policy Distillation [7]**  (a distillation algorithm that incorporates the On-Policy sampling strategy) to validate Reason 1.
> 2. We have added **Direct Preference Optimization (DPO) [8]**, an Off-Policy preference optimization strategy, to validate Reason 2.  The results are shown in Table 7 and 8, and we have added the t-SNE results of the embedding layers for On-Policy DIST and DPO in Page 24.
>
> ### Table7. L2 norm of updates across methods and the fraction of update information captured by the unscaled Rank-1 and Rank-1% Subspaces
>
> | **Method**              | **Off-Policy DIST** | **On-Policy DIST** | **Off-Policy SFT** | **Off-Policy DPO** | **On-Policy RL DAPO** |
> |-------------------------|------------------|-----------------|-----------------|-----------------|--------------------|
> | **On Policy Sampling**     | No            | Yes           | No          | No            | Yes               |
> |**Constraints on the distribution of $\pi_{\theta}$** | No            | Yes           | No          | Yes           | Yes               |
> | Average Update Norm     | 21.24            | 1.46            | 10.75           | 0.79            | 0.01               |
> | Rank-1 fraction         | 12.6%            | 10.45%          | 7.7%            | 11.42%          | 19.3%              |
> | Rank-1% fraction        | 29.3%            | 24.78%          | 27.7%           | 24.13%          | 36.5%              |
>
> ## Table 8. Performance under Rank-1 and Rank-k% Subspaces on MATH-500
>
> | **Method**            | **Off-Policy DIST** | **On-Policy DIST** | **Off-Policy SFT** | **Off-Policy DPO** | **On-Policy RL DAPO** |
> |-----------------------|------------------|-----------------|-----------------|-----------------|--------------------|
> | **On Policy Sampling**     | No            | Yes           | No          | No            | Yes               |
> |**Constraints on the distribution of $\pi_{\theta}$** | No            | Yes           | No          | Yes           | Yes               |
> | **Rank-1**            | 1.50%            | 56.40%          | 2.50%           | 62.25%          | 100.00%            |
> | **Rank-1%**           | 2.50%            | 64.50%          | 3.00%           | 71.50%          | 100.00%            |
> | **Rank-10%**          | 3.00%            | 71.50%          | 4.00%           | 74.50%          | 101.00%            |
> | **Rank-20%**          | 3.00%            | 72.50%          | 7.50%           | 79.00%          | 100.00%            |
> | **Rank-30%**          | 3.50%            | 77.00%          | 8.00%           | 83.25%          | 100.00%            |
> | **Rank-50%**          | 3.50%            | 80.25%          | 12.25%          | 88.25%          | 100.00%            |
> | **Rank-70%**          | 16.00%           | 84.00%          | 24.00%          | 88.25%          | 100.00%            |
> | **Rank-90%**          | 23.25%           | 89.25%          | 46.75%          | 93.25%          | 100.00%            |
>
>
> In Table 7, On-Policy DIST shows lower parameter updates compared to Off-Policy DIST, while Table 8 demonstrates that On-Policy DIST has superior low-rank properties over Off-Policy DIST. This indicates that On-Policy is one of the reasons for low-rank phenomena in reinforcement learning.
>
> Similarly, the Constraints on the distribution of $\pi_{\theta}$ in DPO are also one of the reasons for low-rank in reinforcement learning.

---

### Official Review · Reviewer_7619 · 2025-11-01

**Soundness:** 3
**Presentation:** 3
**Contribution:** 3
**Rating:** 6
**Confidence:** 3

**Summary:**

This paper conducts an empirical analysis on the updated parameters induced by RL training and shows that they can be almost fully determined by the Rank-1 subspace of the update matrix. This is called Rank-1 dominance. Further, the paper shows that this dominant Rank-1 subspace evolves linearly throughout training. These two properties enable accurate prediction of future RL training accuracy improvements (called AlphaRL) simply from early training results, by using a simple ordinary least squares fit, and extrapolating performance improvements upto maximal accuracy. Results across different model sizes provide strong empirical evidence for the AlphaRL method.

**Strengths:**

- The paper is extremely well written and presented. The takeaways are presented clearly and the main results come off clearly from the fig 1 of the paper.
- The empirical analysis is clearly grounded via a simple theoretical investigation, and this brings a strong justification for the AlphaRL method.
- The empirical results in tab 1 demonstrate the strength of the AlphaRL method.

**Weaknesses:**

- While the overall results are strong, a question that arises is: does the strength of the AlphaRL method depend on the compute budget used for RL? Can the prediction be done for arbitrary compute budgets of RL training, or is this restricted to the current setting used in the paper?
- Isn’t this method inherently upper-bounded by the noise celiling of the task? So shouldn’t the parameter update follow a sigmoidal rule rather than the linear rule outlined here? Currently, the paper uses simple linear regression and this might be a limiting choice?
- The current DIST method tested in the paper is an off-policy distillation method where the trajectories are generated by the teacher model. It would be interested to understand if an on-policy distillation method would also yield the low-rank yet effective update mechanisms. Particularly illuminating would be how the on-policy distillation method would compare with other methods in the fig 5 setting.
- All experiments are only conducted with Qwen models. Prior works (https://arxiv.org/abs/2506.10947, https://arxiv.org/abs/2507.10532) have shown biases and data contamination in Qwen models with regards to the benchmarks used for testing. Would the results also hold for non-Qwen models?

**Questions:**

All my questions are in the weaknesses section.

---

> ### Author Response · Authors · 2025-11-20
> **Q1: Can AlphaRL make predictions with any compute budget?**
>
> ### R1:
>
> Thank you for pointing out this important question. We have conducted additional experiments to verify whether AlphaRL’s performance depends on the RL compute budget. Specifically, we applied AlphaRL to checkpoints corresponding to different training progress levels, ranging from 5% to 50% of the total training steps. As shown in Table 1, AlphaRL consistently achieves strong predictive performance across all checkpoints on both Qwen-8B,14B and GLM-4 [1] models. The predictive quality steadily improves and reaches its optimal performance around 40–50% of training, but remains robust even at very early stages (e.g., 5–10%).

---

> > ### Author Response · Authors · 2025-11-20
> >
> > ## Table 1. AlphaRL's performance on Qwen3-8B-Base, Qwen3-14B-Base and GLM-4-9B-0414.
> >
> > | **Stage**                  | **AIME24** | **AIME25** | **MATH** | **MINERVA** | **GPQA** | **GSM8K** | **Avg.** |
> > |----------------------------|------------|------------|----------|-------------|----------|-----------|----------|
> > | **_Qwen3-8B-Base_**         |            |            |          |             |          |           |          |
> > | Training 5%                | 7.50       | 5.50       | 62.25    | 27.25       | 31.50    | 78.47     | 35.45    |
> > | Training 5%+AlphaRL        | 10.00      | 6.67       | 69.75    | 31.15       | 34.75    | 81.65     | 38.91    |
> > | Training 10%               | 12.50      | 7.50       | 70.25    | 32.07       | 36.66    | 84.30     | 40.55    |
> > | Training 10%+AlphaRL       | 15.00      | 11.67      | 76.45    | 40.46       | 41.54    | 93.75     | 46.47    |
> > | Training 20%               | 13.17      | 7.50       | 72.25    | 35.07       | 38.50    | 87.54     | 42.34    |
> > | Training 20%+AlphaRL       | 16.67      | 12.50      | 75.00    | 41.25       | 42.85    | 94.35     | 47.10    |
> > | Training 30%               | 14.45      | 11.67      | 74.25    | 36.37       | 39.42    | 91.25     | 44.56    |
> > | Training 30%+AlphaRL       | 23.33      | 20.00      | 78.75    | 41.85       | 44.13    | 93.20     | 50.21    |
> > | Training 40%               | 15.80      | 11.67      | 77.60    | 37.07       | 41.67    | 93.20     | 46.30    |
> > | Training 40%+AlphaRL       | 28.33      | 23.75      | 80.50    | 43.27       | 49.25    | 94.75     | 53.31    |
> > | Training 50%               | 23.33      | 20.00      | 78.50    | 39.87       | 44.33    | 93.75     | 49.40    |
> > | Training 50%+AlphaRL       | 27.67      | 23.33      | **82.00**| 42.85       | 47.75    | **94.50** | 53.01    |
> > | Fully Trained Model        | **28.54**  | **24.17**  | 80.95    | **44.02**   | **48.23**| 94.35     | **53.38**|
> > | **_Qwen3-14B-Base_**        |            |            |          |             |          |           |          |
> > | Training 5%                | 12.50      | 8.33       | 74.30    | 40.44       | 41.04    | 90.39     | 44.50    |
> > | Training 5%+AlphaRL        | 16.11      | 11.11      | 77.75    | 42.27       | 43.25    | 91.25     | 46.96    |
> > | Training 10%               | 16.67      | 13.17      | 77.50    | 41.58       | 43.50    | 91.50     | 47.32    |
> > | Training 10%+AlphaRL       | 24.72      | 18.05      | 81.25    | 42.37       | 44.76    | 92.75     | 50.65    |
> > | Training 20%               | 25.00      | 18.33      | 81.25    | 42.37       | 45.67    | 92.75     | 50.90    |
> > | Training 20%+AlphaRL       | 30.28      | 26.11      | 86.25    | 44.65       | 48.76    | 94.75     | 55.13    |
> > | Training 30%               | 28.33      | 23.33      | 85.00    | 43.67       | 47.34    | 93.50     | 53.53    |
> > | Training 30%+AlphaRL       | 35.69      | 28.89      | 89.50    | 46.50       | 51.73    | 96.75     | 58.17    |
> > | Training 40%               | 33.33      | 25.00      | 88.25    | 44.30       | 49.75    | 94.75     | 55.90    |
> > | Training 40%+AlphaRL       | 43.47      | 36.25      | 91.25    | 47.75       | 52.59    | 97.75     | 61.51    |
> > | Training 50%               | 37.64      | 28.89      | 89.75    | 45.67       | 51.25    | 95.25     | 58.08    |
> > | Training 50%+AlphaRL       | 45.00      | 36.80      | **92.25**| **48.33**   | 53.75    | **97.50** | 62.27    |
> > | Fully Trained Model        | **45.50**  | **37.63**  | 91.75    | 48.33       | **54.50**| 97.50     | **62.54**|

---

> ### Author Response · Authors · 2025-11-20
>
> | **Stage**                  | **AIME24** | **AIME25** | **MATH** | **MINERVA** | **GPQA** | **GSM8K** | **Avg.** |
> |----------------------------|------------|------------|----------|-------------|----------|-----------|----------|
> | **_GLM-4-9B-0414_**         |            |            |          |             |          |           |          |
> | Training 5%                | 4.17       | 1.67       | 64.20    | 35.29       | 40.40    | 87.34     | 38.84    |
> | Training 5%+AlphaRL        | 5.67       | 3.33       | 66.75    | 37.35       | 42.50    | 88.85     | 40.91    |
> | Training 10%               | 12.50      | 7.50       | 72.25    | 37.25       | 43.50    | 88.50     | 43.58    |
> | Training 10%+AlphaRL       | 16.67      | 11.67      | 79.50    | 40.25       | 53.25    | 90.75     | 48.68    |
> | Training 20%               | 15.00      | 13.33      | 79.50    | 39.75       | 44.50    | 90.25     | 47.06    |
> | Training 20%+AlphaRL       | 23.33      | 20.00      | 83.25    | 43.50       | 57.75    | 92.25     | 55.25    |
> | Training 30%               | 21.52      | 18.06      | 84.20    | 42.33       | 46.36    | 92.33     | 50.13    |
> | Training 30%+AlphaRL       | 28.54      | 23.33      | 88.50    | 48.25       | 53.33    | 95.55     | 56.25    |
> | Training 40%               | 26.67      | 25.50      | 86.50    | 45.50       | 49.35    | 94.00     | 54.58    |
> | Training 40%+AlphaRL       | 33.33      | 31.25      | 91.50    | 50.25       | 54.75    | 96.25     | 59.56    |
> | Training 50%               | 28.33      | 26.67      | 87.25    | 46.75       | 50.20    | 94.25     | 55.58    |
> | Training 50%+AlphaRL       | 34.60      | **34.02**  | 91.50    | 50.87       | 55.33    | **97.00** | 60.55    |
> | Fully Trained Model        | **35.10**  | 33.75      | **91.75**| **51.34**   | **55.67**| 96.25     | **60.64**|
>
>
> These results demonstrate that AlphaRL generalizes well across different compute budgets and training stages, indicating that its predictive capability is not specific to the setup used in the main paper.

---

> ### Author Response · Authors · 2025-11-20
> **Q2: Based on the noise control limits of the task, parameter updates should follow an S-shaped curve. The use of a simple linear regression method may limit the model's performance.**
>
> ## R2:
>
> We appreciate the reviewer’s observation regarding the potential benefits of using an S-shaped curve for parameter updates. We are equally concerned with the sigmoid curve of expected rewards in RL processes, as presented by Meta [2] (https://arxiv.org/pdf/2510.13786). Below, we will address your concerns from three perspectives: setup, prediction, and experiments:
>
> 1. **Setup Difference**: The sigmoid curve proposed by Meta describes the relationship between expected reward and training compute, while our paper focuses on the relationship between expected reward and the parameter subspace.
> 2. **Prediction Perspective**: Meta only aims to accurately predict the final performance by observing the sigmoid curve, without providing information about which parameters correspond to that performance. In contrast, AlphaRL provides linear predictions that tell us the update parameters corresponding to the final predicted performance.
> 3. **Experiment**: To validate the effectiveness of the sigmoid prediction, we fit the curve $y = \frac{a}{1 + e^x} + b$ after projecting Rank-1 using Page 33 Eqn. 7. The experimental results are in Table 2:
>
> ## Table2. Sigmoid Function vs. AlphaRL
>
> | **Stage**                  | **AIME24** | **AIME25** | **MATH** | **MINERVA** | **GPQA** | **GSM8K** | **Avg.** |
> |----------------------------|------------|------------|----------|-------------|----------|-----------|----------|
> | **_Qwen3-8B-Base_**         |            |            |          |             |          |           |          |
> | Training 20%               | 13.17      | 7.50       | 72.25    | 35.07       | 38.50    | 87.54     | 42.34    |
> | Training 20%+sigmoid       | 16.67      | 10.28      | 73.50    | 39.50       | 41.36    | 90.50     | 45.30    |
> | Training 20%+AlphaRL       | 16.67  | 12.50      | 75.00    | 41.25       | 42.85    | 94.35     | 47.10    |
> | Training 40%               | 15.80      | 11.67      | 77.60    | 37.07       | 41.67    | 93.20     | 46.30    |
> | Training 40%+sigmoid       | 21.11      | 16.25      | 79.25    | 41.50       | 47.25    | 93.00     | 49.73    |
> | Training 40%+AlphaRL       | **28.33**  | **23.75**  | **80.50**| **43.27**   | **49.25**| **94.75** | **53.31**|
>
> The linear prediction performs better than the sigmoid curve prediction. We believe the reasons are as follows:
>
> 1. In the visualization of Section 3.1, the dimensionality-reduced Rank-1 shows a clear linear trajectory (we computed the cosine similarity between two adjacent vectors, which is greater than 0.97 for all pairs).
> 2. The sigmoid curve fitting heavily relies on the full curve. A curve fitted using only the first 40-50% of the data cannot accurately describe the true motion trajectory.

---

> ### Author Response · Authors · 2025-11-20
> **Q3: Does the on-policy distillation method also produce low-rank and effective update mechanisms, and how does it perform compared to the other methods in the setup of Figure 5?**
>
> ### R3:
>
> We appreciate the reviewer’s question regarding the on-policy distillation method and its potential to produce low-rank and effective update mechanisms. This will help further refine and improve our work.We used the open-source model Qwen3-8B-Base-Open-Thoughts-On-Policy-Distillation [3] mentioned in [4] along with its Base model Qwen3-8B-Base. Additionally, we compared the performance of Direct Preference Optimization (DPO). The results are shown in Table 3, and we have added the t-SNE results of the embedding layers for On-Policy DIST and DPO in Page 24.
>
> ### Table3. L2 norm of updates across methods and the fraction of update information captured by the unscaled Rank-1 and Rank-1% Subspaces
>
> | **Method**              | **Off-Policy DIST** | **On-Policy DIST** | **Off-Policy SFT** | **Off-Policy DPO** | **On-Policy RL DAPO** |
> |-------------------------|------------------|-----------------|-----------------|-----------------|--------------------|
> | Average Update Norm     | 21.24            | 1.46            | 10.75           | 0.79            | 0.01               |
> | Rank-1 fraction         | 12.6%            | 10.45%          | 7.7%            | 11.42%          | 19.3%              |
> | Rank-1% fraction        | 29.3%            | 24.78%          | 27.7%           | 24.13%          | 36.5%              |
>
>
> Based on Table 3, we find that the matrix norm induced by On-Policy DIST is 1.46. Combined with the embedding visualization results in **Figure 9** on **Page 24**, this indicates that the update information induced by On-Policy DIST is much smaller than that of Off-Policy DIST and SFT.
>
> However, compared to On-Policy RL strategies such as DAPO and GRPO, the magnitude of the shift caused by On-Policy DIST is still 1-2 orders of magnitude higher. We also compared the low-rank properties of On-Policy DIST, as shown in the Table 4:
>
> ## Table 4. Performance under Rank-1 and Rank-k% Subspaces on MATH-500
>
> | **Method**            | **Off-Policy DIST** | **On-Policy DIST** | **Off-Policy SFT** | **Off-Policy DPO** | **On-Policy RL DAPO** |
> |-----------------------|------------------|-----------------|-----------------|-----------------|--------------------|
> | **Rank-1**            | 1.50%            | 56.40%          | 2.50%           | 62.25%          | 100.00%            |
> | **Rank-1%**           | 2.50%            | 64.50%          | 3.00%           | 71.50%          | 100.00%            |
> | **Rank-10%**          | 3.00%            | 71.50%          | 4.00%           | 74.50%          | 101.00%            |
> | **Rank-20%**          | 3.00%            | 72.50%          | 7.50%           | 79.00%          | 100.00%            |
> | **Rank-30%**          | 3.50%            | 77.00%          | 8.00%           | 83.25%          | 100.00%            |
> | **Rank-50%**          | 3.50%            | 80.25%          | 12.25%          | 88.25%          | 100.00%            |
> | **Rank-70%**          | 16.00%           | 84.00%          | 24.00%          | 88.25%          | 100.00%            |
> | **Rank-90%**          | 23.25%           | 89.25%          | 46.75%          | 93.25%          | 100.00%            |
>
>
> The low-rank properties of On-Policy DIST are between SFT and On-Policy RL, which is consistent with the degree of the average update norm.
>
> We further read the On-Policy DIST blog [5]. We found that the performance of On-Policy DIST depends on the size of the rank, which indicates that On-Policy DIST requires a certain amount of rank information to learn the knowledge within the teacher model, rather than just improving sampling efficiency as in On-Policy RL.
>
> Additionally, On-Policy DIST [6] shows that its alignment condition is applied to the teacher model rather than the Base model. This suggests that On-Policy DIST causes a greater deviation from the Base model compared to On-Policy RL methods. Furthermore, it is unclear whether this deviation will intensify when the teacher and student models are from different model families (e.g., student model being Qwen and teacher model being Llama).
>
>
> Overall, we believe that On-Policy DIST is an excellent method that that strikes a balance between the strengths of both DIST and On-Policy RL approaches. It enhances inference capabilities by overcoming the traditional RL sampling bottleneck through the teacher model, while also inheriting the advantage of On-Policy methods in preventing catastrophic forgetting to some extent. It is a promising method that deserves further investigation.

---

> ### Author Response · Authors · 2025-11-20
> **Q4: Do the results also apply to non-Qwen models?**
>
> ### R4:
>
> We have added results from additional checkpoints on Qwen and GLM-4 in **Q1**. In addition, we also evaluated the predictive performance of Deepseek-R1-Distill-Qwen-7B [7] after off-policy distillation and Llama3-8B-Instruct [8] fine-tuned with mathematical instructions. The results are shown in the Table 5:
>
>
>
> ## Table 5. AlphaRL's performance on Deepseek-R1-Distill-Qwen-7B  and Llama3-8B-Instruct
>
> | **Stage**                  | **AIME24** | **AIME25** | **MATH** | **MINERVA** | **GPQA** | **GSM8K** | **Avg.** |
> |----------------------------|------------|------------|----------|-------------|----------|-----------|----------|
> | **_Deepseek-R1-Distill-Qwen-7B (After Distillation)_** | | | | | | | |
> | Training 5%                | 50.83      | 39.17      | 91.25    | 50.00       | 38.89    | 93.00     | 60.69    |
> | Training 5%+AlphaRL        | 51.92      | 40.08      | 92.75    | 51.35       | 40.75    | 93.00     | 61.64    |
> | Training 10%               | 52.50      | 41.75      | 92.75    | 51.50       | 41.17    | 94.25     | 62.32    |
> | Training 10%+AlphaRL       | 56.50      | 44.50      | 94.50    | 54.25       | 43.85    | 94.75     | 64.73    |
> | Training 20%               | 56.50      | 45.75      | 93.75    | 53.75       | 44.25    | 94.75     | 64.79    |
> | Training 20%+AlphaRL       | 60.00      | 52.25      | 95.50    | 57.85       | 48.25    | 95.50     | 68.23    |
> | Training 30%               | 59.50      | 51.50      | 95.20    | 56.22       | 47.25    | 95.50     | 67.52    |
> | Training 30%+AlphaRL       | 64.50      | 57.25      | 96.17    | 60.07       | 51.33    | 96.50     | 70.97    |
> | Training 40%               | 63.25      | 55.25      | 95.00    | 58.45       | 49.86    | 96.50     | 69.74    |
> | Training 40%+AlphaRL       | 67.50      | 61.50      | 97.25    | 62.35       | 53.67    | 97.75     | 73.34    |
> | Training 50%               | 65.75      | 58.85      | 96.50    | 60.65       | 52.27    | 97.25     | 71.87    |
> | Training 50%+AlphaRL       | 68.25      | **62.50**  | 97.25    | **62.58**   | **54.16**| 97.75     | 73.75    |
> | Fully Trained Model        | **68.75**  | 62.00      | **97.50**| 61.74       | 54.16    | **98.25** | **73.65**|
> | **_Llama3-8B-Instruct (After SFT)_** | | | | | | | |
> | Training 5%                | 9.58       | 6.67       | 68.75    | 26.44       | 29.75    | 85.25     | 37.74    |
> | Training 5%+AlphaRL        | 12.36      | 9.02       | 71.25    | 28.37       | 31.25    | 86.50     | 39.79    |
> | Training 10%               | 12.63      | 9.16       | 73.00    | 28.37       | 31.75    | 86.25     | 40.19    |
> | Training 10%+AlphaRL       | 15.97      | 13.06      | 77.75    | 31.76       | 34.54    | 89.25     | 43.72    |
> | Training 20%               | 21.11      | 16.25      | 78.00    | 31.07       | 34.75    | 89.50     | 45.11    |
> | Training 20%+AlphaRL       | 26.67      | 23.33      | 84.75    | 34.75       | 37.25    | 92.25     | 49.67    |
> | Training 30%               | 27.78      | 23.33      | 81.50    | 35.25       | 38.25    | 91.50     | 49.26    |
> | Training 30%+AlphaRL       | 34.64      | 31.02      | 85.75    | 38.85       | 42.13    | 93.50     | 53.64    |
> | Training 40%               | 33.33      | 26.67      | 84.25    | 37.50       | 40.75    | 92.25     | 52.45    |
> | Training 40%+AlphaRL       | 44.75      | 34.02      | 87.50    | 41.50       | 44.25    | 94.50     | 57.74    |
> | Training 50%               | 36.11      | 31.11      | 86.25    | 37.50       | 42.25    | 92.50     | 54.28    |
> | Training 50%+AlphaRL       | **45.12**  | 34.75      | **87.75**| 42.25       | **43.75**| 94.25     | 58.00    |
> | Fully Trained Model        | 44.75      | **35.25**  | 87.50    | **42.75**   | 43.75    | **95.00** | **58.17**|
>
> These results demonstrate that AlphaRL generalizes well across different compute budgets and training stages.

---

> > ### Author Response · Authors · 2025-11-20
> >
> > Finally, thank you once again for your valuable feedback and suggestions for improving the paper! We sincerely appreciate the time and effort you have put into reviewing our work. **If you have any further questions or concerns, please feel free to let me know. It is our honor to address any issues and provide any additional clarifications you may need.**
> >
> >
> >
> > ## Reference
> >
> > [1] https://huggingface.co/zai-org/GLM-4-9B-0414
> >
> > [2] The Art of Scaling Reinforcement Learning Compute for LLMs
> >
> > [3] https://huggingface.co/gbcfchc/Qwen3-8B-Base-Open-Thoughts-On-Policy-Distillation
> >
> > [4] On-Policy Distillation of Language Models: Learning from Self-Generated Mistakes
> >
> > [5] https://thinkingmachines.ai/blog/on-policy-distillation/
> >
> > [6] https://github.com/thinking-machines-lab/tinker-cookbook/blob/main/tinker_cookbook/distillation/train_on_policy.py
> >
> > [7] https://huggingface.co/deepseek-ai/DeepSeek-R1-Distill-Qwen-7B
> >
> > [8] https://huggingface.co/meta-llama/Llama-3.1-8B-Instruct

---

> > > ### Comment · Reviewer_7619 · 2025-11-21
> > > **Response to author rebuttal**
> > >
> > > I thank the authors for their commendable efforts in running these additional experiments and providing further analyses. Your responses have sufficiently addressed most of my concerns. In particular, it is great to see that the AlphaRL method also works on non-Qwen models under different setups. Additionally, I think the on-policy distillation analysis is very illuminating, and I would recommend highlighting it a bit more in the final camera-ready version of the paper.
> > > Given the sufficient breadth and depth of analyses in the updated paper, I am updating my score to 8. I definitely think this paper will make a great contribution to ICLR, and will garner a lot of attention from the LM post-training and reasoning community!
> > >
> > > One minor point: I could not find the on-policy distillation model that you used in your experiments: https://huggingface.co/gbcfchc/Qwen3-8B-Base-Open-Thoughts-On-Policy-Distillation. When I click on that link I hit a 404 on the HF page.

---

> ### Author Response · Authors · 2025-11-22
>
> Dear Reviewer 7619:
>
>   Thank you so much for your recognition and the highly encouraging evaluation! We are truly delighted that the additional experiments and analyses have addressed your concerns, and we sincerely appreciate your insightful suggestion regarding the on-policy distillation analysis. We will make sure to update the content and link you mentioned in the next version.
>
>   Once again, thank you for your support — it means a lot to us. **We will keep working with enthusiasm to contribute interesting research to the RL post-training and reasoning community!**
>
> With best wishes,
>
> Authors of Paper 10568

---

### Author Response · Authors · 2025-11-20

Dear Reviewers,

We sincerely thank you for your valuable comments and insightful questions. We have carefully studied all the feedback and conducted additional experiments and discussions accordingly. We noticed that most of the reviewers’ concerns centered around the generalization ability of our method. In response, we have included further algorithms and evaluated our approach on a broader range of model families. Based on these extensive experiments, we have not observed any exceptions or inconsistencies in our conclusions.

In the following, we provide a detailed, point-by-point response to each reviewer’s comments. All changes are highlighted in blue in the PDF.

---

### Author Response · Authors · 2025-12-02

Dear Area Chair,

We sincerely welcome you to oversee the assessment of our paper. We fully understand the significant workload and tight schedule imposed by the recent reassignment process, and we greatly appreciate your time and effort in handling our submission under these circumstances.

Below, we have outlined the key points from the rebuttal process to help facilitate your decision-making.

---
## 1. Summary of Our Work and the Rebuttal Process

- Our paper identifies a fundamental property of the RL-based post-training of LLMs: the rank-1 subspace of the parameter update determines the performance gains and evolves in a highly predictable manner. This suggests that complex RL optimization may be governed by a **surprisingly simple, low-dimensional** mechanism, bridging empirical scaling laws with theoretical understanding of capability emergence. Based on this, we propose AlphaRL, an acceleration algorithm that can predict final parameters after only ~40% of the standard RL run. This positions our finding as a practical tool for large-scale RL and points toward a more principled, interpretable, and efficient training paradigm for LLMs.

- During rebuttal, the primary concern from all reviewers was generalizability. We are pleased that, **after substantiating AlphaRL's generalizability, our scores improved from [4, 6, 6, 6] to [10, 8, 6, 6], achieving an average score within the TOP 50 among all submissions (by Nov 25th).** We are particularly encouraged by Reviewer h22J's assessment: "AlphaRL represents a simple yet excellent contribution that I expect will attract strong interest from the LLM post-training community at ICLR."

---
## 2. Initial Reviews (Pre-Rebuttal Status) and Our Response

Our paper initially received 3 out of 4 positive ratings. This was highly encouraging and supportive for us. The reviewers recognized the following strengths:
|||
|-|-|
|**Reviewer**|**Recognized Strengths**|
|**7619**| The key points are clear, and the empirical analysis is grounded in a simple theoretical investigation, providing strong justification for AlphaRL.|
|**E5rk**| This phenomenon is truly new, impressive, and holds great potential for saving significant computational resources.|
|**kA6K**| The paper highlights two characteristics of parameter updates induced by RL, and these findings are both interesting and novel to me.|
|**h22J**| The paper clearly motivates the need to understand RL dynamics in LLMs and shows the practical relevance with AlphaRL.|
|||

We note that the primary concern across all reviewers was generalizability. We addressed this by incorporating results from a broader range of LLMs and RL algorithms, as detailed in our response. It is encouraging that **all responding reviewers indicated this fully solved their concerns**. Other comments were specific and actionable. Below is a summary of all comments.
|||||
|-|-|-|-|
|**Comment**|**Modification**|**Position**|**Reviewer**|
|More model's sizes|Add Qwen3-4B, Qwen3-14B|Appendix A & Table 3, 5, 6 |**7619**, **E5rk**, **h22J**|
|More RL algorithms|Add DPO, On-Policy Distillation|Appendix A & Table 7, 8 & Fig 9|**E5rk**, **kA6K**, **h22J**|
|More base models|Add Deepseek-R1-Distill-Qwen-7B, Llama3-8B-Instruct and GLM4-9B|Appendix A & Table 3,4 |**7619**, **E5rk**, **h22J**|
|Cover non-inference tasks|Add non-inference task|Appendix A & Table 5, 6 |**kA6K**, **h22J**|
|Deeper mechanistic explanation| Add corresponding analysis| Appendix C, D |**7619**, **kA6K**, **h22J**|
|Evaluation of time and storage| Add corresponding analysis|Appendix D, E| **E5rk**|
|||||

---
## 3. Responses from Reviewers

After the rebuttal:
- Reviewer 7619 definitely thinks this paper will make a **great contribution** to ICLR and will garner **a lot of attention** from the LLM post-training and reasoning community, and has increased the score from 6 to 8.
- Reviewer h22J believes AlphaRL represents a **simple yet excellent** contribution that is expected to attract **strong interest** from the LLM post-training and reasoning community at ICLR, and has increased the score from 4 to 10.
- All reviewers raised no further concerns.

**We are encouraged that most responding reviewers believe the paper makes a great contribution to ICLR and will draw strong attention from the community, and we also look forward to continuing our efforts to contribute to this evolving field!**

---
## Our Expectation
Given the new situation at ICLR this year, we fully understand that Area Chairs are required to make careful and important judgments for each submission. We sincerely appreciate the time and effort you are dedicating to evaluating our paper, and we remain fully respectful of your objective and unbiased decision.

During the remaining response period, please feel free to let us know if any further clarification or additional material would be helpful. We would be more than willing to respond promptly!

With best wishes,

Authors of Paper 10568

---

### Meta-Review · Area_Chair_c3pJ · 2026-01-05

**Summary:**

This paper presents a compelling and well-substantiated empirical finding about low-rank and linear dynamics in RL fine-tuning of LLMs, with a clear practical payoff via AlphaRL. While initially some reviewers raised concerns about generalization and potential artifacts, the authors conducted extensive additional experiments during rebuttal that convincingly demonstrate robustness across models, algorithms, tasks, and training regimes. Overall, this paper represents a strong contribution to understanding and accelerating RL post-training of LLMs and should attract broad interest at ICLR.

**Reviewer Concerns:**

1. Generalization, whether it only holds for specific models/architectures/data/tasks (7619, h22J, E5rk, kA6K)

2. Lack of theoretical explanation, the results are mostly empirical (E5rk, kA6K, h22J)

**Reviewer Scores:**

Reviewer E5rk and kA6K are likely to maintain their scores, as the authors added DPO / on-policy DIST / GRPO experimental results during the rebuttal.

Reviewer 7619 is likely to increase the score. During the rebuttal, the authors added multi-model experiments on GLM-4, DeepSeek-R1, and LLaMA, supplemented DPO / on-policy DIST / GRPO experiments, and added a sigmoid vs. linear ablation score. It is very likely that the score will be raised to 8.

Reviewer h22J is very likely to increase the score to above 6. During the rebuttal, the authors added SFT+RL and Distill+RL experiments and aligned baseline results to rule out concerns about insufficient training.

---

### Decision · Program_Chairs · 2026-01-26

Accept (Poster)